# Divalent anion-driven framework regulation in Zr-based halide solid electrolytes for all-solid-state batteries

Jae-Seung Kim [1,9], Daseul Han [2,9], Jinyeong Choe[1], Youngkyung Kim[3], Hae-Yong Kim[2], Soeul Lee [2], Jiwon Seo [1], Seung-Hui Ham[1], You-Yeob Song[1], Chang-Dae Lee[1], Juho Lee[4], Hiram Kwak [5], Jinsoo Kim [4], Yoon Seok Jung [5] ✉, Sung-Kyun Jung [3,6,7,8] ✉, Kyung-Wan Nam [2] ✉ & Dong-Hwa Seo [1] ✉

Research into solid electrolytes for all-solid-state batteries has intensified due to demand for safer and higher-energy-density batteries. Halide solid electrolytes are valued for their high ionic conductivity, oxidative stability, and ductility. Among them, $Li_2ZrCl_6$ is cost-effective but has a relatively lower $Li^+$ ionic conductivity (0.4 mS cm$^{-1}$ at 25 °C) compared to other halides, such as $Li_3InCl_6$ (> 1 mS cm$^{-1}$ at 25 °C). Here, we elucidate a fundamental mechanism of divalent-anion-driven framework modification that enables enhanced ionic conduction in Zr-based halides. Specifically, we demonstrate enhanced $Li^+$ conductivities for oxygen- ($0.8Li_2O-ZrCl_4$: 1.78 mS cm$^{-1}$ at 25 °C) and sulfur- ($0.8Li_2S-ZrCl_4$: 1.01 mS cm$^{-1}$ at 25 °C) substituted lattices. Synchrotron-based X-ray analyses identify distinct anionic sublattices and first-principles calculations reveal that divalent anions locally cluster within the lattice, inducing structural distortion and Li-site destabilization. These changes widen lithium conduction channels and alter the bonding environment, weakening and diversifying Li–Cl interactions. As a result, the energy landscape for lithium migration is flattened, leading to improved ionic conduction. These findings highlight design strategies for divalent-anion-driven framework regulation in halide solid electrolytes.

The increasing demand for sustainable development and Li-ion batteries (LIBs) for large-scale applications, such as electric vehicles (EVs) and energy storage systems (ESS), has led to the desire for safer and higher-energy-density batteries[1–3]. All-solid-state batteries (ASSBs) using inorganic solid electrolytes (SEs) are thus considered a promising battery system to replace conventional LIBs[4–6]. Over the past

decades, solid Li-ion conductors, such as sulfides and oxides, have been actively explored[7–10]. The outstanding (electro)chemical stability of oxide SEs (e.g., $Li_7La_3Zr_2O_{12}$ (LLZO)) has led to extensive studies; however, their inferior processability, caused by the brittle nature of the materials, impedes the fabrication of ASSBs[11–13]. The combined ductility and high ionic conductivity of sulfide SEs (e.g., $Li_6PS_5Cl$

[1]Department of Materials Science and Engineering, Korea Advanced Institute of Science and Technology (KAIST), Daejeon, Republic of Korea. [2]Department of Energy and Materials Engineering, Dongguk University, Seoul, Republic of Korea. [3]Department of Materials Science and Engineering, Seoul National University (SNU), Seoul, Republic of Korea. [4]Department of Energy Science and Engineering, Daegu Gyeongbuk Institute of Science and Technology (DGIST), Daegu, Republic of Korea. [5]Department of Chemical and Biomolecular Engineering, Yonsei University, Seoul, Republic of Korea. [6]School of Transdisciplinary Innovations, Seoul National University (SNU), Seoul, Republic of Korea. [7]Research Institute of Advanced Materials (RIAM), Seoul National University (SNU), Seoul, Republic of Korea. [8]Institute for Rechargeable Battery Innovations Research, Seoul National University (SNU), Seoul, Republic of Korea. [9]These authors contributed equally: Jae-Seung Kim, Daseul Han. ✉e-mail: yoonsjung@yonsei.ac.kr; naecard@snu.ac.kr; knam@dongguk.edu; dseo@kaist.ac.kr

(LPSC)) has attracted many researchers in the development of materials; however, their intrinsic narrow electrochemical stability window still limits the use of general 4-V-class layered positive electrode materials (LiMO$_2$, M = Ni, Co, Mn)[14–16].

In the past few years, halide SEs (Li$_a$MX$_b$, M = metal) have emerged as promising catholytes due to their high oxidative limit ( > 4 V vs. Li/Li$^+$), which even surpasses that of oxide SEs ( ~ 4 V), and high ionic conductivity ( > 1 mS cm$^{-1}$) with ductile nature like sulfide SEs[17,18]. Initiated from cubic-close-packed (ccp) monoclinic Li$_3$YBr$_6$ (space group $C2/m$, 1.7 mS cm$^{-1}$ at 25 °C) and hexagonal-close-packed (hcp) trigonal Li$_3$YCl$_6$ (space group $P\bar{3}m1$, 0.51 mS cm$^{-1}$ at 25 °C), many compositions have been actively developed, including Li$_3$InCl$_6$ (1.5 mS cm$^{-1}$ at 25 °C), Li$_3$ScCl$_6$ (1.5 mS cm$^{-1}$ at 25 °C), and Li$_2$ZrCl$_6$ (0.40 mS cm$^{-1}$ at 25 °C)[19–24]. In addition, studies based on multi-metal applications (Li$_a$M$_{1-x}$Zr$_x$Cl$_b$, M = Y, Er, In, Sc, Fe. Cr, V, Yb, > 1 mS cm$^{-1}$ at room temperature) have been conducted[23,25–27], leading to the uncovering of design principles in Li–M–X halide SEs with ionic-potential-based structure dependency[28,29].

Meanwhile, oxygen incorporation in halide SEs has proven to be an effective strategy for designing superionic conductors and is currently one of the most critical design approaches applied to halide-based systems, including Nb and Ta oxychlorides[30]. This emphasizes the importance of accurately elucidating the structural impact of introducing divalent anions, which play a pivotal role in advancing material performance. Notably, this strategy has been generally applied to cost-effective Zr-based materials, and recent studies have reported the oxychlorination of lithium zirconium chloride[31–33]. Halide nanocomposite solid electrolytes (ZrO$_2$–2Li$_2$ZrCl$_6$ and 0.44ZrO$_2$–1.26LiCl–0.56Li$_2$ZrCl$_6$, 1.1 mS cm$^{-1}$ and 1.3 mS cm$^{-1}$ at 30 °C), which exhibit interfacial superionic conduction, have been designed using a conduction enhancement strategy via interfacial anionic substitution[31]. Amorphous Li–Zr–Cl–O phases (Li$_{1.75}$ZrCl$_{4.75}$O$_{0.5}$ and Li$_3$ZrCl$_4$O$_{1.5}$, 2.42 mS cm$^{-1}$ and 1.35 mS cm$^{-1}$ at 25 °C) have also been reported, highlighting that the increase in the amorphous phase is key to the improvement of ion conduction[32,33]. These strategies represent powerful approaches to developing cost-effective yet high-performance materials.

However, the challenge lies in the unclear understanding of divalent anion incorporation in Zr-based halide SEs, with ongoing debates on whether the synthesized compounds exist in an amorphous state, adopt a ccp structure, or form a nanocrystalline phase[34,35]. Although the synthesized phase may vary depending on experimental conditions, accurate phase identification and a clear elucidation of how the observed phase contributes to enhanced material properties are essential. Notably, the formation of structure and conduction mechanisms are distinctly different from the superionic conductive oxychloride SEs (Li–M–Cl–O, M = Ta, Nb) currently being actively researched[30,36–38], and a fundamental understanding of how heterogeneous divalent anions operate within the halide frameworks remains critically unexplored, particularly regarding the effect of divalent anions.

Herein, we systematically investigate how divalent anions (not only oxygen but also sulfur) modulates the anion sublattice and associated superionic conductivity of Zr-based halide SEs ($x$Li$_2$A–ZrCl$_4$, ($x$ = 0.6, 0.8, 1.0, 1.2, 1.6, 2.0; A = O, S)). By regulating the anion sublattice via the incorporation of divalent anion, we propose 0.8Li$_2$O–ZrCl$_4$ and 0.8Li$_2$S–ZrCl$_4$ with enhanced ionic conductivities of 1.78 and 1.01 mS cm$^{-1}$, respectively, at 25 °C. We addressed limitations in structural interpretation via conventional lab-based X-ray diffraction (XRD) by employing high-resolution synchrotron X-ray and pair distribution function (PDF) analyses, enabling a precise structural characterization of their hcp and ccp lattice structures and LiCl phase. Based on the precise analysis of the phase, we investigated the origins of fast Li-ion conduction in these

structures using density functional theory (DFT) calculations and ab initio molecular dynamics (AIMD) simulations. We identified the conduction mechanism of induced framework regulations by investigating lattice changes caused by divalent anions and the resulting variations in the lithium environment. These findings elucidate the fundamental design principles governing ionic conductivity enhancement in halide SEs and highlight the critical role of divalent-anion-driven framework regulation in the rational design of practical, high-performance ASSBs.

## Results

### Synthesis of $x$Li$_2$A–ZrCl$_4$ ($x$ = 0.6, 0.8, 1.0, 1.2, 1.6, 2.0, A = O, S)

The anionic sublattice of halide SEs is generally categorized by an anionic stacking sequence, either ccp or hcp[29]. Li$_2$ZrCl$_6$ (LZC) in both lattices can be synthesized using LiCl and ZrCl$_4$ precursors under varying synthetic conditions (ball-milling and heat treatment, Supplementary Fig. 1). Notably, the ionic conductivity of hcp-LZC (0.40 mS cm$^{-1}$ at 30 °C) exceeds that of ccp-LZC ( ~ 10$^{-3}$ mS cm$^{-1}$ at 30 °C)[23]. We used oxygen- or sulfur-based lithium compounds (Li$_2$O and Li$_2$S) to investigate the impact of divalent anions on the structure of $x$Li$_2$A–ZrCl$_4$ via a mechanochemical ball-milling method.

Figure 1a,b presents the XRD patterns measured by lab-source XRD equipment and the ionic conductivities of $x$Li$_2$O–ZrCl$_4$, respectively. When 0.6 ≤ $x$ ≤ 1.2, no crystalline phase was detected in the lab-XRD data ($\lambda$ = 1.5406 Å), which can be interpreted as amorphization of materials with ionic conductivity maximized at $x$ = 0.8 (1.78 mS cm$^{-1}$ at 25 °C, Supplementary Fig. 2). For 1.6 ≤ $x$ ≤ 2.0, a low-ionic-conductive phase, LiCl ($Fm\bar{3}m$), begins to form. However, synchrotron-based high-resolution XRD ($\lambda$ = 0.1665 Å) results reveal distinguishable intensive peaks (1.72°, 3.20° and 4.39°) corresponding to the hcp trigonal ($P\bar{3}m1$) for 0.6 ≤ $x$ ≤ 0.8 (Fig. 1c). Upon increasing the Li$_2$O ratio from 1.0 to 1.2, both crystallinity and ionic conductivity decreased (from 1.33 to 0.41 mS cm$^{-1}$ at 25 °C) with evolution of LiCl peaks. In addition, further increasing the Li$_2$O ratio from 1.6 to 2.0 leads to an increased peak intensity for the LiCl phase, confirming the transition to a less conductive crystalline LiCl phase. The synchrotron XRD results highlight that the nano-sized crystalline grain in the hcp phase can complicate phase analysis and that the resolution limits of lab-XRD may lead to misinterpretations regarding superionic phases. Thus, synchrotron XRD proves essential for accurately discerning phase evolution and understanding the conductivity behavior in these materials.

In $x$Li$_2$S–ZrCl$_4$, unlike in $x$Li$_2$O–ZrCl$_4$, the lab-source XRD patterns in Fig. 1d show similar peaks over the entire composition range (0.6 ≤ $x$ ≤ 2.0). The ionic conductivity, as shown in Fig. 1e, is maximized for 0.8Li$_2$S–ZrCl$_4$ (1.01 mS cm$^{-1}$ at 25 °C), which corresponds to the same lithium source ratio as in 0.8Li$_2$O–ZrCl$_4$. Synchrotron XRD patterns of $x$Li$_2$S–ZrCl$_4$ in Fig. 1f reveal three key features: First, ccp monoclinic ($C2/m$) and LiCl ($Fm\bar{3}m$) phases are difficult to distinguish because of their similar peak positions; however, there are subtle differences in the relative intensities (3.19° vs. 3.61°, 6.09° vs. 6.38° and 8.01° vs. 8.19°). Second, the peak shift from the ccp phase (0.6 ≤ $x$ ≤ 1.2) to the LiCl phase (1.6 ≤ $x$ ≤ 2.0) occurs near 5.09°, 7.27°, and 8.91° (yellow dotted line) with LiCl intensity increasing at 6.09°, 7.98°, and 9.59° (square symbol) as the Li$_2$S ratio increased from 1.2 to 2.0. Finally, unique peaks exclusively for the ccp phase appear at 2.85°, 4.70°, and 5.51° (star symbol) in the 0.6 ≤ $x$ ≤ 1.0 region. Overall, the ccp phase enhances ionic conductivity, but excess Li$_2$S content beyond $x$ = 1.2 results in the formation of the LiCl phase, reducing conductivity. Notably, while the LiCl phase appears in both $x$Li$_2$O–ZrCl$_4$ and $x$Li$_2$S–ZrCl$_4$ compounds at $x$ = 2, the structures exhibiting high ionic conductivity in 0.8Li$_2$A–ZrCl$_4$ differ: the oxygen-substituted compound adopts an hcp structure, whereas the sulfur-substituted one exhibits a ccp structure.

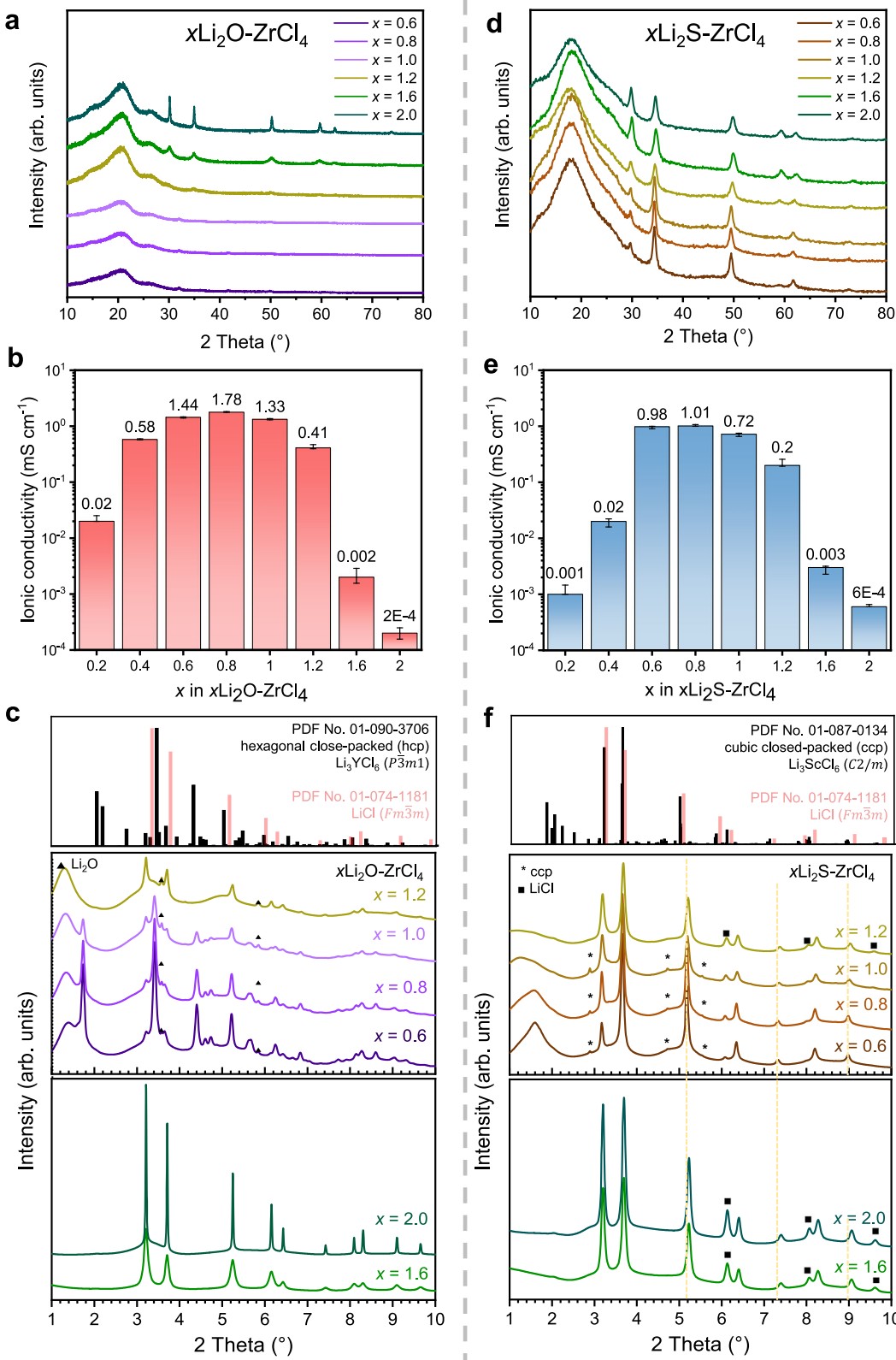

**Fig. 1 | Synthesis of $x$Li$_2$A–ZrCl$_4$ (A = O, S). a–c** Lab-source XRD patterns (**a**), ionic conductivities (**b**), and synchrotron XRD patterns (**c**) of $x$Li$_2$O–ZrCl$_4$ ($x$ = 0.6, 0.8, 1.0, 1.2, 1.6, 2.0). **d–f** Lab-source XRD patterns (**d**), ionic conductivities (**e**), and synchrotron XRD patterns (**f**) of $x$Li$_2$S–ZrCl$_4$ ($x$ = 0.6, 0.8, 1.0, 1.2, 1.6, 2.0). Error bars represent the standard deviation from $n$ = 4 independent ionic conductivity measurements.

## Local structure of $x$Li$_2$O–ZrCl$_4$

PDF analysis, X-ray absorption spectroscopy (XAS), Raman spectroscopy and high-resolution transmission electron microscopy (HRTEM) images are presented in Fig. 2 ($x$Li$_2$O–ZrCl$_4$) and Fig. 3 ($x$Li$_2$S–ZrCl$_4$),

respectively, enabling detailed analysis of the local structure of $x$Li$_2$A–ZrCl$_4$.

The PDF G(r) results for $x$Li$_2$O–ZrCl$_4$ in Fig. 2a indicate that the overall atomic pair distribution aligns with an hcp structure up to

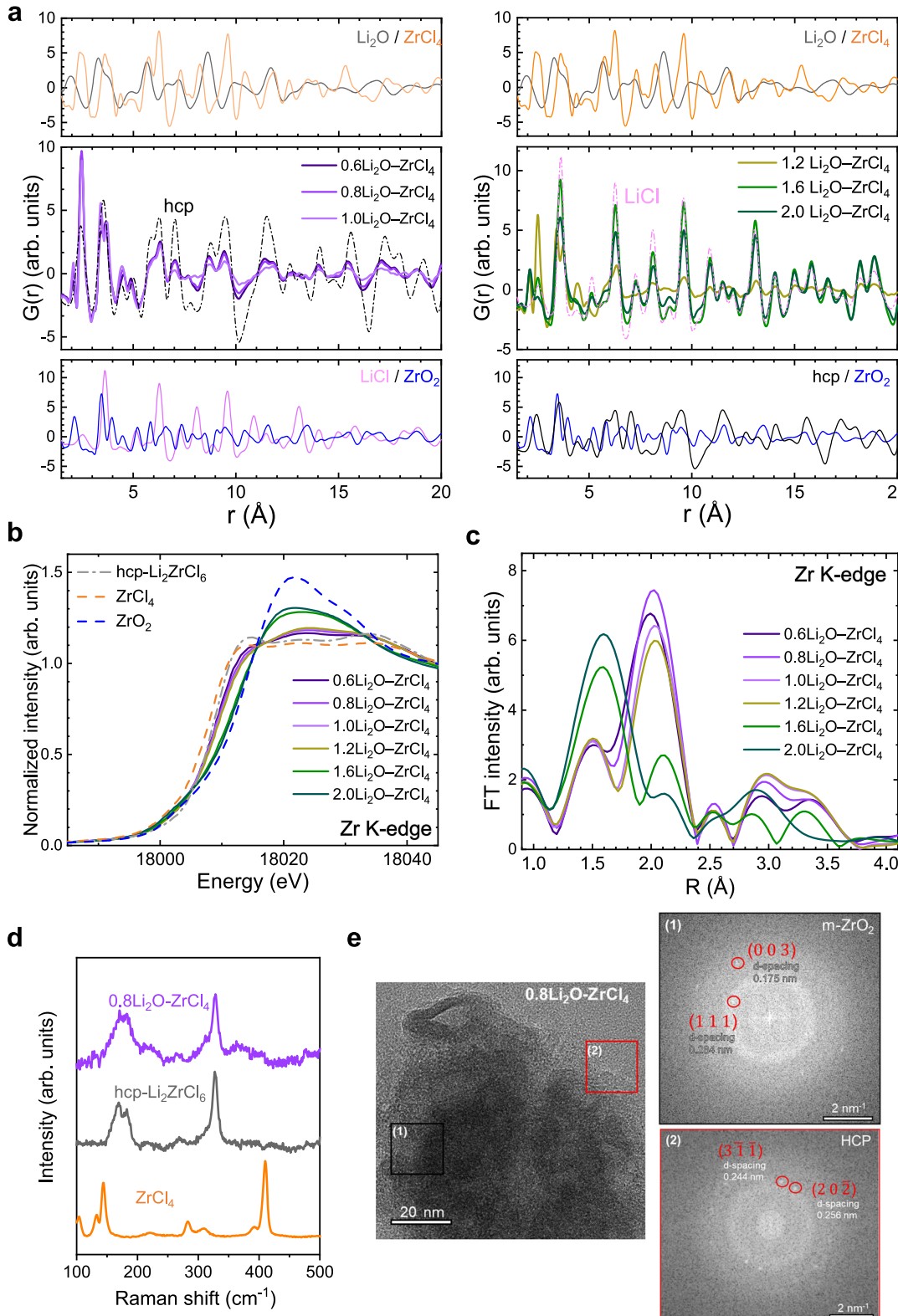

**Fig. 2 | Characterization of $x$Li$_2$O–ZrCl$_4$. a–c** PDF G(r) (**a**) normalized Zr K-edge XANES spectra (**b**) and Zr K-edge EXAFS spectra (**c**) of $x$Li$_2$O–ZrCl$_4$ ($x$ = 0.6, 0.8, 1.0, 1.2, 1.6, 2.0). **d** Raman spectra for ZrCl$_4$, hcp-LZC, and 0.8Li$_2$O–ZrCl$_4$. **e** HRTEM image with FFT patterns of 0.8Li$_2$O–ZrCl$_4$(scale bar: 20 nm) with corresponding FFT patterns (scale bar: 2 nm$^{-1}$).

$x \leq 1.2$; however, beyond $x = 1.2$, the overall distribution resembles that of LiCl. Additionally, the presence of a Zr–O pair in monoclinic ZrO$_2$ (Supplementary Fig. 3 and Supplementary Text 1) at 2.1 Å in all compositions suggests the formation of nanosized ZrO$_2$ domains.

Additionally, PDF fitting was conducted to estimate the phase fractions of ZrO$_2$ and the hcp phase (Supplementary Fig. 4 and Supplementary Table 1-6). For $x = 0.6$ and 0.8, the long-range region (15−30 Å), where only crystalline phases contribute, was well-reproduced using a single

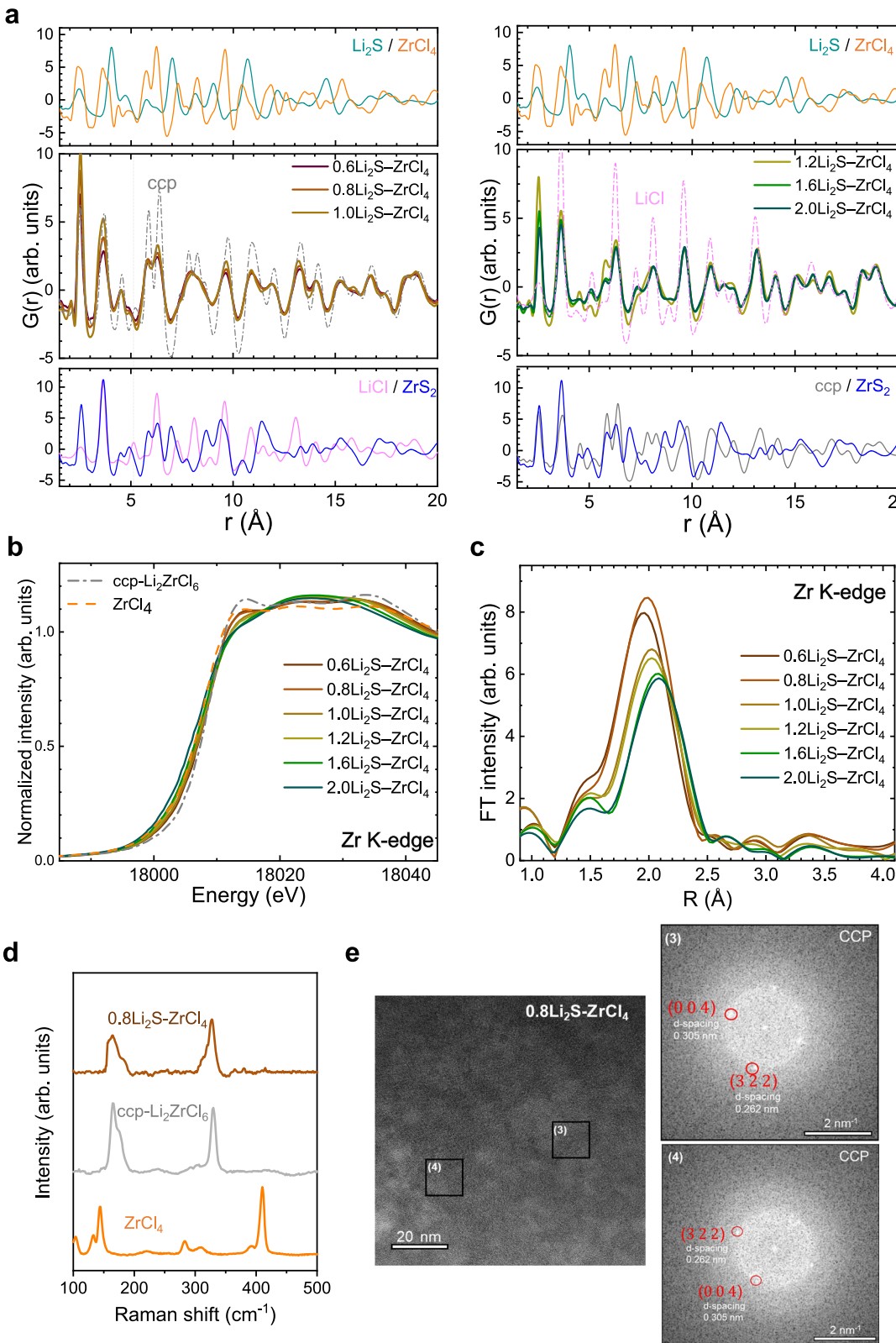

**Fig. 3 | Characterization of $x$Li$_2$S–ZrCl$_4$. a–c**, PDF $G(r)$ (**a**) normalized Zr K-edge XANES spectra (**b**) and Zr K-edge EXAFS spectra (**c**) of $x$Li$_2$S–ZrCl$_4$ ($x$ = 0.6, 0.8, 1.0, 1.2, 1.6, 2.0). **d** Raman spectra for ZrCl$_4$, ccp-LZC, and 0.8Li$_2$S–ZrCl$_4$. **e** HRTEM image with FFT patterns of 0.8Li$_2$S–ZrCl$_4$ (scale bar: 20 nm) with corresponding FFT patterns (scale bar: 2 nm$^{-1}$).

hcp model with ~50% M2–M3 site disorder, confirming the coexistence of crystalline hcp and short range ordered ZrO$_2$ domains. In contrast, for $x$ = 1.0, the fitting yielded a substantially higher R$_w$ value, suggesting progressive structural distortion of the hcp framework due to

oxygen incorporation and the onset of phase transformation (Supplementary Table 6).

The Zr-K edge X-ray absorption near-edge structure (XANES) spectra (Fig. 2b) further reveal that the absorption edge features

closely resemble those of the hcp-LZC phase up to $x = 1.2$. At higher $Li_2O$ levels, however, the edge features shift to match those of $ZrO_2$, with the hcp features disappearing. Furthermore, the extended X-ray absorption fine structure (EXAFS) spectra in Fig. 2c show overall positive peak shifts toward a longer region for the Zr–Cl bond in the hcp phase around 2.0 Å, as well as broadening of the Zr-Zr bonding peaks from 3.0 to 3.4 Å compared to bare hcp-LZC and $ZrO_2$, respectively (Supplementary Fig. 5). Also, our EXAFS fitting results show such elongation (Supplementary Fig. 6). This elongation of the Zr–Cl bond is attributed to oxygen substitution in $ZrCl_6$ octahedra, which strongly compensates for the positive charge of Zr, weakening the Zr–Cl bond and thereby increasing Zr–Cl bond length (Supplementary Fig. 7). Conversely, the substitution of Cl in $ZrO_2$ increases the Zr–Zr distance and local distortion due to the larger anion radius of Cl, suggesting anion exchange between $ZrO_2$ and $Li_2ZrCl_6$, consistent with our previous study[31]. Additionally, phase fraction for $ZrO_2$ in $xLi_2O$-$ZrCl_4$ ($x = 0.6$-$1.0$) is well aligned with our PDF fitting results, further supporting reliability of our refinement (Supplementary Table 7 and Supplementary Text 2).

In Fig. 2d, the Raman spectrum of $ZrCl_4$ exhibited characteristic features of polymeric zigzag chains with bridged octahedra, whereas hcp-LZC and $0.8Li_2O$-$ZrCl_4$ showed distinct peaks at $\approx 325$ $cm^{-1}$ and $\approx 161$ $cm^{-1}$, respectively, indicative of isolated octahedra in lattice[23]. Also, HRTEM image and fast Fourier transform (FFT) images confirm the presence of a discrete hcp phase with the $ZrO_2$ (Fig. 2e). Moreover, the $^7Li$ magic angle spinning-nuclear magnetic resonance (MAS-NMR) spectra in Supplementary Fig. 8 show two distinct lithium environments, indicating that Li coordination with Cl has relatively strong (by $ZrCl_6$ polyhedron) and weak (by $ZrO_xCl_{6−x}$ polyhedron) bonding environments. Overall, within the nanocomposite of nanoscopic $ZrO_2$ (or $Li_2O$) and hcp-LZC domains, interfacial oxygen substitution in hcp-LZC occurs with a non-uniform concentration gradient, reaching its higher level at their interfaces of $ZrO_2$ (or $Li_2O$) with hcp-LZC (Fig. 2e, Supplementary Fig. 9). The primary origin of the enhanced conductivity is these oxygen-substituted hcp-LZC phases.

## Local structure of $xLi_2S$–$ZrCl_4$

For $xLi_2S$–$ZrCl_4$, the PDF G(r) results in Fig. 3a reveal that the overall atomic pair distribution corresponds to the ccp structure up to $x \leq 1.2$ while the overall structure transforms to LiCl for compositions beyond $x = 1.2$. Notably, small peaks at ~4.98 Å, attributed to $ZrS_2$, suggest the presence of $ZrS_2$ with only short-range ordering. The PDF fitting results and phase fraction of $ZrS_2$ for $xLi_2S$–$ZrCl_4$ ($x = 0.6$-$1.0$) are shown in Supplementary Fig. 10 and Supplementary Table 8–13.

XANES spectra in Fig. 3b also shows that increasing $Li_2S$ over $x = 1.0$ leads to the disappearance of the ccp feature at ~18013 eV. The EXAFS results in Fig. 3c display a noticeable positive peak shift upon increasing $Li_2S$, with low fractions of $Li_2S$ ($x \leq 1.2$) showing a similar tendency of divalent-anion-driven bond elongation observed in $xLi_2O$–$ZrCl_4$, but within the ccp structure (Supplementary Fig. 11) and our EXAFS fitting further support such elongation (Supplementary Fig. 12). Additionally, the phase fraction of $ZrS_2$ refined by EXAFS fitting demonstrates similarity to those predicted by PDF, analogous to the case of O-substituted hcp phases. (Supplementary Table 14 and Supplementary Text 3,4). At higher $Li_2S$ ratios ($x \geq 1.6$), distinct peaks appear, markedly different from those observed at low $Li_2S$ levels. The bond length of Zr–S (2.57 Å) in $ZrS_2$, notably longer than the Zr–Cl bond (2.49 Å) in ccp-LZC, becomes more pronounced at higher fractions of $Li_2S$, highlighting the distinct characteristics of the $ZrS_2$ component (Supplementary Table 15). These findings suggest a phase transition from ccp to LiCl with $ZrS_2$ through the reaction $2Li_2S + ZrCl_4 \rightarrow 4LiCl + ZrS_2$ as increase of $Li_2S$ contents above $x = 1.6$.

Since the Zr octahedra are also isolated in the ccp lattice, the Raman spectrum in Fig. 3d also exhibited distinct peaks, differentiating it from $ZrCl_4$. Moreover, the HRTEM images with FFT pattern

(Fig. 3e) confirm a single ccp phase, indicating the homogeneous distribution of the sulfur-substituted ccp phase across the material. Overall analyses thus far demonstrate compelling evidence of a ccp phase for $0.8Li_2S$-$ZrCl_4$ with sulfur substitutions occurring within the bulk of the ccp structure rather than at the interface. This conclusion is supported by three key observations. First, although mechanochemically synthesized Zr-based SEs typically form hcp phases[23,24]; only ccp phases arise when sulfur is introduced, suggesting that the sulfur anion triggers the formation of a ccp anion sublattice. Second, DFT calculations show that the energy above the hull for $Li_{2+x}ZrCl_{6−x}S_x$ is relatively low (-11 meV atom$^{-1}$) compared with that for $Li_{2+x}ZrCl_{6−x}O_x$ (-63 meV atom$^{-1}$), as shown in Supplementary Table 16,17. Finally, no residual $Li_2S$ is detected in synchrotron-based analyses, indicating that all sulfur sources are consumed to form the ccp phase or amorphized $ZrS_2$. Additionally, the X-ray photoelectron spectroscopy (XPS) analysis (Supplementary Fig. 13) was presented to confirm the presence of bonding, while scanning electron microscope (SEM) images (Supplementary Fig. 14,15) were provided to examine the particle size.

Comprehensive analyses reveal the local structural characteristics and the structural features driven by the incorporation of divalent anions. The oxygen sources facilitate the formation of a nanocomposite exhibiting an hcp structure with oxygen substitution, optimizing interfacial oxygen incorporation to achieve enhanced ionic conductivity. In contrast, the sulfur sources are integrated into the bulk, where mechanochemical synthesis with sulfur promotes a phase transition of superionic conductive ccp lattice. These findings are further supported by the consistency between the experimentally identified reaction products and the phase compositions predicted by reaction energy calculations, reinforcing the reliability of phase fraction estimations across both theoretical and experimental approaches (Supplementary Table 18,19 with Supplementary Text 5-7).

## Ionic conduction properties and underlying mechanisms of $0.8Li_2A$–$ZrCl_4$ (A = O, S)

The Arrhenius plots (Fig. 4a) and Nyquist plots (Supplementary Fig. 16,17) of $0.8Li_2A$–$ZrCl_4$ indicate enhanced $Li^+$ ionic conductivity ($\sigma_{Li^+}$) of 1.78 mS cm$^{-1}$ ($0.8Li_2O$–$ZrCl_4$) and 1.01 mS cm$^{-1}$ ($0.8Li_2S$–$ZrCl_4$) at 25 ° with reduced activation energy ($E_a$) of 0.281 eV ($0.8Li_2O$–$ZrCl_4$) and 0.317 eV ($0.8Li_2S$–$ZrCl_4$) compared with those of $Li_2ZrCl_6$ (hcp: $\sigma_{Li^+}$ = 0.37 mS cm$^{-1}$ at 25 °C with $E_a$ = 0.346 eV and ccp: $\sigma_{Li^+}$ = 4.3 × 10$^{-3}$ mS cm$^{-1}$ at 25 °C with $E_a$ = 0.467 eV). Also, electrochemical impedance spectroscopy (EIS) fitting of $0.8Li_2A$–$ZrCl_4$ confirms that the observed ionic conduction originates from a single-phase component, corresponding to the oxygen substituted hcp phase and the sulfur substituted ccp phase, as previously confirmed by structural analysis (Supplementary Fig. 18). These findings are further corroborated by hopping rate-based conduction parameter analysis (Supplementary Fig. 19 **and** Supplementary Table 20), where we decoupled key descriptors governing $Li^+$ transport[39]. Notably, both $0.8Li_2O$–$ZrCl_4$ and $0.8Li_2S$–$ZrCl_4$ exhibited reduced migration energies and elevated hopping frequencies compared to both hcp- and ccp-type $Li_2ZrCl_6$. Moreover, the higher carrier concentration factor ($C$) and migration entropy ($\Delta S_m$) of $0.8Li_2A$–$ZrCl_4$ suggest enhanced configurational dynamics and a flatter energy landscape (Supplementary Text 8).

To clarify the structural modifications induced by the divalent anion and the subsequent variations in the conductive properties, we conducted DFT calculations to construct model structures of hcp-$Li_{2+x}ZrCl_{6−x}O_x$ (Supplementary Fig. 20) and ccp-$Li_{2+x}ZrCl_{6−x}S_x$ (Supplementary Fig. 21). All structures retained the original anionic sublattice, and the elongation of the Zr-Cl bond length, demonstrated through EXAFS in Fig. 2c and Fig. 3c, was verified using the simulated radial distribution function (RDF), confirming the consistency between the modeled structures and the experimentally observed configurations (Supplementary Fig. 22,23). Details of the calculations are provided in the *Theoretical calculations* part of the "Method" section. With

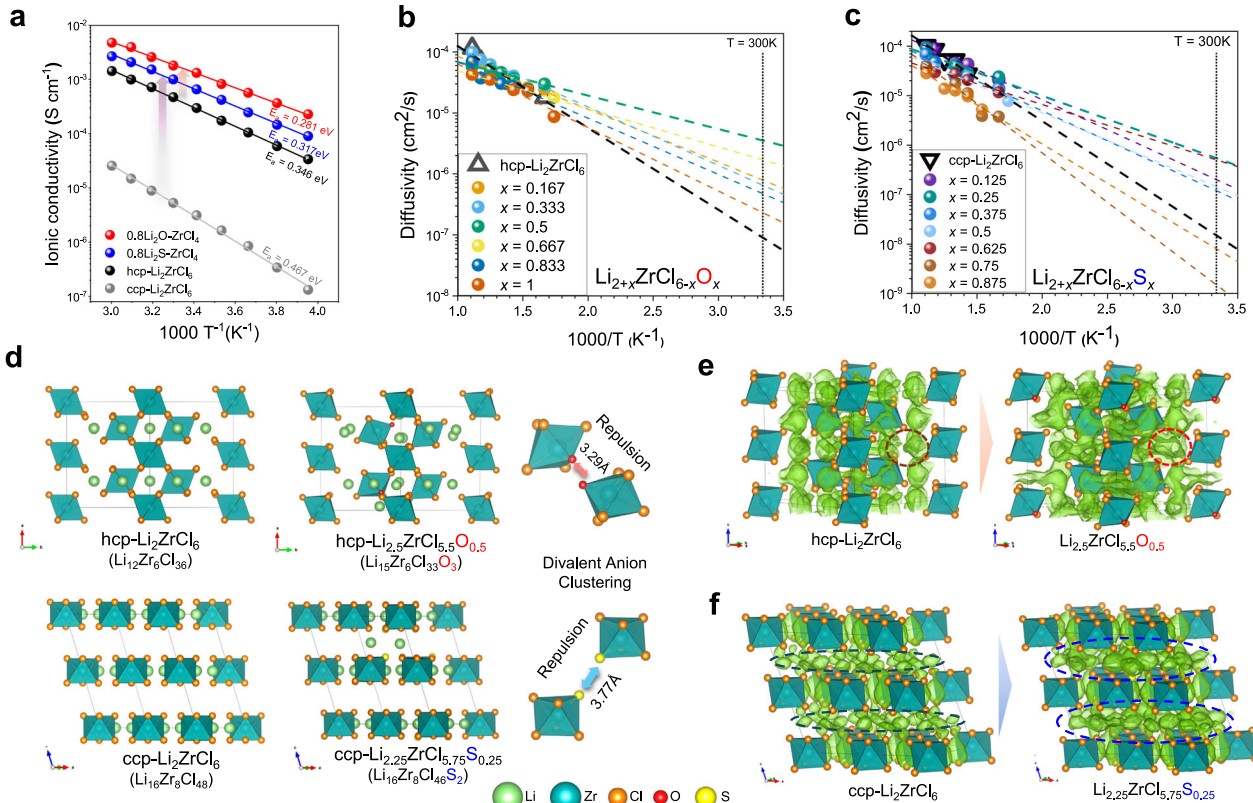

**Fig. 4 | Ionic-conduction properties of divalent-anion-substituted lattice.**
**a** Arrhenius plots of ionic conductivities of $0.8Li_2A–ZrCl_4$ (A = O, S) with hcp-
$Li_2ZrCl_6$ and ccp-$Li_2ZrCl_6$. **b, c** Arrhenius plots of the AIMD simulation of hcp-
$Li_{2+x}ZrCl_{6-x}O_x$ (x = 0, 0.166, 0.333, 0.5, 0.667, 0.833, 1) (**b**) and ccp-$Li_{2+x}ZrCl_{6-x}S_x$
(x = 0, 0.125, 0.25, 0.375, 0.5, 0.625, 0.75, 0.875, 1) (**c**) with extrapolated linier fit line.
**d** Crystal structure of hcp-$Li_{2.5}ZrCl_{5.5}O_{0.5}$ with hcp-$Li_2ZrCl_6$ and ccp-
$Li_{2.25}ZrCl_{5.75}S_{0.25}$ with ccp-$Li_2ZrCl_6$. **e, f** Li probability density at 600 K in ~300 ps
(isosurface value P = $P_{max}$/100) of hcp-$Li_{2.5}ZrCl_{5.5}O_{0.5}$ with hcp-$Li_2ZrCl_6$ (**e**) and ccp-
$Li_{2.25}ZrCl_{5.75}S_{0.25}$ with ccp-$Li_2ZrCl_6$ (**f**).

an increasing ratio of divalent anions, the lattice volume first increases due to divalent-anion-driven lattice expansion and then decreases later due to contraction by additional lithium cation (Supplementary Table 21,22). To reveal the inherent conduction mechanisms, ab initio molecular dynamics simulations (AIMD) were conducted. Figure 4b and Fig. 4c present Arrhenius plots of the AIMD simulations for hcp-$Li_{2+x}ZrCl_{6-x}O_x$ and ccp-$Li_{2+x}ZrCl_{6-x}S_x$, respectively, along with extrapolated linear fits. The ionic diffusion in $Li_{2.5}ZrCl_{5.5}O_{0.5}$ (LZCO) is ~39 times faster than that in hcp-LZC ($3.56 \times 10^{-6}$ cm$^2$ s$^{-1}$ vs. $9.21 \times 10^{-8}$ cm$^2$ s$^{-1}$ at 300 K), and that of $Li_{2.25}ZrCl_{5.75}S_{0.25}$ (LZCS) is 36 times faster than that in ccp-LZC ($5.49 \times 10^{-7}$ cm$^2$ s$^{-1}$ vs. $1.52 \times 10^{-8}$ cm$^2$ s$^{-1}$ at 300 K).

To identify the structural impact on the diffusion, it is necessary to understand the differences driven by divalent anionic effects in the lattice. The model structures of $Li_{2.5}ZrCl_{5.5}O_{0.5}$ with hcp-LZC and $Li_{2.25}ZrCl_{5.75}S_{0.25}$ with ccp-LZC are presented in Fig. 4d, and there are common features between the divalent-anion-induced structures. First, divalent anions are in neighboring sites because of the effective charge compensation for additional Li cations. The structural stability is enhanced when additional cations and divalent anions form local clusters compared to when dispersed anions conduct charge compensation separately (Supplementary Fig. 24). Second, the structures show lattice expansion with distortion, as shown in Supplementary Table 21,22, which are driven by relatively intensive coulombic repulsion between neighboring divalent anions ($O^{2-}–O^{2-}$ and $S^{2-}–S^{2-}$ vs. $Cl^-–Cl^-$).

Indeed, the probability density at 600 K shows a wider diffusion pathway and larger regions of Li probability density in the divalent anion substituted structure (Fig. 4e,f). A more noteworthy observation is that sulfur substitution leads to enhanced lithium-ion conduction along the ab plane, corresponding to the Li layers, compared to ccp-

LZC (Fig. 4f, **dashed line**). In ccp framework, the lattice spacing of the ab plane primarily influences the activation of lithium-ion conduction[28,40]. Given that the ccp-LZC exhibits limited Li$^+$ transport along this plane, the lattice expansion and increased Li concentration induced by S$^{2-}$ substitution allow the structure to reach a critical threshold necessary for enabling conduction in this previously suppressed direction (Supplementary Fig. 25). This highlights a key phenomenological role of sulfur substitution in facilitating multi-dimensional conduction pathways, which is experimentally supported by over two orders of magnitude increase in Li$^+$ conductivity.

We performed topological analysis to investigate the channel size of the lithium-ion migration pathways due to lattice changes and observed that both divalent anions enlarged the lithium-transport channel size in each hcp and ccp structure (Fig. 5a and Supplementary Fig. 26). To further understand the structural origin of this enhancement, we performed continuous symmetry measure (CSM) analysis based on the principle that lithium-site distortion raises site energy and reduces the energy gap to the transition state, thus promoting ionic conductivity (Fig. 5b)[41]. The results reveal that the LZCS shows a modest increase in distortion relative to the original ccp phase, and the LZCO exhibits higher distortion compared to the hcp framework (Supplementary Fig. 27 and Supplementary Table 23). This distortion, induced by divalent anion substitution, also alters the bonding environment in the lattice. Critically, the crystal orbital Hamilton population (COHP) analysis revealed weaker bonding of Li–Cl when divalent anions are introduced to the lattice because of the distant Zr–Zr polyhedron distance and the enlarged coordinated Li site of hcp-LZCO (Fig. 5c) and ccp-LZCS (Fig. 5d). Importantly, the bonding energy of lithium with chlorine has wide and diverse integrated COHP (ICOHP)

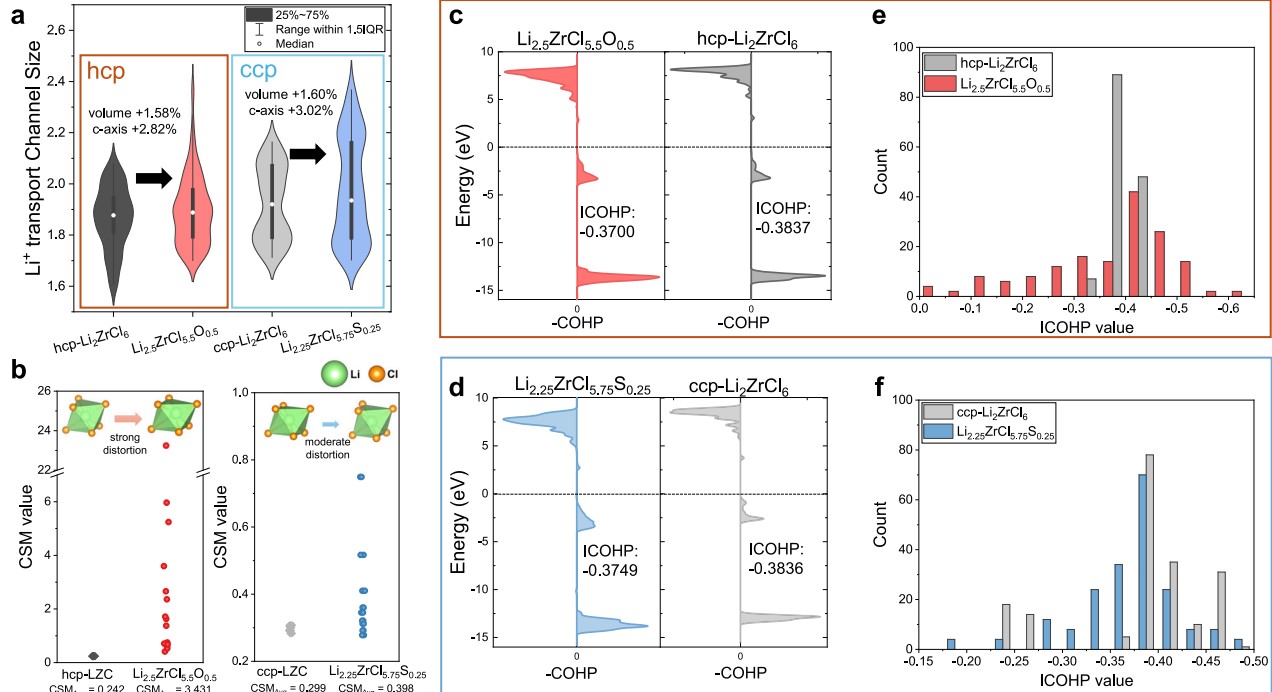

**Fig. 5 | Enhanced conduction mechanism of divalent-anion-substituted structure. a** Topological analysis and Li⁺-transport channel size of hcp-$Li_2ZrCl_6$, hcp-$Li_{2.5}ZrCl_{5.5}O_{0.5}$, ccp-$Li_2ZrCl_6$, and ccp-$Li_{2.25}ZrCl_{5.75}S_{0.25}$. White dots indicate the median, boxes represent the interquartile range (25–75%), and whiskers extend to 1.5× the interquartile range. **b** Continuous symmetry measure (CSM) values of hcp-$Li_2ZrCl_6$, hcp-$Li_{2.5}ZrCl_{5.5}O_{0.5}$, ccp-$Li_2ZrCl_6$ and ccp-$Li_{2.25}ZrCl_{5.75}S_{0.25}$. A minimum of 0 corresponds to a perfectly symmetric coordination environment and the

maximum of 66.7 corresponds to infinite elongation along one direction. **c**, **d** Crystal orbital Hamilton population (COHP) curves with averaged individual COHP (ICOHP) values for Li-Cl bonds (**c**) and histograms for distribution of ICOHP values for Li-Cl bonds (interval width = 0.5) (**d**) in hcp-$Li_{2.5}ZrCl_{5.5}O_{0.5}$ with hcp-$Li_2ZrCl_6$. **e**, **f** COHP curves with averaged ICOHP values for Li-Cl bonds (**c**) and histograms for distribution of ICOHP values for Li-Cl bonds (interval width = 0.25) (**d**) in ccp-$Li_{2.25}ZrCl_{5.75}S_{0.25}$ with ccp-$Li_2ZrCl_6$ (**e**).

distributions, which means that the site energy of each lithium is not discrete but appears to be flattened (Fig. 5e,f).

Comprehensively, the fundamental cause of the enhanced lithium-ion conduction is that the structural variation induces flattening of the energy landscape in site-to-site conduction for Li-ions[42]. This effect stems from the incorporation of clustered divalent anions, which modulate lattice framework by widening conduction channels, weakening Li−Cl interactions, and diversifying the energy states of Li sites. Both divalent anion incorporation strategies retain their conductivity-enhancing effects despite of intrinsic cationic disorder (Supplementary Fig. 28-32 and Supplementary Table 24). Interestingly, the lattice volume expansion, which underlies the design principles of sulfide SEs, and the Li site distortion, which is central to ionic conduction enhancement in oxide SEs, can both be achieved through the incorporation of divalent anions in monovalent halide lattices[31,43]. However, though based on this fundamental principle, the dominant mechanisms affecting the structure differ depending on the divalent anion type. Oxygen in hcp lattice causes strong local distortions due to its size mismatch with Cl⁻, which significantly alters Li-site coordination and facilitates migration. Sulfur incorporation in ccp lattice follows the same mechanism but is particularly effective in activating transport along the ab-plane in the ccp-LZC structure. This is achieved by expanding interlayer spacing and slightly increasing Li content, which together enable conduction in an otherwise inactive dimension. A comparative summary of these mechanisms and their associated structural, morphological, and electrochemical effects is presented in Fig. 6 and Supplementary Table 25.

**Electrochemical properties of 0.8Li₂A−ZrCl₄ (A = O, S) catholyte**
Cyclic voltammetry (CV) curves from 3 to 5 V (vs. Li/Li⁺) of $0.8Li_2A−ZrCl_4$ (A = O, S) are presented in Fig. 6a. The $0.8Li_2O−ZrCl_4$

exhibits good anodic stability ( ~ 4.2 V vs. Li/Li⁺) comparable to that of $Li_2ZrCl_6$ (Supplementary Fig. 33) and other chloride SEs[44,45], whereas the $0.8Li_2S−ZrCl_4$ shows a lower oxidation limit (~3.4 V vs Li/Li⁺). This lower limit is due to the incorporation of sulfur atoms, which are divalent and less electronegative (electronegativity, E.N. = 2.5) compared to Cl (E.N. = 3.0) or O (E.N. = 3.5), within the framework[46,47]. However, for both materials, there were no distinct reduction peaks observed during the reverse scan up to 3 V, and the amount of oxidation decreased in the second scan. Additionally, like $Li_2ZrCl_6$, $0.8Li_2O−ZrCl_4$ and $0.8Li_2S−ZrCl_4$ exhibited a reduction limit near 2 V (vs. Li/Li⁺) due to the same central metal (Zr) element, as confirmed by linear sweep voltammetry (LSV) and the intrinsic stability window (Supplementary Fig. 34-36). Furthermore, the electronic conductivity of $Li_2ZrCl_6$, $0.8Li_2O−ZrCl_4$, and $0.8Li_2S−ZrCl_4$ was confirmed to be as low as ~$10^{-10}$ S cm⁻¹ (Supplementary Fig. 37 and Supplementary Table 26).

To investigate the electrochemical performance when $0.8Li_2A−ZrCl_4$ (A = O, S) are used as a catholyte, solid-state batteries were fabricated (Supplementary Fig. 38). The single-crystalline $LiNi_{0.6}Mn_{0.2}Co_{0.2}O_2$ (s-NCM622) composites with $0.8Li_2O−ZrCl_4$ and $0.8Li_2S−ZrCl_4$ exhibit high initial discharge capacities of 179.6 mAh g⁻¹ and 176.8 mAh g⁻¹ at 0.2 C (1 C = 180 mA g⁻¹) in the potential range of 2.8 V–4.3 V versus Li/Li⁺ at 25 °C and initial coulombic efficiencies of 92.8% and 89.4%, respectively (Fig. 7b). The cycling tests over 100 cycles at 36 mA g⁻¹ (0.2 C) (Fig. 7c) demonstrate that capacity retention of $0.8Li_2O−ZrCl_4$ (140.5 mAh g⁻¹ after 100 cycles) exhibits greater cycling stability compared to that of $0.8Li_2S−ZrCl_4$ (93.4 mAh g⁻¹ after 100 cycles). This difference can be attributed to the side reactions induced by the sulfur anion in $0.8Li_2S−ZrCl_4$, which leads to inferior cyclability. However, it is inferred that the formation of a passivation layer within the positive electrode composite occurs partially[48,49],

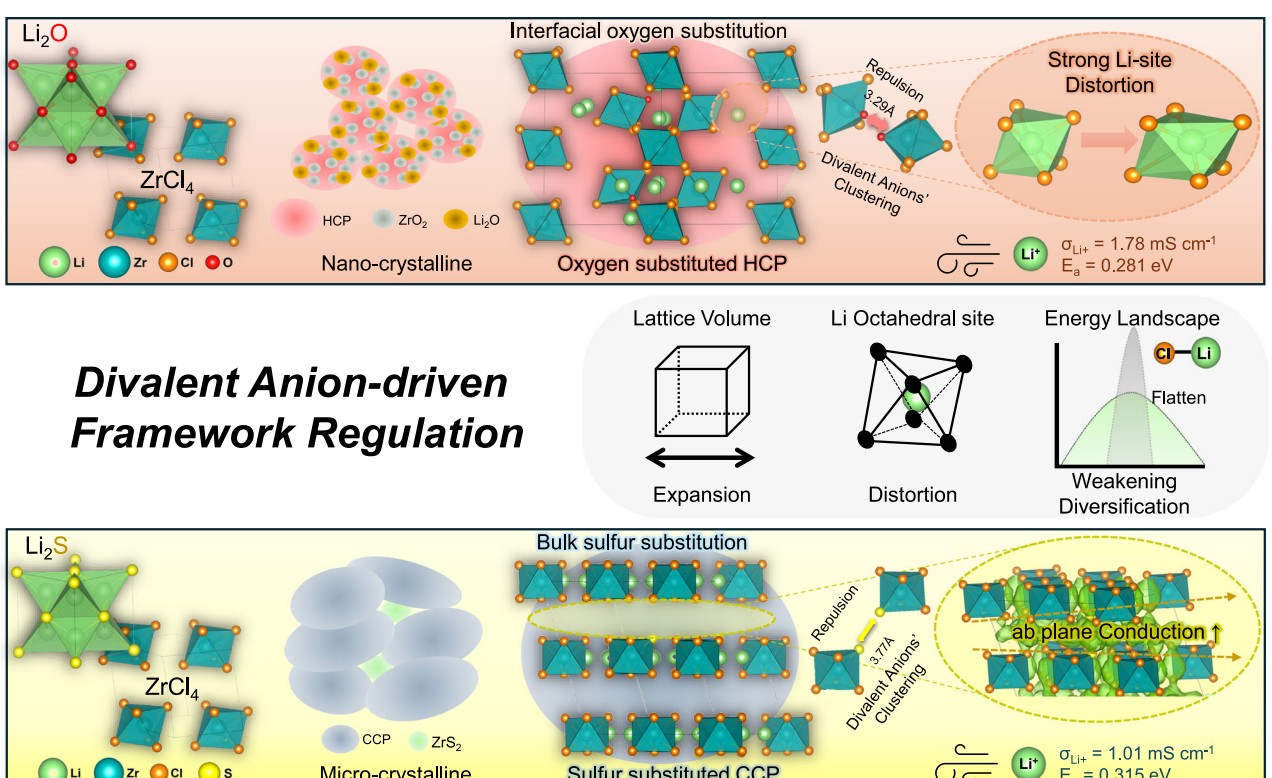

**Fig. 6 | Divalent anion-driven framework regulations.** Schematic illustration of divalent anion-driven framework regulation.

demonstrating cycling performance comparable to that of $Li_2ZrCl_6$ (Supplementary Fig. 39,40). Furthermore, $0.8Li_2O–ZrCl_4$ (Fig. 7d) outperforms $0.8Li_2S–ZrCl_4$ and $Li_2ZrCl_6$ (Supplementary Fig. 41) in rate performance due to its relatively higher ionic conductivity and shows the prominent cycling performance at 90 mA g$^{-1}$ (0.5 C) (Fig. 7e). Comprehensively, in terms of charge–discharge capability, divalent-anion-regulated frameworks act as a basis for improved electrochemical performance compared to LZC because of the structural regulation driven ionic-conduction enhancement.

## Discussion

We demonstrated that divalent anions play a crucial role in regulating the framework of Zr-based SEs, facilitating the formation of superionic conductive lattices. Specifically, the oxygen source ($Li_2O$) promotes the nanostructuring of an oxygen-substituted hcp anionic sublattice. In contrast, the sulfur source ($Li_2S$) drives the formation of a sulfur-substituted ccp anionic sublattice. Divalent anions introduced into the hcp and ccp lattice tend to form clusters, which induce lattice expansion and consequently enlarge the available topological space for lithium conduction. Moreover, Li site distortion, triggered by divalent-anion incorporation, results in weakened and diversified Li–Cl bonding environments, which contribute to the flattening of the energy landscape for ion migration. Oxygen and sulfur both contribute to enhanced ionic conduction through a common underlying mechanism involving lattice modulation induced by divalent anion. Oxygen primarily inducing severe distortion of Li sites and sulfur more effectively promoting ab-plane conduction. Therefore, in optimized compositions of $0.8Li_2A–ZrCl_4$ (A = O and S), improved Li$^+$ conduction compared with $Li_2ZrCl_6$ results in ionic conductivities of 1.78 and 1.01 mS cm$^{-1}$ at 25 °C in $0.8Li_2O–ZrCl_4$ and $0.8Li_2S–ZrCl_4$, respectively. Irrespective of the specific anion lattice type (hcp or ccp), framework regulation through the incorporation of divalent anions serves as a universal strategy to flatten the energy landscape and enhance ionic conduction. This modulation became possible through the incorporation of oxygen

and sulfur into the chloride anion sublattice while maintaining the original lattice structure. In this mechanism, local hetero-anion clustering, induced by the aliovalent incorporation of higher-valent anions into the lattice, modulates the conduction properties. The advanced electrochemical properties of the optimized SEs demonstrate improved cycling stability and rate performance in ASSBs. These results establish a generalized design principle for enhancing ionic conductivity in halide SEs through anion-induced lattice regulation, with the potential to enable the development of practical, high-performance ASSBs.

## Methods

### Preparation of materials

A stoichiometric mixture of $Li_2O$ (99.5%, Alfa Aesar), $Li_2S$ (99.9%, Alfa Aesar), LiCl (99.995%, Alfa Aesar) with $ZrCl_4$ (99.5%, Alfa Aesar) was ball-milled using a Pulverisette 7PL (Fritsch GmbH). Stoichiometrically weighed 1 g batch powder mixtures at a weight ratio of 1/20 ratio with 5-mm $Si_3N_4$ balls were pre-mixed at 150 rpm for 30 min and instantly ball-milled at 600 rpm for 24 h in a $Si_3N_4$ pot (45 mL) sealed in an Ar atmosphere. For ccp-$Li_2ZrCl_6$, the ball-milled powders were annealed at 300 °C for 5 h under an Ar atmosphere. Single-crystalline $LiNi_{0.6}Mn_{0.2}Co_{0.2}O_2$ (s-NCM622, BASF, $D_{50}$ ~ 4 μm) and $Li_6PS_5Cl$ (POSCO JK Solid Solution (5 μm, > 3 mS cm$^{-1}$) were used for fabricating ASSBs.

### Material characterization

Lab-source powder XRD patterns were collected using a Rigaku Ultima IV diffractometer (scan rate of ~1.2 °/min and step size of 0.01°) with Cu $K_\alpha$ radiation ($\lambda = 1.5406$ Å). High-energy X-ray total scattering measurements were performed at the 28-ID-1 (PDF) beamline of the National Synchrotron Light Source II (Brookhaven National Laboratory) using an incident beam energy of 74.5 keV, corresponding to a wavelength of 0.1665 Å. Powder samples were placed in polyimide (Kapton) capillaries and sealed with epoxy to prevent air contact

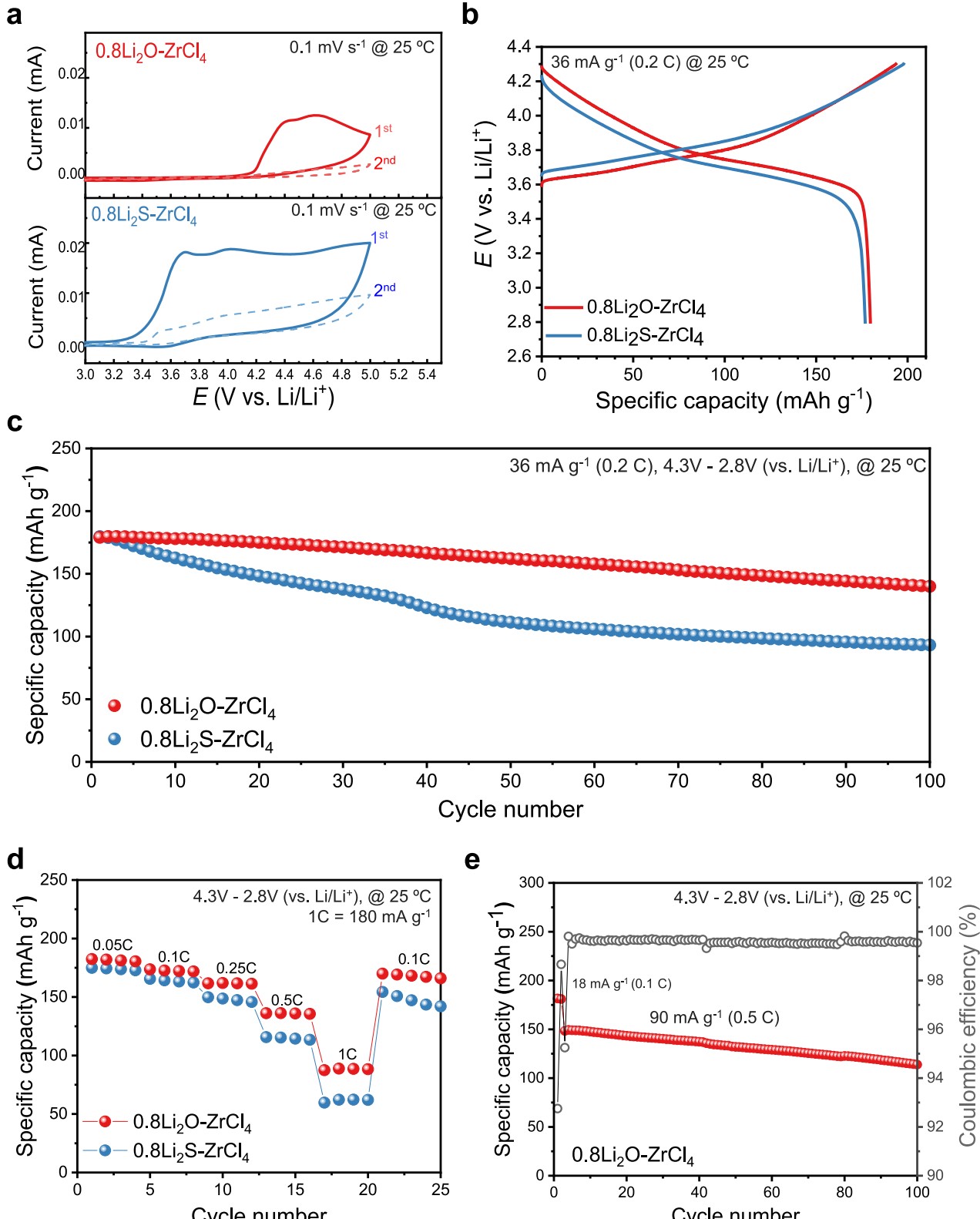

**Fig. 7 | Electrochemical performance of ASSBs with 0.8Li₂A–ZrCl₄ (A = O, S).**
**a** CV curves for 0.8Li₂A–ZrCl₄ (A = O, S) in (Li–In)|LPSC | SE | (SE–carbon) cells from 3.0 to 5.0 V (vs. Li/Li⁺) at 0.1 mV s⁻¹ and 25 °C. The weight ratio of SE:carbon is 7:3. **b**, First-cycle charge–discharge voltage profiles for s-NCM622 electrode with 0.8Li₂A–ZrCl₄ (A = O, S) (4.3–2.8 V vs. Li/Li⁺) at 36 mA g⁻¹ (0.2 C) and 25 °C. **c**, **d** Cycling performance at 36 mA g⁻¹ (0.2 C) (**c**) and rate performance at different C-rates (1 C = 180 mA g⁻¹) (**d**) with 0.8Li₂A–ZrCl₄ (A = O, S) at 25 °C. **e** Cycling performance at 0.5 C with 0.8Li₂O–ZrCl₄ at 25 °C.

during measurements. Two-dimensional diffraction images were integrated into one-dimensional profiles using Dioptas[50]. The pair distribution function, G(r), was obtained by Fourier transforming the total scattering intensity over a Q range of $1.5-23\,Å^{-1}$ using the xPDFsuite program[51]. Zr K-edge X-ray absorption spectra were collected at beamlines 7D, 8C, and 10C of the Pohang Accelerator Laboratory (PAL), employing a Si (111) double-crystal monochromator operated in both transmission and fluorescence mode. Energy calibration was referenced to a Zr metal foil, and subsequent XANES and EXAFS analyses were carried out with the Demeter software package[51]. The spectra were processed and normalized using *ATHENA* by removing a linear pre-edge background (from $-200$ eV to $-60$ eV) and scaling the post-edge region (from 100 eV to 750 eV) so that the absorption edge step at $E_0$, defined as the first maximum in the derivative spectrum, was unity. The extracted EXAFS signal, $k3\chi(k)$, was Fourier transformed over a $k$-range of $3.1-12.8\,Å^{-1}$ for $xLi_2O-ZrCl_4$ and $3.8-11.2\,Å^{-1}$ for $xLi_2S-ZrCl_4$ to obtain magnitude plots of the EXAFS spectra. EXAFS are fitted over a $r$-range of $1.35-3.5$ Å for $xLi_2O-ZrCl_4$ and $1.7-3.7$ Å for $xLi_2S-ZrCl_4$. Detailed fitting procedures for PDF and EXAFS are shown in Supplementary Text 2,3. Raman spectra were collected using a ARAMIS spectrometer (Horiba Jobin Yvon) equipped with an Ar-ion laser operating at an excitation wavelength of 785 nm. For the HRTEM measurements, the samples were loaded onto a lacey Cu grid in an Ar-filled glove box, and HRTEM images with FFT patterns were obtained using a JEM-ARM 300 F NEOARM (JEOL). Imaging was conducted at a reduced accelerating voltage of 160 keV, rather than the standard 300 keV, to mitigate degradation during measurement, and beam exposure was further minimized by performing search and focus procedures on separate regions and limiting the area of interest to a single-frame acquisition without additional alignment or prolonged exposure. $^7Li$ MAS NMR spectroscopy was conducted using a Varian VNMRS 600 (solid system) at a 14.1 T magnetic field (600 MHz for $^1H$, 233 MHz for $^7Li$) and chemical shifts were referenced to LiCl. XPS measurements were performed with a monochromatic Al $K_\alpha$ source (1486.6 eV) at 12 kV and 6 mA using K-Alpha$^+$ (Thermo Fisher Scientific), with a pass energy of 20 eV and a beam spot size of 400 μm. The binding energies were calibrated against the adventitious C 1 s peak at 284.8 eV. Samples were mounted on a NMR holder and XPS vacuum transfer holder in the Ar-filled glove box and transferred into equipment without exposure to air. The particle size was observed using a Hitachi SU8220 Cold FE-SEM operated at 7 kV, with the samples vacuum-packed to minimize air exposure during transfer.

## Theoretical calculations

Density functional theory (DFT) calculations were performed using the Vienna Ab initio Simulation Package (VASP)[52]. The exchange–correlation effects were treated within the generalized gradient approximation using the Perdew–Burke–Ernzerhof (PBE) functional, and the projector-augmented wave (PAW) approach was employed[53]. A plane-wave cutoff energy of 520 eV was applied, and full structural optimizations were carried out until all atomic forces were less than $0.05\,eV\,Å^{-1}$, allowing relaxation of both cell parameters and atomic coordinates. Possible atomic configurations of hcp $Li_{2+x}ZrCl_{6-x}O_x$ and ccp $Li_{2+x}ZrCl_{6-x}S_x$ were systematically generated through an enumeration approach that sampled various occupancies of the partially filled atomic sites. The enumeration process was carried out using the *pymatgen* library[54] to efficiently construct all symmetry-distinct configurations. Potential lithium interstitial positions were identified in advance with the *Topography Analyzer*[55] module in *pymatgen*, which screens geometrically accessible sites for Li insertion. For anionic substitution, specific sites were chosen according to the target substitution ratios. Among the generated models, the thirty configurations with the lowest electrostatic (Ewald) energies were selected as initial candidates. These structures were

then fully optimized using consistent DFT parameters, and the configuration yielding the minimum total energy was adopted for subsequent analyses.

Ab initio molecular dynamics (AIMD) simulations were employed to investigate lithium-ion diffusion and migration behavior. Calculations were carried out in the NVT ensemble using a Nosé–Hoover thermostat with an oscillation period of 80 fs[56]. The simulation models were constructed as 1×1×2 supercells of $Li_2ZrCl_6$ and its derivative structures, each exceeding 10 Å along all lattice vectors. A Γ-centered 1×1×1 k-point mesh was used for Brillouin zone sampling. Each system was equilibrated by gradually heating from 100 K to the target temperature (600–900 K) over 2 ps, followed by production runs of 300 ps with a 2 fs time step. Lithium diffusivities ($D_{Li+}$) were determined from the linear slope of the mean square displacement (MSD) of $Li^+$ ions according to standard diffusion relations:

$$MSD = \frac{1}{N}\sum_{i=1}^{N}\left|r_i(t+\Delta t) - r_i(t)\right|^2 \quad (1)$$

$$D = \frac{1}{2dtN}\sum_{i=1}^{N}\left|r_i(t+\Delta t) - r_i(t)\right|^2. \quad (2)$$

In these expressions, $r_i$ represents the position of the $i^{th}$ Li-ion at time t, $\Delta t$ denotes the simulation time interval, N is the total number of Li ions in the simulation cell, and d corresponds to the dimensionality factor. The mean square displacement (MSD) profiles and diffusion coefficients were evaluated using the diffusion analysis module implemented in the *pymatgen* library[54]. The Li-ion migration pathways derived from AIMD trajectories were further examined through topological channel analysis using the Zeo$^{++}$ package[57]. Structural and energetic data for all reported compounds in the Li–Zr–Cl–O(S) chemical space were retrieved from the *Materials Project* database[58], and their total energies were recalculated under the same DFT settings to determine the energy above the convex hull and interfacial reaction energies. By constructing the grand potential phase diagrams as a function of the Li chemical potential ($\mu_{Li}$), we determined the intrinsic stability window for $Li_2ZrCl_6$, $Li_{2.5}ZrCl_{5.5}O_{0.5}$ and $Li_{2.25}ZrCl_{5.75}OS_{0.25}$.

$$\Delta E_D = E_{eq}(Phase\ equilibria, \mu_{Li}) - E_{material}(Phase) - \Delta n_{Li}\mu_{Li} \quad (3)$$

where $\mu_{Li}$ is the chemical potential of lithium and $\Delta n_{Li}$ is the number difference of element Li from the original composition.

The LOBSTER code was used in conjunction with VASP outputs to perform COHP calculations for analysis of the Li and Cl bonding properties[59]. The COHP was plotted over an energy range of $-16$ eV to 10 eV. To focus on the nearest Li-Cl interactions, a cutoff distance of 3.6 A was applied, and contributions from both up-spin and down-spin states were included. The integrated COHP (ICOHP) values for Li-Cl were calculated by integrating the bonding states from the lowest energy to the Fermi level. The ICOHP values of all Li-Cl interactions within the defined range were averaged to represent the overall characteristics of Li-Cl bonds. The optimized computational structures and the atomic configurations in AIMD simulations used in this study are available as Supplementary Data 1–23.

## Electrochemical characterization

The ionic conductivities were measured using the AC impedance method with ion-blocking SUS | SE | SUS symmetric cells. Pellets with a diameter of 13 mm were fabricated by cold pressing under a uniaxial pressure of 370 MPa for 2 min. Electrochemical impedance spectra were collected at open-circuit voltage using a VSP-300 (Bio-Logic) analyzer, with a 14.2 mV AC perturbation over a frequency range of 7 MHz to 10 mHz. Each decade in frequency contained ten sampling points. The ionic conductivities were determined from four

independent measurements to verify reproducibility, and the resulting deviations are shown as error bars in the corresponding figures. For the all-solid-state cells (13 Ø), the Li–In alloy was fabricated by uniaxial pressing (30 MPa) of Li foils (0.02 mm, Wellcos, 0.53 g/cm³) and In foils (0.05 mm, Wellcos, 7.31 g/cm³), and it was used as the counter and reference electrodes. We employed a Li–In alloy electrode prepared from Li and In foils (nominal $Li_{0.5}In$ composition). This composition lies in the In/InLi two-phase region, which shows a characteristic plateau at ~0.62 V vs. Li/Li[+31]. Therefore, the measured Li–In/Li⁺ potentials were shifted by ~0.62 V to report the potentials vs. Li/Li⁺. The Li–In and $Li_6PS_5Cl$ powders (80 mg, ~300 µm) were pelletized under 50 MPa for 1 min, and the SE powders ($Li_2ZrCl_6$, $0.8Li_2O–ZrCl_4$ and $0.8Li_2S–ZrCl_4$, 60 mg, ~200 µm) were poured onto the pellets and pressed under 100 MPa. Then, the powders for the working electrode were placed on top, and the entire assembly was pressed at 370 MPa for 2 min. For the LSV and CV measurements, SE powders were manually mixed with carbon (carbon nanofibers, VGCF, Aldrich) in a weight ratio of 7:3. The SE–carbon mixture (10 mg) was used for the working electrode. For the all-solid-state (Li–In)|LPSC|SE|s-NCM622 cells, s-NCM622, SE, and VGCF were mixed with a mortar and pestle in a weight ratio of 75:22.5:2.5 for 30 min and 15 mg was used for the working electrode (mass loading ≈ 11.3 mg cm⁻²). Then, a constant pressure of 50 kgf·cm torque was applied in the uniaxial direction (stack pressure ≈ 35 MPa). The voltammetry measurements were conducted using a VMP-300 (Bio-Logic) at a scan rate of 0.1 mV s⁻¹, and charge–discharge measurements were carried out using a WBCS3000 (Wellcos). All cells were assembled using a stainless-steel (SUS304) die set equipped with upper and lower plungers, with PEEK insulation applied to the side walls to prevent electrical shorting during compression (13 mm diameter active area), and a uniaxial pressure of ~35 MPa was maintained during electrochemical measurements. The electrochemical performance data correspond to representative cells, while the results from all tested cells (n ≥ 3) showed consistent trends. All fabrication processes were performed in an Ar-filled glove box, and electrochemical energy storage tests were conducted in a climate chamber set at 25 ± 1 °C.

## Data availability
All data generated and analyzed in this study are included in the main text and Supplementary Information. Source data are provided with this paper. The structural files used in the simulations are provided as Supplementary Data. Further details are available from the corresponding authors. Source data are provided with this paper.

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

## Acknowledgements

This research was supported by the Samsung Science and Technology Foundation under project no. SRFC-MA2102-03 (Y.S.J.), and by the National Research Foundation of Korea (NRF), funded by the Ministry of Science, ICT (2021M3H4A1A04093050, RS-2024-00435493, GTL24011-000 for D.-H.S.) and (No. GTL24012-000 for S.-K.J.). The PDF research used beamline 28-ID-1(PDF) of the National Synchrotron Light Source II, a US Department of Energy (DOE) Office of Science User Facility operated for the DOE Office of Science by Brookhaven National Laboratory under contract no. DE-SC0012704 (K.-W.N.). The computational work was supported by the Supercomputing Center/Korea Institute of Science and Technology Information with supercomputing resources, including technical support (KSC-2024-CRE-0243 for S.-K.J.).

## Author contributions

J.-S.K and D.-H.S. conceived the concept and designed the experiments, with D.-H.S. supervising the project. With advice from Y. S.J. and S.-K.J., J.-S.K. conducted synthesis, material characterization, and electrochemical analysis. D.H., H.-Y.K., S.L. and K.-W.N. conducted the synchrotron X-ray characterization. J.-S.K. and J.C. performed the theoretical calculations with support from Y.-Y.S. and C.-D.L. Experimental assistance was provided by Y.K., J.S., S.-H.H., J.L., H.K. and J.K. Thereafter, J.-S.K., Y. S.J., S.-K. J., K.-W. N. and D.-H.S. wrote and revised the manuscript. All the authors reviewed and provided feedback on the manuscript.

## Competing interests

The authors declare no competing interests.
