## [Peer Review File · Nature Communications]

Divalent Anion-Driven Framework Regulation in Zr-Based Halide Solid Electrolytes for All-Solid-State Batteries

Corresponding Author: Professor Dong-Hwa Seo

Version 0:

Reviewer comments:

Reviewer #1

(Remarks to the Author)

Kim and collaborators propose a new class of anion-enriched oxy-halide and sulfur-halide solid electrolytes with superior ionic conductivities compared to their halide analogues. They claimed that ion-conductivity is increased upon mixing oxygen or sulfur in Li_2ZrCl_6 , as divalent anions reduce the extent of local Coulombic electrostatic interactions between Li and Cl. This, in turn, flattens the potential energy surface of the materials and reduces Li-ion migration barriers. These conclusions are achieved by combining several characterization techniques, including bench and synchrotron X-ray diffraction, scattering techniques PDF, HR transmission electron microscopy, and electrochemical impedance spectroscopy. Subsequently, they used a specific Oxygen/sulfur-rich composition to realize all-solid-state battery devices. In general, this manuscript targets a critical problem and should be considered by this journal after my comments are addressed:

1. The current title is too general because the manuscript only deals with two specific Zr-based halide electrolytes. The revised title should be narrowed to indicate the precise class of materials explored in this investigation.
2. In the abstract, "oxidation stability" should be replaced by "oxidative stability".
3. The X-ray diffraction analysis in Fig. 1a-c and Fig. 1d-f appears somewhat superficial. They observed the formation of Li_2O and LiCl at specific x compositions. I cannot find whether they did any Rietveld and estimate the phase fractions of Li_2O and LiCl from the diffractograms in these figures. Likewise, they should report the phase fraction of unreacted/residual Li_2ZrCl_6 . The absence of this analysis diminishes the importance of their findings and makes their claims very speculative.
4. I'm surprised that ZrO_2 is only detected in the PDF analysis and doesn't appear in the XRD diffractogram. Have they tried to fingerprint ZrO_2 in the Synchrotron diffractogram? Is this monoclinic or cubic ZrO_2 ? As per comment #3, these authors should attempt to quantify the amount of ZrO_2 in the sample.
5. I was wondering if Raman would be a better technique for describing the amorphous portion of these anion-enriched materials.
6. The lack of determination of the phase fraction Li_2O , LiCl , ZrO_2 , and residual/unreacted Li_2ZrCl_6 partially invalidates the interpretation of the impedance data. In particular, they don't analyze the grain boundaries of these materials, and whether these impurities (or unreacted) phases segregate at the grain boundaries, thus affecting the impedance responses.
7. I don't know what they perform HR-TEM on these samples which are known to be beam sensitive.

Reviewer #2

(Remarks to the Author)

This manuscript reported the improvement of the ionic transport properties of halide based solid electrolytes by incorporating divalent anions such as oxygen and sulfur. Authors have used comprehensive characterization and computational techniques to understand the underlying mechanism for the improved transport property. The key takeaway from the work is that, while both oxygen and sulfur can help improve the ionic conductivity of halides, they work in a very different mechanism: oxygen promotes the formation of nanocomposite and sulfur drives the formation of cubic closed packed anion

framework. The findings are quite novel, and the insights can help develop new halide based solid electrolytes. Below are some comments that need to be addressed before I can fully recommend its publication.

1. More clarifications are needed to explain the ionic transport mechanism of oxygen substituted halides. Authors stated that the substitution optimizes the interfacial oxygen incorporation which increased the ionic conductivity. The formation of space charge layer has been used as a way to improve the interfacial ionic conductivity of solid electrolytes. What is the difference between the current mechanism and enhancement of interfacial ionic transport by space charge layer? Many halide electrolytes, if not all, have exhibited improved ionic conductivity with the decrease in crystallinity due to the local order/disorder of cations in the lattice. Refs. 32 and 33 shows that amorphous Li-Zr-Cl-O has a very high ionic conductivity. Is that possible the increase in ionic conductivity is due to the formation of amorphous Li-Zr-Cl-O phase, rather than the stated mechanism? What is the exact role of ZrO₂ or Li₂O?
2. Body centered anion framework has been proposed by Ceder et al, as a basic structural unit for superionic sulfide based solid electrolytes. Authors mentioned sulfur-incorporated halide also benefit from the formation of cubic closed packed anion framework. There seems to be some consistency here and warrants more discussion comparing the ionic transport mechanism between sulfides and halides.
- 3, The computational study utilized a model structure of Li_{2.5}ZrCl_{5.5}O_{0.5}. with hcp-LZC. However, if the ionic transport mechanism is based on the formation of nanocomposite? How can this computational study, based on single-crystalline Li_{2.5}ZrCl_{5.5}O_{0.5}, help reveal the mechanism?
- 4, Electrochemical stability. Authors stated that all the halide electrolytes are not stable below 2.0 V due to the Zr reduction. The results are also shown in Figure S18. More details are needed here to indicate how the CV tests were performed. If the cell was polarized to a higher voltage first, then the reduction peak can be from the oxidized product during the CV test, instead of the intrinsic anodic stability of solid electrolytes. Authors are suggested to test the linear sweep of two cells: one to high voltages, and the other to low voltages to understand the electrochemical stability of the electrolytes.

Reviewer #3

(Remarks to the Author)

The authors investigate how divalent anions (O/Cl and S/Cl) modulate the anion sublattice and associated superionic conductivity within Zr-based halide solid electrolytes, aiming to resolve ambiguities concerning their structure and diffusion mechanisms. Given the growing body of research in oxychloride solid electrolytes, the topic of this manuscript is timely and significant. However, the current manuscript lacks detailed analysis of the structural characterization results, and its arguments regarding the diffusion mechanism require further clarification, particularly how they differ from findings in previous reports on zirconium-based oxyhalides (Nat Commun 14, 2459 (2023), Nat Commun 14, 3807 (2023), and J. Am. Chem. Soc. 2024, 146, 5, 2977–2985). A major revision is required before considering it for publication in Nature Communications.

1. The discussion of the synchrotron XRD, PDF, and EXAFS data is primarily qualitative. Quantitative analyses, such as results from XRD refinement and PDF/EXAFS fitting, are essential for substantiating the structural claims and are currently missing from the manuscript.
2. The authors claim that “the Li₂O promotes the nanostructuring of an oxygen-substituted hcp anionic sublattice, generating a nanocomposite state with oxide compounds (Li₂O and ZrO₂). In contrast, the sulfur source (Li₂S) drives the formation of a sulfur-substituted ccp anionic sublattice.” This argument is not fully persuasive due to the absence of quantitative structural analysis (as noted in comments 1). What is the ratio of Li₂O and ZrO₂ in the structure? How many O and S are substituted into the hcp or ccp lattice?
3. It appears that O-substitution might favor the formation of nanoscopic oxide domains, which could potentially facilitate interfacial Li-ion diffusion. Could this explain why the Li₂O-ZrCl₄ derived material exhibits higher ionic conductivity than the Li₂S-ZrCl₄ derived material? The reviewer suggested the author provide more discussion on it.
4. The manuscript presents simulations comparing pristine and substituted samples. The reviewer suggested expanding the discussion to elaborate specifically on the differences between O- and S-substitution effects. For example, the topological analysis reveals an expanded transport channel in the S-substituted sample compared to the O-substituted one, but why does the O-substituted sample present higher ionic conductivity?
5. What is the electronic conductivity of O- and S- substituted electrolytes? This may vary with the divalent anion composition and affect battery stability.
6. Standard error bars should be added for the ionic conductivity measurement.
7. For FFT patterns in the HRTEM figures, the scale bar and the diffraction pattern should be indicated.
8. In Figure 5d,e, it is hard to read the x-axis for each column, please set a gap between different groups of columns, and there is a typo in Figure 5a: “Li_{2.25}ZrCl_{5.75}Cl_{0.25}”.

Reviewer #4

(Remarks to the Author)

As one of the promising Li ionic conductors, lithium transition metal halides exhibit notable electrochemical stability, yet their Li ion conductivity is generally lower than that of their competitors, such as LGPS. O substitutions have been observed to be compelling in improving their Li conductivity, which was attributed to the wider Li ion diffusion channels in these oxyhalides compared to that in pure halides. In this manuscript, the authors expanded the anionic substitution from O to S and reported a positive effect on the ionic conduction for both. Detailed synchrotron characterizations and first-principles simulations were conducted to elucidate the conduction mechanism. The work has proceeded well, and the manuscript is well organized. However, I am not entirely convinced by the significance of the findings of this work as required by premier journals like Nature Communications. One of the major discoveries of this study is the trial of S substitution along with O, which indeed

shows positive effects on electrolyte performance. However, the improvement (1.78 mS/cm for O and 1.01 mS/cm for S) is rather limited, especially compared to the previously reported LiMOCi4 (M=Nb-10.4 mS/cm, Ta-12.4 mS/cm). S substitution seems to be a bad idea for mixed anionic halide electrolyte design, which, to be honest, is not very surprising. Another vital section of this work is clarifying the underlying mechanism of the S/O substituted halide on Li ion conduction, where the authors weren't able to bring any new understandings but reemphasizing the possibility that the incorporated S or O may broaden the Li channel as raised by previous works. The authors claimed that they would put forward design principles for halide electrolytes, yet I received barely any guidance after a thorough reading of their manuscript. I would recommend its publication in alternative journals unless the authors can provide more evidence.

Version 1:

Reviewer comments:

Reviewer #1

(Remarks to the Author)

I have read carefully the report and the changes introduced to the manuscript. These authors have addressed my initial concerns excellently. The manuscript should be considered by Nature Communications.

Reviewer #2

(Remarks to the Author)

The authors have addressed all of my comments. It can be accepted as is.

Reviewer #3

(Remarks to the Author)

The authors have conducted extensive analyses and experiments to address most of the reviewers' concerns. Despite their significant efforts, the structural analysis, especially for Li₂O-ZrCl₄ system, is still not sufficiently clear.

1. The proposed structure of the Li₂O-ZrCl₄ system is not clearly described and is likely to confuse both reviewers and the broader audience of Nature Communications. Specifically, the phrase "an anion-substituted hcp structure at the interface of ZrO₂ (or Li₂O) with hcp-LZC" is ambiguous. Are the authors suggesting that oxygen substitutes only on the outer surface of the hcp-LZC crystallites, or are they referring to a heterogeneous interface between ZrO₂ and hcp-LZC?

2. There appears to be a contradiction between the the experimentally derived structure and the simulated structure. In response to Comment 2, the authors suggest a final product of 2Li₂O + 5ZrO₂ + 10Li₂ZrCl₆ according to the PDF and EXAFS fitting, implying a heterogeneous structure composed of ZrO₂ and hcp-LZC. If this is the case, the reviewer interprets the authors' response as implying that no oxygen was incorporated into the hcp-LZC structure. However, the computational simulations continue to focus solely on O-doped hcp-LZC, which does not align with the proposed experimental outcome.

3. Questionable PDF and EXAFS fitting details: The current treatment of local structure (i.e., as ZrO₂ + hcp-LZC and ZrS₂ + ccp-LZC) in the PDF and EXAFS analyses is not convincing. It is unclear whether the O-doped and S-doped LZC structures obtained from the authors' calculations fail to provide reasonable fits to the PDF data. For EXAFS, further clarification is required regarding the "S02" parameter values: why is it set to 0.6 in Supplementary Table 7 but 1.0 in Supplementary Table 14? Additionally, the reported coordination numbers appear unrealistically high. For example, the total coordination number for 0.8Li₂O-ZrCl₄ is 36, which is physically implausible. As such, the reliability of the current fitting results is highly questionable.

Reviewer #4

(Remarks to the Author)

I believe the authors have solved all the concerns I raised.

Version 2:

Reviewer comments:

Reviewer #3

(Remarks to the Author)

The authors have addressed all of my concerns. The manuscript should be considered for publication in Nature Communications.

RESPONSES TO REVIEWERS' COMMENTS

Dear Reviewers,

Thank you very much for your insightful and constructive comments. We have provided our point-by-point responses below, highlighted in blue. All changes in the revised manuscript are marked in red. Please note that all figures, tables, and texts cited in the responses refer to those in the revised version of the manuscript and supplementary information. References in the main text are cited using numerical superscripts, whereas references necessary for the response are denoted in the format of R1, R2, etc. We sincerely appreciate the reviewers' thoughtful feedback, which has helped us to significantly improve the quality and clarity of our work.

The major revisions (or responses) are outlined as follows:

1. Quantitative phase analysis was conducted using PDF and EXAFS refinements, enabling reliable identification and quantification of key phases (hcp and ccp), including nanosized monoclinic ZrO₂ and ZrS₂ domains that were previously undetectable by XRD due to nano-crystallinity and short-range ordering.
2. Phase evolution was also quantitatively resolved: oxygen substitution predominantly yields interfacial nanocomposite phases of ZrO₂ and hcp-type structures (~1:2 ratio), whereas sulfur substitution leads to homogeneous bulk formation of ZrS₂ and ccp phases (~3:7 ratio). These assignments were supported by PDF/EXAFS fitting, STEM-EELS mapping, and thermodynamic modeling.
3. The structural origin of ionic conductivity was clarified through a combination of low-temperature EIS, hopping parameter analysis, and PDF/EXAFS fitting. These results confirmed that the enhanced conductivity in oxygen and sulfur substituted samples arises from intrinsic lattice modulation, rather than grain boundary effects, amorphization, or secondary phases.
4. Raman spectroscopy further confirmed the preservation of halide framework polyhedral motifs, reinforcing the structural integrity of substituted phases and ruling out significant amorphization.
5. The conduction mechanisms of oxygen substituted hcp and sulfur substituted ccp were systematically differentiated: oxygen incorporation induces strong Li-site distortion, enhancing hopping frequency and flattening the energy landscape, while sulfur substitution expands the ccp lattice along the ab-plane, activating suppressed transport pathways. These findings establish a generalized design strategy by transferring conduction-enhancing principles from sulfide (volume-driven octahedral destabilization) and oxide (local distortion) systems to halide frameworks.
6. Finally, topological and electronic analyses clarified that despite S-substituted systems exhibiting larger transport channels, O-substituted lattices achieve superior conductivity through greater structural distortion, more asymmetric Li coordination (CSM), and weaker, more diversified Li-Cl bonding environments (ICOHP distributions).

Reviewer #1

Kim and collaborators propose a new class of anion-enriched oxy-halide and sulfur-halide solid electrolytes with superior ionic conductivities compared to their halide analogues. They claimed that ion-conductivity is increased upon mixing oxygen or sulfur in Li_2ZrCl_6 , as divalent anions reduce the extent of local Coulombic electrostatic interactions between Li and Cl. This, in turn, flattens the potential energy surface of the materials and reduces Li-ion migration barriers. These conclusions are achieved by combining several characterization techniques, including bench and synchrotron X-ray diffraction, scattering techniques PDF, HR transmission electron microscopy, and electrochemical impedance spectroscopy. Subsequently, they used a specific Oxygen/sulfur-rich composition to realize all-solid-state battery devices. In general, this manuscript targets a critical problem and should be considered by this journal after my comments are addressed:

Response to comment: We are thankful to the reviewer for a thorough review of our manuscript and the constructive comments.

1. The current title is too general because the manuscript only deals with two specific Zr-based halide electrolytes. The revised title should be narrowed to indicate the precise class of materials explored in this investigation.

Response to comment 1: We sincerely appreciate the reviewer's insightful suggestion regarding the manuscript title. In response, we have revised the title to more accurately reflect the specific materials investigated in this study by explicitly including "Zr-based halide solid electrolytes.". We believe this modification provides greater clarity and appropriately articulated the scope of the work in line with the reviewer's recommendation. Thank you again for your valuable feedback.

[Revision]

Revised Manuscript (Title)

Divalent Anion-Driven Framework Regulation in **Zr-Based** Halide Solid Electrolytes for All-Solid-State Batteries

2. In the abstract, "oxidation stability" should be replaced by "oxidative stability".

Response to comment 2: We appreciate the reviewer's suggestion to revise the term "oxidation stability" to "oxidative stability". While both expressions are often used interchangeably in the literature, "oxidation stability" was originally chosen to emphasize the experimentally determined resistance to oxidation (e.g., anodic decomposition potential). Nevertheless, we agree that "oxidative stability" is more consistent with the general description of material properties in this context, and we have revised the manuscript accordingly.

[Revision]

Revised Manuscript (Abstract)

Halide SEs are valued for their high ionic conductivity, **oxidative stability**, and ductility.

3. The X-ray diffraction analysis in Fig. 1a-c and Fig. 1d-f appears somewhat superficial. They observed the formation of Li₂O and LiCl at specific x compositions. I cannot find whether they did any Rietveld and estimate the phase fractions of Li₂O and LiCl from the diffractograms in these figures. Likewise, they should report the phase fraction of unreacted/residual Li₂ZrCl₆. The absence of this analysis diminishes the importance of their findings and makes their claims very speculative.

Response to comment 3: We thank the reviewer for the insightful comment. Although the synthesized samples exhibit crystallinity, they possess nanocrystalline features that result in significant background signals and peak broadening in the XRD patterns, introducing substantial uncertainty in phase quantification and the potential for misleading interpretations. These limitations significantly hinder reliable phase fraction quantification using XRD. Additionally, residual phases such as ZrO₂ and ZrS₂, which exhibit only short-range ordering, coexist within the samples and remain undetectable even when using high-energy X-rays.

Therefore, to obtain more reliable estimates of the phase fractions, we employed pair distribution function (PDF) analysis and extended X-ray absorption fine structure (EXAFS) fitting, both of which are less sensitive to long-range order and thus more suitable for accurate structural quantification in such systems. Similarly, refinements for compositions beyond $x = 1.0$ in the $x\text{Li}_2\text{O}-\text{ZrCl}_4$ and $x\text{Li}_2\text{S}-\text{ZrCl}_4$ systems were also not pursued, as significant reductions in first-shell intensity (Zr-Cl and Zr-S pairs in hcp and ccp phases, respectively) indicated the formation of the potential non-ionic conductive by-products (LiCl). In response to the reviewer's suggestion, we have now performed quantitative PDF and EXAFS analyses to estimate the respective phase fractions with improved reliability.

We performed PDF refinements to identify the dominant phase and quantify phase fraction in $x\text{Li}_2\text{O}-\text{ZrCl}_4$ and $x\text{Li}_2\text{S}-\text{ZrCl}_4$ system (**Supplementary Fig. 4,9** and **Supplementary Tables 1-6, 8-13**). Given that overwhelming scattering power from Zr in ZrO₂, ZrS₂, and the hcp, ccp phases compared to that of Li₂O and LiCl, the latter two were excluded in fitting to reduce parameter thereby securing fitting reliability. Additionally, anion exchange was not considered due to the following reasons. First, a slight first shell intensity reduction was observed in $x\text{Li}_2\text{O}-\text{ZrCl}_4$ ($x=0.6, 0.8, 1.0$) due to anion exchange (oxygen substitution) in hcp-Li₂ZrCl₆ (**Fig. 2a**). However, as this reduction is marginal because of the slight amount of oxygen substitution level, the fitting for hcp-Li₂ZrCl₆ was performed considering only the Zr-Cl coordination and excluding small amount of Zr-O bond in hcp-Li₂ZrCl₆. Second, the isoelectronic feature of Cl and S (18 electrons) make it heavily challenging to refine substitution fractions due to similar X-ray scattering power.

We also conducted EXAFS refinement to figure out Zr-Zr distance in ZrO₂/ZrS₂ and hcp/ccp lattice element selectively. (**Supplementary Fig. 6,11** and **Supplementary Tables 7,14**) We constructed EXAFS fitting model for ZrO₂ which nine coordinated Zr-Zr distances were divided into three subsets to accurately capture Zr-Zr correlations in ZrO₂. The site disorder at M2-M3 in the hcp phase was fixed at 50% based on our long-range region (15-30 Å) PDF fitting results, which demonstrate experimental observation for M2-M3 site disorder is near 0.5 (**Supplementary Tables 2,4 and 6**). The phase fraction results obtained from EXAFS analyses align well with those from PDF refinements, further reinforcing the reliability of our fitting methodology.

Although Li₂O was excluded from the refinements due to low scattering factors, the obtained fractions of ZrO₂ vs. hcp phase ($33:66 \cong 1:2$) and ZrS₂ vs. ccp phase ($7:16 \cong 3:7$), as shown in **Response Table 1**, closely matched computational predictions (interfacial substitution in hcp) $2\text{Li}_2\text{O} + 5\text{ZrO}_2 + 10\text{Li}_2\text{ZrCl}_6$ for $0.8\text{Li}_2\text{O}-\text{ZrCl}_4$ and (bulk substitution in ccp) $7\text{ZrS}_2 + 16\text{Li}_{2.25}\text{ZrCl}_{5.75}\text{S}_{0.25}$ for $0.8\text{Li}_2\text{S}-\text{ZrCl}_4$ in **Supplementary Table 18,19**). This agreement strongly suggests the negligible unidentified phases or additional features that could be attributed to alternative components in the $x\text{Li}_2\text{O}-\text{ZrCl}_4$ and $x\text{Li}_2\text{S}-\text{ZrCl}_4$ systems. Therefore, we suggest that the phase fractions of Li₂O, ZrO₂ and ZrS₂, as well as the amount of O and S substitution in the hcp and ccp phases closely align with computationally predicted values. This agreement between the reaction phases and their ratios, as experimentally observed and computationally predicted based on reaction energy calculations, reinforces the reliability of phase fraction estimations from both theoretical and experimental perspectives.

Composition	ZrO ₂ (PDF) (%)	hcp (PDF) (%)	ZrO ₂ (EXAFS) (%)	hcp (EXAFS) (%)
0.6Li ₂ O–ZrCl ₄	35.4	64.6	29.9	70.0
0.8Li₂O–ZrCl₄	34.9	65.1	33.12	67.9
1.0Li ₂ O–ZrCl ₄	50.1	49.9	48.2	51.8
Composition	ZrS ₂ (PDF) (%)	ccp (PDF) (%)	ZrS ₂ (EXAFS) (%)	ccp (EXAFS) (%)
0.6Li ₂ S–ZrCl ₄	35.8	64.2	33.3	66.7
0.8Li₂S–ZrCl₄	31.9	68.1	29.9	70.1
1.0Li ₂ S–ZrCl ₄	28.6	71.4	23.4	76.56

Response Table 1. Phase fraction of PDF and EXAFS fitting in xLi₂O–ZrCl₄ (x = 0.6, 0.8, 1.0) and xLi₂S–ZrCl₄ (x = 0.6, 0.8, 1.0)

(Added in revision) Supplementary Fig. 4. Observed and calculated PDF short-range region (1.5–4.1 Å, top) and long-range region (15–30 Å, bottom) fitting curves for 0.6Li₂O–ZrCl₄, 0.8Li₂O–ZrCl₄ and 1.0Li₂O–ZrCl₄. The detailed results are summarized in **Supplementary Table 1-6**.

(Added in revision) Supplementary Fig. 6. Zr K-edge EXAFS results. The real/imaginary part of FT (left) and $k^3\chi(k)$ (right) based on R-space curve fitting of $x\text{Li}_2\text{O-ZrCl}_4$ ($x = 0.6, 0.8, 1.0$). The detailed results are summarized in **Supplementary Table 7**.

(Added in revision) Supplementary Fig. 9. Observed and calculated PDF short-range region (1.5-4.1 Å, top) and long-range region (15-30 Å, bottom) fitting curves for 0.6Li₂S-ZrCl₄, 0.8Li₂S-ZrCl₄ and 1.0Li₂S-ZrCl₄. The detailed results are summarized in **Supplementary Table 8-13**.

(Added in revision) Supplementary Fig. 11. Zr K-edge EXAFS results. The real/imaginary part of FT (left) and $k^3\chi(k)$ (right) based on R-space curve fitting of $x\text{Li}_2\text{S-ZrCl}_4$ ($x = 0.6, 0.8, 1.0$). The detailed results are summarized in **Supplementary Table 14**.

Following the reviewer's valuable recommendations, we have included detailed descriptions of our quantitative analysis and phase fraction determination in the manuscript and **Supplementary Texts**. The *Methods* section has also been updated accordingly.

[Revision]

Revised manuscript

- **Line 151:** Additionally, the presence of a Zr–O pair in monoclinic ZrO_2 (**Supplementary Fig. 3 and Supplementary Text 1**) at 2.1 Å in all compositions suggests the formation of nanosized ZrO_2 domains. Additionally, PDF fitting was conducted to estimate the phase fractions of ZrO_2 and the hcp phase (**Supplementary Fig. 4 and Supplementary Table 1-6**). For $x = 0.6$ and 0.8, the long-range region (15–30 Å), where only crystalline phases contribute, was well-reproduced using a single hcp model with ~50% M2–M3 site disorder, confirming the coexistence of crystalline hcp and short range ordered ZrO_2 domains. In contrast, for $x = 1.0$, the fitting yielded a substantially higher R_w value, suggesting progressive structural distortion of the hcp framework due to oxygen incorporation and the onset of phase transformation (**Supplementary Table 6**).
- **Line 164:** Furthermore, the extended X-ray absorption fine structure (EXAFS) spectra in **Fig. 2c** show overall positive peak shifts toward a longer region for the Zr–Cl bond in the hcp phase around

2.0 Å, as well as broadening of the Zr-Zr bonding peaks from 3.0 to 3.4 Å compared to bare hcp-LZC and ZrO₂, respectively (**Supplementary Fig. 5**). Also, our EXAFS fitting results clearly show such elongation (**Supplementary Fig. 6**).

- **Line 171:** Conversely, the substitution of Cl in ZrO₂ increases the Zr-Zr distance and local distortion due to the larger anion radius of Cl, suggesting anion exchange between ZrO₂ and Li₂ZrCl₆, consistent with our previous study.³¹ Additionally, phase fraction for ZrO₂ in xLi₂O-ZrCl₄ (x = 0.6-1.0) are well aligned with our PDF fitting results, further supporting reliability of our refinement (**Supplementary Table 7** and **Supplementary Text 2**).
- **Line 190:** Notably, small peaks at ~4.98 Å, attributed to ZrS₂, suggest the presence of ZrS₂ with only short-range ordering. The PDF fitting results and phase fraction of ZrS₂ for xLi₂S-ZrCl₄ (x = 0.6-1.0) are shown in **Supplementary Fig. 9** and **Supplementary Table 8-13**.
- **Line 195:** The EXAFS results in **Fig. 3c** display a noticeable positive peak shift upon increasing Li₂S, with low fractions of Li₂S (x ≤ 1.2) showing a similar tendency of divalent-anion-driven bond elongation observed in xLi₂O-ZrCl₄, but within the ccp structure (**Supplementary Fig. 10**) and our EXAFS fitting further support such elongation (**Supplementary Fig. 11**). Additionally, the phase fraction of ZrS₂ refined by EXAFS fitting demonstrates similarity to those predicted by PDF, analogous to the case of O-substituted hcp phases. (**Supplementary Table 14** and **Supplementary Text 3**).

Revised Manuscript (Methods)

XANES and EXAFS data were processed using the Demeter software package. The extracted EXAFS signal, $k^3\chi(k)$, was Fourier transformed over a k -range of 3.1-12.8 Å⁻¹ for xLi₂O-ZrCl₄ and 3.8-11.2 Å⁻¹ for xLi₂S-ZrCl₄ to obtain magnitude plots of the EXAFS spectra. EXAFS are fitted over a r -range of 1.35-3.5 Å for xLi₂O-ZrCl₄ and 1.7-3.7 Å for xLi₂S-ZrCl₄. Detailed fitting procedures for PDF and EXAFS are shown in **Supplementary Text 2**.

Revised Supplementary Information (Figure)

(Added in revision) Supplementary Fig. 4. Observed and calculated PDF short-range region (1.5-4.1 Å, top) and long-range region (15-30 Å, bottom) fitting curves for 0.6Li₂O-ZrCl₄, 0.8Li₂O-ZrCl₄ and 1.0Li₂O-ZrCl₄. The detailed results are summarized in **Supplementary Table 1-6**.

(Added in revision) Supplementary Fig. 6. Zr K-edge EXAFS results. The real/imaginary part of FT (left) and $k^3\chi(k)$ (right) based on R-space curve fitting of xLi₂O-ZrCl₄ (x = 0.6, 0.8, 1.0). The detailed results are summarized in **Supplementary Table 7**.

(Added in revision) Supplementary Fig. 9. Observed and calculated PDF short-range region (1.5-4.1 Å, top) and long-range region (15-30 Å, bottom) fitting curves for 0.6Li₂S-ZrCl₄, 0.8Li₂S-ZrCl₄ and 1.0Li₂S-ZrCl₄. The detailed results are summarized in **Supplementary Table 8-13**.

(Added in revision) Supplementary Fig. 11. Zr K-edge EXAFS results. The real/imaginary part of FT (right) and $k^3\chi(k)$ (left) based on R-space curve fitting of xLi₂S-ZrCl₄ (x = 0.6, 0.8, 1.0). The detailed results are summarized in **Supplementary Table 14**.

Revised Supplementary Information (Table)

(Added in revision) Supplementary Table 1-6. PDF fitting results of xLi₂O-ZrCl₄ (x = 0.6, 0.8, 1.0)

(Added in revision) Supplementary Table 7. EXAFS fitting results of xLi₂O-ZrCl₄ (x = 0.6, 0.8, 1.0) corresponding to Supplementary Fig. 6.

(Added in revision) Supplementary Table 8-13. PDF fitting results of xLi₂S-ZrCl₄ (x = 0.6, 0.8, 1.0)

(Added in revision) Supplementary Table 14. EXAFS fitting results of xLi₂S-ZrCl₄ (x = 0.6, 0.8, 1.0) corresponding to Supplementary Fig. 11.

Revised Supplementary Information (Text)

(Added in revision) Supplementary Text 2. Phase ratio of PDF and EXAFS fitting model for Supplementary Fig. 4,6,9,11 and Supplementary Table 1-14.

Given that overwhelming scattering power from Zr in ZrO₂, ZrS₂, and the hcp, ccp phases compared to that of Li₂O and LiCl, the latter two were excluded in PDF and EXAFS fitting to reduce parameter thereby securing fitting reliability. Additionally, anion exchange was not considered in PDF and EXAFS refinement due to the following reasons:

1. A slight first shell intensity reduction was observed in xLi₂O–ZrCl₄ (x = 0.6, 0.8, 1.0) due to anion exchange in hcp-Li₂ZrCl₆. (**Fig. 2a**) However, as this reduction is marginal, the fitting for hcp-Li₂ZrCl₆ was performed considering only the Zr-Cl coordination and excluding small amount of Zr-O bond.
2. The isoelectronic feature of Cl and S (18 electrons) make it heavily challenging to refine substitution fraction due to similar X-ray scattering power.

We constructed EXAFS fitting model for ZrO₂ which nine coordinated Zr-Zr distances were divided into three subsets to capture Zr-Zr correlations in amorphous ZrO₂. Continually, site disorder at M2-M3 in the hcp phase was fixed at 50% based on our PDF fitting results, which demonstrate M2-M3 site disorder is near 0.5. (**Supplementary Tables 2, 4, 6**)

Although Li₂O and Li₂S were excluded from the refinements, the obtained fractions of ZrO₂ vs. hcp phase and ZrS₂ vs. ccp phase by both PDF and EXAFS fitting closely matched with computational predictions (**Supplementary Tables 1-14**) This agreement strongly suggests the absence of residual precursor or unreacted Li₂ZrCl₆ phases in the xLi₂O–ZrCl₄ and xLi₂S–ZrCl₄ systems. Therefore, we suggest that the phase fractions of Li₂O and Li₂S, as well as the amount of O and S substitution in the hcp and ccp phases, closely align with computationally predicted values.

4. I'm surprised that ZrO₂ is only detected in the PDF analysis and doesn't appear in the XRD diffractogram. Have they tried to fingerprint ZrO₂ in the Synchrotron diffractogram? Is this monoclinic or cubic ZrO₂? As per comment #3, these authors should attempt to quantify the amount of ZrO₂ in the sample.

Response to comment 4: We are thankful to reviewer's careful comment.

1. Undetectable ZrO₂ in XRD but observable ZrO₂ in PDF

As pointed out, ZrO₂ was not observed in the synchrotron XRD patterns (**Fig. 1c**), but its presence is clearly revealed through both high-resolution transmission electron microscopy (**Fig. 2e**) and pair distribution function (**Fig. 2a, Supplementary Fig. 3**) analysis. To further investigate the structural and chemical nature of ZrO₂, we performed scanning transmission electron microscopy combined with electron energy loss spectroscopy (STEM-EELS) analysis (**Response Fig. 1**).

Response Fig. 1. STEM image and EELS (Li, Zr, Cl, O, S) elemental mapping (Li, Zr, Cl, O, S) of 0.8Li₂O–ZrCl₄ and 0.8Li₂S–ZrCl₄.

The STEM-EELS analysis reveals spatially localized regions corresponding to ZrO₂, clearly distinguishing them from the surrounding matrix. Consistent with the HRTEM results, these domains are approximately 10 nm in size. According to the Scherrer equation ($\tau = \frac{K\lambda}{\beta \cos\theta}$, where τ : **detectable grain size**, K : shape factor, λ : **wavelength**, β : line broadening at FWHM, θ : Bragg angle), such nanosized and disordered domains generate broad diffraction features that fall below the detection limit of XRD, especially in the presence of dominant crystalline hcp phase reflections. These observations collectively explain the absence of discernible ZrO₂ peaks in the synchrotron XRD diffractograms. To provide further clarity, we present the evolution of XRD and PDF patterns for 0.8Li₂O–ZrCl₄ with varying synthesis times (**Response Fig. 2**).

Response Fig. 2. Synchrotron XRD (left) and PDF analysis (right) of 0.8Li₂O–ZrCl₄ with increasing synthesis time.

During the early stage of the synthesis process, Li_2O and ZrCl_4 are transiently observed on the XRD patterns as intermediate by-products, likely forming LiCl . However, these phases disappear completely as the reaction proceeds. Even after extended milling durations beyond the optimal synthesis time (24 h), such as for 30 h, no secondary phases other than the hcp phase are detected in the XRD patterns. This may misleadingly suggest phase-pure formation when relying solely on XRD, even though the short-range ordered ZrO_2 is present but remains undetectable due to its nanoscale domain size and lack of long-range order. In contrast, the intensity of the ZrO_2 -related peak in the PDF data continues to increase over time. This discrepancy indicates that ZrO_2 forms nanoscale domains rather than growing into larger crystallites detectable by XRD measurement, remaining only observable via the short-range structural sensitivity of PDF analysis.

2. Presence of monoclinic ZrO_2 and Phase fraction of ZrO_2

Synchrotron XRD shows no identifiable peaks corresponding to monoclinic or cubic ZrO_2 (**Fig. 1c**), indicating the formation of nanosized ZrO_2 phases, making direct quantification through diffraction impossible. In contrast, PDF and EXAFS analyses effectively quantified features associated with ZrO_2 , typically invisible in conventional diffraction. Also, we conducted PDF calculations using pdfgui software of xPDFsuite⁵¹ to identify whether ZrO_2 is cubic or monoclinic. The PDF calculations clearly revealed characteristic features of monoclinic ZrO_2 in the $0.8\text{Li}_2\text{O-ZrCl}_4$ composition, confirming the presence of monoclinic ZrO_2 (**Supplementary Fig. 3**). As stated in our **Response to Comment 3**, quantitative determination of the phase fractions was made possible through the analysis of both PDF and EXAFS data.

(Added in revision) Supplementary Fig. 3. Observed and calculated PDF fitting curves for $0.8\text{Li}_2\text{O-ZrCl}_4$ compared with cubic ZrO_2 , monoclinic ZrO_2 and hcp- Li_2ZrCl_6 phase. The detailed results are summarized in **Supplementary Text 1**.

We have revised the manuscript and added a supplementary text discussing the presence of monoclinic ZrO_2 . Indeed, we successfully fitted the short-range region of PDF (1.5-4.1 \AA) and EXAFS, including the ZrO_2 feature of short-range order. For detailed quantitative analysis, please refer to our response to **Response to Comment 3**.

[Revision]

Revised manuscript

- **Line 151:** Additionally, the presence of a Zr–O pair in monoclinic ZrO₂ (**Supplementary Fig. 3 and Supplementary Text 1**) at 2.1 Å in all compositions suggests the formation of nanosized ZrO₂ domains.

Revised Supplementary Information (Figure)

(Added in revision) Supplementary Fig. 3. Observed and calculated PDF fitting curves for 0.8Li₂O–ZrCl₄ compared with cubic ZrO₂, monoclinic ZrO₂ and hcp-Li₂ZrCl₆ phase. The detailed results are summarized in **Supplementary Text 1**.

Revised Supplementary Information (Text)

(Added in revision) Supplementary Text 1. The presence of monoclinic ZrO₂ We conducted PDF calculations to identify whether ZrO₂ is cubic or monoclinic. The PDF calculations clearly revealed characteristic features of monoclinic ZrO₂ in the 0.8Li₂O–ZrCl₄ composition, confirming the presence of monoclinic ZrO₂ (**Supplementary Fig. 3**).

5. I was wondering if Raman would be a better technique for describing the amorphous portion of these anion-enriched materials.

Response to comment 5: We thank the reviewer for the insightful question about Raman spectroscopy. Raman analysis is expected to reflect the bonding characteristics of the dominant phase, and since the ZrCl₆ polyhedra in our hcp and ccp materials are all discretely separated, such bonding features should be identifiable. Indeed, it is a powerful tool to probe structural features in materials with reduced crystallinity or features of polyhedral unit (such as conner-sharing, edge-sharing).

In systems where amorphization or significant loss of crystallinity occurs due to anion substitution, the vibrational modes associated with characteristic polyhedral units (e.g., MX₄^{a-}, MX₆^{a-} and MX₉^{a-} (M = metal and X= anion)) tend to lose their well-defined spectral signatures. In addition, the Raman spectrum of ZrCl₄ shows characteristic signatures of zigzag, polymer-like chain-structured [(ZrCl_{4/2})Cl₂]_n of bridged octahedra.^{R1,R2} If Zr octahedra formed with a bridging (corner-sharing) configuration by amorphization, similar peaks would be expected to appear in the Raman spectrum. As such, Raman spectroscopy allows one to qualitatively assess the degree of amorphization and provides insight into the evolution of local bonding environments under anion-enriched conditions. To provide local structural feature, we measured the Raman spectra of ZrCl₄, hcp-LZC, ccp-LZC and the 0.8Li₂A–ZrCl₄ (A = S, O) samples (**Response Fig. 3**).

Response Fig. 3. Raman spectra of ZrCl₄, hcp-Li₂ZrCl₆ (hcp-LZC), 0.8Li₂O–ZrCl₄, ccp-Li₂ZrCl₆ (ccp-LZC) and 0.8Li₂S–ZrCl₄.

In our case, the Raman spectra (now included as main figure in the revised manuscript) show that the characteristic vibrational modes of Zr–Cl (≈ 325 and ≈ 161 cm^{-1}) in isolated octahedron remain clearly identifiable across all compositions (hcp-LZC, $0.8\text{Li}_2\text{O-ZrCl}_4$, ccp-LZC and $0.8\text{Li}_2\text{S-ZrCl}_4$). Also, in contrast to ZrCl_4 , the signatures of bridging octahedra (≈ 410 and ≈ 145 cm^{-1}) disappeared, and two distinct peaks at ≈ 325 and ≈ 161 cm^{-1} emerged, corresponding to the A_{1g} stretching and F_{2g} bending modes, respectively, which are characteristic of elpasolite-type compounds such as $\text{Cs}_2\text{LiYCl}_6$ and $\text{Cs}_2\text{NaYCl}_6$.^{R3} These modes closely resemble those observed in closed packed structures (hcp and ccp), suggesting that the main phase with polyhedral framework is structurally preserved despite the introduction of divalent anions. This observation indicates that the divalent anion incorporation does not lead to significant amorphization but instead maintains a close-packed structural environment that retains the key bonding features in main phase.

Additionally, to assess the presence and structural impact of amorphous phase in our system, we performed detailed PDF analyses and examined both local and intermediate-range orders. The results collectively indicate that while short-range ordered ZrO_2 and ZrS_2 phases account for approximately 30% of the structure, amorphization within the conductive hcp and ccp frameworks is not observed (**Supplementary Fig. 4,9** and **Supplementary Table 3,4,10,11**).

First, the peaks corresponding to the hcp and ccp phases were well maintained within the PDF $G(r)$ range of 1.5-6 Å for the compositions $x\text{Li}_2\text{O-ZrCl}_4$ and $x\text{Li}_2\text{S-ZrCl}_4$ ($x = 0.6, 0.8, 1.0$) (**Fig. 2a, 3a**). Given that ~ 6 Å corresponds approximately to the lattice parameter (interlayer distance) along the c -axis of Li_2ZrCl_6 (6.030 Å for hcp and 6.376 Å for ccp), the sustained intensity of this peak suggests that the long-range periodicity associated with these lattice frameworks is preserved. The consistent peak behavior in this region thus supports the retention of hcp and ccp structural motifs without meaningful evidence of amorphous phase development. Thus, we conclude that there is a negligible amount of amorphous phase formation in these materials at compositions corresponding to the highest ionic conductivity ($x = 0.6, 0.8, 1.0$).

In this context, Raman spectroscopy has proven highly effective not only for revealing the negligible amorphization of main phase, but also for confirming the robustness of local coordination environments in structurally preserved halide frameworks. We thank the reviewer for motivating this clarification and have included this data more prominently in the revised manuscript.

Accordingly, we have incorporated the Raman spectra into **Fig. 2d** and **3d**, added the measurement details to the **Methods** section, and expanded the corresponding discussion in the main text to highlight its significance in supporting structural interpretations. Additionally, we have included **Supplementary text 3** for amorphous phase and the corresponding PDF fitting data to further clarify the discussion on potential amorphization and reinforce the structural interpretation.

(Added in revision) **Fig. 2d** Raman spectra of ZrCl_4 , hcp- Li_2ZrCl_6 (hcp-LZC) and $0.8\text{Li}_2\text{O-ZrCl}_4$, (**left**) and **Fig. 3d** Raman spectra for ZrCl_4 , ccp- Li_2ZrCl_6 (ccp-LZC) and $0.8\text{Li}_2\text{S-ZrCl}_4$. (**right**)

[Revision]

Revised Manuscript

- **Line 145:** PDF analysis, X-ray absorption spectroscopy (XAS), Raman spectroscopy and high-resolution transmission electron microscopy (HRTEM) images are presented in **Fig. 2** and **Fig. 3**, respectively, enabling detailed analysis of the local structure of $x\text{Li}_2\text{A}-\text{ZrCl}_4$.
- **Line 177:** In **Fig. 2d**, the Raman spectrum of ZrCl_4 exhibited characteristic features of polymeric zigzag chains with bridged octahedra, whereas hcp-LZC and $0.8\text{Li}_2\text{O}-\text{ZrCl}_4$ showed distinct peaks at $\approx 325\text{ cm}^{-1}$ and $\approx 161\text{ cm}^{-1}$, respectively, indicative of isolated octahedra in lattice.²³
- **Line 208:** Since the Zr octahedra are also isolated in the ccp lattice, the Raman spectrum in **Fig. 3d** also exhibited distinct peaks, clearly differentiating it from ZrCl_4 . Moreover, the HRTEM images with FFT pattern (**Fig. 3e**) confirm a single ccp phase, indicating the homogeneous distribution of the sulfur-substituted ccp phase across the material.

Revised Manuscript (Methods)

(Added in revision) Raman spectra were collected using a ARAMIS spectrometer (Horiba Jobin Yvon) equipped with an Ar-ion laser operating at an excitation wavelength of 785 nm.

Revised Figure

(Added in revision) Fig. 2d Raman spectra for ZrCl_4 , hcp-LZC, and $0.8\text{Li}_2\text{O}-\text{ZrCl}_4$.

(Added in revision) Fig. 3d Raman spectra for ZrCl_4 , ccp-LZC, and $0.8\text{Li}_2\text{S}-\text{ZrCl}_4$.

Revised Supplementary Information (Text)

(Added in revision) Supplementary text 3. Amorphous phase

The peaks corresponding to the hcp and ccp phases were well maintained within the PDF $G(r)$ range of 1.5-6 Å for the compositions $x\text{Li}_2\text{O}-\text{ZrCl}_4$ and $x\text{Li}_2\text{S}-\text{ZrCl}_4$ ($x = 0.6, 0.8, 1.0$) (**Fig. 2a, 3a**). Given that ~ 6 Å corresponds approximately to the lattice parameter (interlayer distance) along the c-axis of Li_2ZrCl_6 (6.030 Å for hcp and 6.376 Å for ccp), the sustained intensity of this peak suggests that the long-range periodicity associated with these lattice frameworks is preserved. The consistent peak behavior in this region thus supports the retention of hcp and ccp structural motifs without meaningful evidence of amorphous phase development. Thus, we conclude that there is a negligible amount of amorphous phase formation in these materials at compositions corresponding to the highest ionic conductivity ($x = 0.6, 0.8, 1.0$). While structural amorphization in Zr-based oxyhalide has been proposed in prior studies as a mechanism for enhancing ionic conductivity, particularly within each respective systems, our compositions and results suggest a different origin for the observed conductivity. Samples with lower crystallinity (such as $1.0\text{Li}_2\text{O}-\text{ZrCl}_4$ and $1.2\text{Li}_2\text{O}-\text{ZrCl}_4$) exhibited reduced ionic conductivity compared to the more crystalline $0.8\text{Li}_2\text{O}-\text{ZrCl}_4$ composition (**Fig. 1b,c**). This behavior shows that amorphization was not linked to improved ionic transport, suggesting instead that the high ionic conductivity in our system originates from an optimized crystalline structure rather than from any amorphous contribution. Thus, we conclude that there is only negligible amount of amorphous phase in these materials at compositions corresponding to the highest ionic conductivity ($x = 0.6, 0.8, 1.0$).

6. The lack of determination of the phase fraction Li_2O , LiCl , ZrO_2 , and residual/unreacted Li_2ZrCl_6 partially invalidates the interpretation of the impedance data. In particular, they don't analyze the grain boundaries of these materials, and whether these impurities (or unreacted) phases segregate at the grain boundaries, thus affecting the impedance responses.

Response to comment 6: We appreciate the reviewer's observation regarding the need to determine the phase fractions and assess the possible impact of secondary or unreacted phases on the impedance response. As partially addressed in **response to comment 3**, we performed PDF and EXAFS fitting, which clearly revealed the coexistence of 35% ZrO_2 , and 65% hcp phases in a nanocomposite-like mixture. The relative phase

fractions were estimated, indicating that hcp phases remain the dominant phase, while Li_2O and ZrO_2 are present in smaller quantities.

If ionic conduction were significantly influenced by grain boundaries or residual components, a meaningful semicircle feature as resistance should appear in the impedance spectra. We fully understand the reviewer's concern regarding the potential influence of grain boundaries and secondary/unreacted phases on the impedance response. If grain boundaries or insulating phases affect conduction property, they could introduce additional resistive components, thereby biasing the interpretation of bulk ionic conductivity. However, at high temperatures, the ionic conductivity becomes sufficiently high such that the impedance response often appears as a near-linear profile (Warburg-like behavior), which may obscure the presence of multiple components. To address this, we performed low-temperature EIS measurements which are more appropriate for deconvoluting individual contributions such as bulk, grain boundary and interphase.

(Added in revision) Supplementary Fig. 16. Nyquist plots of EIS results for hcp- Li_2ZrCl_6 , $0.8\text{Li}_2\text{O}-\text{ZrCl}_4$, ccp- Li_2ZrCl_6 and $0.8\text{Li}_2\text{S}-\text{ZrCl}_4$ measured at various temperatures ranging from $-20\text{ }^\circ\text{C}$ to $60\text{ }^\circ\text{C}$.

In **Supplementary Fig. 16**, these data clearly show a single semicircle, supporting the interpretation that the observed ionic transport is dominated by a single, percolating phase with negligible contribution from grain boundaries or insulating domains. Furthermore, the low-temperature EIS data do not exhibit any distinguishable semicircular features that could be attributed to separate components such as grain boundaries or another domain. To further assess the possible presence of grain boundaries or other resistive components affecting ion transport, we performed equivalent circuit fitting of the $-20\text{ }^\circ\text{C}$ EIS data using both single-phase and multi-component models (**Supplementary Fig. 17**).

(Added in revision) Supplementary Fig. 17. Comparison of Nyquist plot with equivalent circuit fitting using single-phase and multi-component models for 0.8Li₂O–ZrCl₄ (top) and 0.8Li₂S–ZrCl₄ (bottom) at -20°C.

The single-phase model yielded a significantly better fit, whereas the resistance value associated with the secondary component in the multi-component model converged to a negligible value ($R_2 \approx 4.0 \times 10^{-35} \Omega$). This result strongly indicates the absence of grain boundary contributions or secondary resistive phases within the resolution of the EIS analysis. Instead, the Nyquist plots of 0.8Li₂A–ZrCl₄ (A = O, S) display a consistent shape across temperatures, closely resembling those observed for both hcp-LZC and ccp-LZC. It is consistent that halide-based electrolytes such as Li₂ZrCl₆ are known for their high ductility, which leads to negligible grain boundary resistance.^{R4-6} These suggest that the measured conductivity arises from a single major conducting phase. Considering that ZrO₂ does not contain Li⁺ and Li₂O possesses extremely low ionic conductivity ($\sim 10^{-12} \text{ S cm}^{-1}$)^{R7}, their contribution to overall ion transport is negligible.

Furthermore, SEM image and EDS mapping of the pelletized samples confirms that elemental distribution is uniform and that there is minimal porosity, supporting the densely packed microstructure without significant phase segregation at the grain boundaries (**Response Fig. 5**).

Response Fig. 5. SEM-EDS data of $0.8\text{Li}_2\text{O-ZrCl}_4$, $0.8\text{Li}_2\text{S-ZrCl}_4$ and Li_2ZrCl_6 .

To provide a more comprehensive interpretation of our impedance data, we have adopted the analytical framework proposed by Li et al.^{R8} which enables the decoupling of critical parameters governing ionic conductivity in solid-state electrolytes. Following this approach, we systematically extracted and analyzed the following key parameters from our experimental data as presented in **Supplementary Fig. 18** and **Supplementary Table 20**.

(Added in revision) Supplementary Fig. 18. Arrhenius plot of hopping frequency, carrier concentration factor, migration entropy of hcp- Li_2ZrCl_6 , $0.8\text{Li}_2\text{O-ZrCl}_4$, ccp- Li_2ZrCl_6 and $0.8\text{Li}_2\text{S-ZrCl}_4$.

(Added in revision) Supplementary Table 20. The calculated values related to ionic conductivity (pre-factor σ_0 , activation energy E_a , migration energy E_m , mobile carrier formation energy E_f , hopping frequency ν , carrier concentration factor C , migration entropy ΔS_m) of hcp- Li_2ZrCl_6 , $0.8\text{Li}_2\text{O-ZrCl}_4$, ccp- Li_2ZrCl_6 and $0.8\text{Li}_2\text{S-ZrCl}_4$.

	hcp- Li_2ZrCl_6	$0.8\text{Li}_2\text{O-ZrCl}_4$,	ccp- Li_2ZrCl_6	$0.8\text{Li}_2\text{S-ZrCl}_4$
σ (25 °C)	$3.7 \times 10^{-4} \text{ S cm}^{-1}$	$1.78 \times 10^{-3} \text{ S cm}^{-1}$	$4.3 \times 10^{-6} \text{ S cm}^{-1}$	$1.01 \times 10^{-3} \text{ S cm}^{-1}$
σ_0	$77848 \text{ S cm}^{-1} \text{ K}^{-1}$	$29836 \text{ S cm}^{-1} \text{ K}^{-1}$	$99451 \text{ S cm}^{-1} \text{ K}^{-1}$	$69324 \text{ S cm}^{-1} \text{ K}^{-1}$
E_a	346.39 meV	280.96 meV	466.75 meV	317.22 meV
E_m	333.11 meV	276.03 meV	445.53 meV	315.04 meV
E_f	13.28 meV	4.93 meV	21.22 meV	2.18 meV
ν	$1.28 \times 10^6 \text{ Hz at } 0^\circ\text{C}$	$5.17 \times 10^6 \text{ Hz at } 0^\circ\text{C}$	$4.30 \times 10^4 \text{ Hz at } 0^\circ\text{C}$	$3.14 \times 10^6 \text{ Hz at } 0^\circ\text{C}$
C	4.36×10^{-8}	4.60×10^{-8}	1.15×10^{-8}	2.41×10^{-8}

	$\text{S cm}^{-1} \text{ Hz}^{-1} \text{ K}$	$\text{S cm}^{-1} \text{ Hz}^{-1} \text{ K}$	$\text{S cm}^{-1} \text{ Hz}^{-1} \text{ K}$	$\text{S cm}^{-1} \text{ Hz}^{-1} \text{ K}$
ΔS_m	$1.56 \times 10^{-4} \text{ eV K}^{-1}$	$2.46 \times 10^{-4} \text{ eV K}^{-1}$	$4.42 \times 10^{-5} \text{ eV K}^{-1}$	$1.38 \times 10^{-4} \text{ eV K}^{-1}$

In the case of $0.8\text{Li}_2\text{O}-\text{ZrCl}_4$, the migration energy ($E_m = 276.03 \text{ meV}$) is notably lower than that of hcp-LZC ($E_m = 333.11 \text{ meV}$), indicating a reduction in the energy barrier for Li^+ migration. In addition, the mobile carrier formation energy ($E_f = 4.93 \text{ meV}$) is smaller than that of hcp-LZC ($E_f = 13.28 \text{ meV}$), suggesting that Li^+ carriers are more readily formed in the O-substituted structure. However, the carrier concentration factor in the O-substituted structure ($C = 4.60 \times 10^{-8} \text{ S cm}^{-1} \text{ Hz}^{-1} \text{ K}^{-1}$) remains comparable to that of hcp-LZC ($C = 4.36 \times 10^{-8} \text{ S cm}^{-1} \text{ Hz}^{-1} \text{ K}^{-1}$), with only a slight increase, this marginal change suggests that the total number of detectable mobile carriers is not drastically altered. Indeed, this implies that although the total number of Li^+ ions may have increased due to O^{2-} substitution and associated charge compensation, a slight fraction of these additional Li^+ ions are situated in strongly coordinating environments near oxygen. These $\text{Li}-\text{O}$ coordinated species experience deeper local potential wells and reduced mobility compared to Li^+ coordinated by Cl^- , effectively making them non-detectable on the timescale probed by conductivity measurements. That is, although the formation of mobile carriers has become more favorable (lower E_f), the number of lithium ions participating in conduction remains comparable to that of hcp-LZC or is only slightly increased (comparable C).

While the amount of lithium participating in long-range transport does not substantially differ, the significant increase in both hopping frequency and migration entropy indicates enhanced lattice distortion within the structure, as demonstrated by our DFT calculations (**Fig. 5b**). This structural perturbation is further supported by the Raman spectra (**Fig. 2d**), which displays slightly broader features compared to hcp-LZC, reflecting increased dynamic disorder. It is evident that the observed improvement in ionic conductivity is primarily driven by structural regulation effects, particularly the reduction in migration energy. These energetic advantages are accompanied by a higher hopping frequency ($\nu = 5.17 \times 10^6 \text{ Hz}$ at 0°C) and greater migration entropy ($\Delta S_m = 2.46 \times 10^{-4} \text{ eV K}^{-1}$), both of which contribute to faster and more thermodynamically favorable ion migration. These findings collectively demonstrate that oxygen incorporation fundamentally reshapes the local lattice environment by introducing significant structural distortion, particularly at Li sites. This distortion lowers the migration energy and enhances dynamic lattice fluctuations, ultimately serving as the primary driver for the observed improvement in ionic conductivity. Thus, the enhanced transport properties in the O-substituted system arise not from an increase in mobile carrier concentration, but from structurally induced distortions that boost ionic mobility.

In case of sulfur, migration energy in $0.8\text{Li}_2\text{S}-\text{ZrCl}_4$ ($E_m = 315.04 \text{ meV}$) is significantly lower than that of ccp-LZC ($E_m = 445.53 \text{ meV}$), indicating that S^{2-} substitution into ccp-LZC significantly flattens the energy landscape for Li^+ migration. More notably, the formation energy of mobile carriers is drastically reduced by an order of magnitude (2.18 meV vs. 21.22 meV), which strongly suggests a fundamentally altered environment for carrier generation compared to oxygen-substituted systems. This reduction is especially meaningful in the context of ccp-type Zr-based frameworks, where a small cationic radius typically results in extremely low Li-ion conductivity^{R9}; thus, the observed low E_f implies not only the formation of mobile Li ions in such typically inactive lattices, but also a fundamental modification of the lattice framework that enables and stabilizes their presence. Furthermore, the hopping frequency increases by nearly two orders of magnitude ($\sim 10^6 \text{ Hz}$ vs. $\sim 10^4 \text{ Hz}$), directly contributing to the enhanced dynamic behavior of Li ions. This is accompanied by a more than two-fold increase in the carrier concentration factor ($C = 2.41 \times 10^{-8} \text{ S}\cdot\text{cm}^{-1}\cdot\text{Hz}\cdot\text{K}^{-1}$), highlighting the synergistic effect of faster ion dynamics.

In contrast to the oxygen-substituted structure, where the carrier concentration factor remains nearly unchanged, the more than two-fold increase observed here highlights not only the reduced energy required for mobile carrier formation, but also the substantially increased number of carriers actively participating in conduction. Crucially, the migration entropy (ΔS_m) also increases significantly, by approximately 3–4 times ($1.38 \times 10^{-4} \text{ eV K}^{-1}$), reflecting increased vibrational freedom and dynamic lattice flexibility. Collectively, these results indicate that the lattice itself undergoes a profound transformation, fundamentally altering the

characteristics of ionic transport within the structure. This dynamic lattice behavior is closely linked to the structural changes induced by sulfur substitution. In the case of ccp-LZC, the small ionic radius of Zr^{4+} initially favors the formation of a metastable hcp-type framework during mechanochemical synthesis, which then transforms into a thermodynamically stable ccp-type structure upon annealing. In this ccp framework, Li^+ migration along the ab-plane is inherently sluggish due to tight interlayer spacing and the absence of structural pillars in the Li layer.^{R10-12} To overcome this, we introduced S^{2-} into the framework, where the increased repulsion between sulfur atoms leads to an expansion of the interlayer distance.

(Added in revision) Supplementary Fig. 24. Li probability density at 600 K in ~ 300 ps (isosurface value $P = P_{\max}/100$) of $ccp-Li_2ZrCl_6$ (top-left) and $ccp-Li_{2.25}ZrCl_{5.75}S_{0.25}$ (top-right). Mean square displacement (MSD) of AIMD simulations at 600 K in ~ 300 ps of $ccp-Li_2ZrCl_6$ (down-left) and $ccp-Li_{2.25}ZrCl_{5.75}S_{0.25}$ (down-right). Sulfur incorporation induces lattice expansion and structural distortion in the ccp framework, activating three-dimensional ionic conduction otherwise limited to the ab plane.

This structural relaxation facilitates Li^+ migration along the ab-plane, as clearly evidenced by the enhanced in-plane mean square displacement (MSD) profiles (**Supplementary Fig. 24**). Since ab-plane transport dominates ionic conductivity in the ccp-type LZC structure, this targeted expansion directly contributes to the significant enhancement in overall ionic conductivity. Collectively, the marked reductions in both E_m and E_f , along with enhanced hopping dynamics and entropy-driven mobility, serve as key factors in transforming a conduction inactive phase ($\sim 10^{-6} \text{ S cm}^{-1}$) into a highly conductive one ($\sim 10^{-3} \text{ S cm}^{-1}$). These findings highlight the pivotal role of lattice dynamics and structural regulations in enabling high ionic conductivity. To further support this interpretation, we have included temperature-dependent EIS data and the derived kinetic parameters from impedance analysis in **Supplementary Fig. 16,17** and **Supplementary Table 20**. A detailed discussion of these results is provided in **Supplementary Text 6**.

- Line 234:** The Arrhenius plots (**Fig. 4a**) and Nyquist plots (**Supplementary Fig. 15,16**) of 0.8Li₂A–ZrCl₄ indicate enhanced Li⁺ ionic conductivity (σ_{Li^+}) of 1.78 mS cm⁻¹ (0.8Li₂O–ZrCl₄) and 1.01 mS cm⁻¹ (0.8Li₂S–ZrCl₄) at 25 ° with reduced activation energy (E_a) of 0.281 eV (0.8Li₂O–ZrCl₄) and 0.317 eV (0.8Li₂S–ZrCl₄) compared with those of Li₂ZrCl₆ (hcp: $\sigma_{Li^+} = 0.37$ mS cm⁻¹ at 25 °C with $E_a = 0.346$ eV and ccp: $\sigma_{Li^+} = 4.3 \times 10^{-3}$ mS cm⁻¹ at 25 °C with $E_a = 0.467$ eV). Also, EIS fitting of 0.8Li₂A–ZrCl₄ confirms that the observed ionic conduction originates from a single-phase component, corresponding to the oxygen substituted hcp phase and the sulfur substituted ccp phase, as previously confirmed by structural analysis (**Supplementary Fig. 17**). These findings are further corroborated by hopping rate-based conduction parameter analysis (**Supplementary Fig. 18 and Supplementary Table 20**), where we decoupled key descriptors governing Li⁺ transport.³⁹ Notably, both 0.8Li₂O–ZrCl₄ and 0.8Li₂S–ZrCl₄ exhibited significantly reduced migration energies and elevated hopping frequencies compared to both hcp- and ccp-type Li₂ZrCl₆. Moreover, the higher carrier concentration factor (C) and migration entropy (ΔS_m) of 0.8Li₂A–ZrCl₄ suggest enhanced configurational dynamics and a flatter energy landscape (**Supplementary Text 6**).
- Line 276:** Indeed, the probability density at 600 K shows a wider diffusion pathway and larger regions of Li probability density in the divalent anion substituted structure (**Fig. 4e,f**). A more noteworthy observation is that sulfur substitution leads to enhanced lithium-ion conduction along the ab plane, corresponding to the Li layers, compared to ccp-LZC (**Fig. 4f, dashed line**). In ccp framework, the lattice spacing of the ab plane significantly influences the activation of lithium-ion conduction.^{28,40} Given that the ccp-LZC exhibits limited Li⁺ transport along this plane, the lattice expansion and increased Li concentration induced by S²⁻ substitution allow the structure to reach a critical threshold necessary for enabling conduction in this previously suppressed direction (**Supplementary Fig. 24**). This highlights a key phenomenological role of sulfur substitution in facilitating multi-dimensional conduction pathways, which is experimentally supported by over two orders of magnitude increase in Li⁺ conductivity.

Revised Supplementary Information (Figure)

(Added in revision) Supplementary Fig. 16. Nyquist plots of EIS results for hcp-Li₂ZrCl₆, 0.8Li₂O–ZrCl₄, ccp-Li₂ZrCl₆ and 0.8Li₂S–ZrCl₄ measured at various temperatures ranging from –20 °C to 60 °C.

(Added in revision) Supplementary Fig. 17. Comparison of Nyquist plot with equivalent circuit fitting using single-phase and multi-component models for 0.8Li₂O–ZrCl₄ (top) and 0.8Li₂S–ZrCl₄ (bottom) at –20 °C.

(Added in revision) Supplementary Fig. 18. Arrhenius plot of hopping frequency, carrier concentration factor, migration entropy of hcp-Li₂ZrCl₆, 0.8Li₂O–ZrCl₄, ccp-Li₂ZrCl₆ and 0.8Li₂S–ZrCl₄.

(Added in revision) Supplementary Fig. 24. Li probability density at 600 K in ~300 ps (isosurface value $P = P_{max}/100$) of ccp-Li₂ZrCl₆ (top-left) and ccp-Li_{2.25}ZrCl_{5.75}S_{0.25} (top-right). Mean square displacement (MSD) of AIMD simulations at 600 K in ~300 ps of ccp-Li₂ZrCl₆ (down-left) and ccp-Li_{2.25}ZrCl_{5.75}S_{0.25} (down-right). Sulfur incorporation induces lattice expansion and structural distortion in the ccp framework, activating three-dimensional ionic conduction otherwise limited to the ab plane.

Revised Supplementary Information (Table)

(Added in revision) Supplementary Table 20. The calculated values related to ionic conductivity (pre-factor σ_0 , activation energy E_a , migration energy E_m , mobile carrier formation energy E_f , hopping frequency ν , carrier concentration factor C , migration entropy ΔS_m) of hcp-Li₂ZrCl₆, 0.8Li₂O–ZrCl₄, ccp-Li₂ZrCl₆ and 0.8Li₂S–ZrCl₄.

Revised Supplementary Information (Text)

(Added in revision) Supplementary Text 6. Ionic conduction properties

The enhanced ionic conductivity in $0.8\text{Li}_2\text{O}-\text{ZrCl}_4$ primarily originates from a significant reduction in migration energy rather than a substantial increase in mobile carrier concentration. Although the formation of mobile carriers becomes more favorable due to the lower formation energy, the overall carrier concentration remains comparable to hcp-LZC, likely due to partial immobilization of Li^+ near strongly coordinating O^{2-} sites. Instead, the key contributing factors are the increased hopping frequency and migration entropy, reflecting enhanced lattice distortion and dynamic disorder. This indicates a substantial distortion of the lattice and hopping site. In sulfur incorporation, the enhanced ionic conductivity in $0.8\text{Li}_2\text{S}-\text{ZrCl}_4$ stems from both a markedly reduced migration energy and a drastically lower mobile carrier formation energy compared to ccp-LZC. Unlike the oxygen-substituted system, sulfur substitution not only facilitates easier carrier formation but also significantly increases the number of mobile Li^+ ions participating in conduction. This is further supported by the substantial rise in hopping frequency, carrier concentration factor, and migration entropy, all of which indicate a pronounced enhancement in ion dynamics. These improvements collectively reflect a fundamental transformation of the ccp lattice, wherein sulfur substitution induces interlayer expansion and dynamic lattice flexibility, effectively activating a framework that is otherwise unfavorable for Li^+ migration.

7. I don't know what they perform HR-TEM on these samples which are known to be beam sensitive.

Response to comment 7: We thank the reviewer for raising this important point regarding the use of HRTEM on beam-sensitive halide-based samples. We found that HRTEM provided valuable complementary insights by enabling direct spatial visualization of how distinct crystalline domains, such as hcp and ccp phases or ZrO_2 inclusions, are distributed within the nanostructure. While techniques like XRD and PDF offer robust bulk-averaged structural information, they inherently lack spatial resolution. HRTEM imaging, even under constrained low-dose conditions, allowed us to assess whether these phases existed as separate domains or were intergrown. This spatial perspective offered additional support for our phase identification and helped contextualize the structural features.

Response Fig. 6. HRTEM image and FFT patterns of $0.8\text{Li}_2\text{O}-\text{ZrCl}_4$ and $0.8\text{Li}_2\text{S}-\text{ZrCl}_4$.

During our measurements, we also observed noticeable real-time degradation of the powder samples under high-energy electron beam exposure (300 keV), consistent with their known beam sensitivity.^{R13} To mitigate this issue, we performed imaging at a reduced accelerating voltage (160 keV), which inevitably

resulted in limited spatial resolution. To minimize beam-induced damage, we carefully controlled the imaging procedure by avoiding electron beam exposure until the final acquisition step. Specifically, the search and focus operations were conducted in separate regions to preserve the integrity of the area selected for imaging. Additionally, single-frame acquisition was used without any further alignment or prolonged exposure, further reducing the cumulative beam damage.

Due to the spatial resolution constraints under these low-dose conditions, Fast Fourier Transform (FFT) pattern identification may be less distinct. To address this, we have annotated representative d-spacing values and their corresponding hkl indices directly within the FFT images to facilitate clearer interpretation. These precautions allowed us to acquire structurally meaningful data while mitigating sample degradation. We will also clearly specify the accelerating voltage and measurement conditions used during HR-TEM measurements in the *Methods* section. Despite these limitations, we were able to obtain distinct FFT patterns corresponding to the hcp and ccp phases as well as ZrO₂ domains, which served our primary purpose of supporting the phase identification in conjunction with XRD and PDF analyses. We appreciate the reviewer's insight, which helped us refine the clarity and focus of our characterization strategy. To enhance the clarity and interpretability of the HRTEM results in **Fig. 2e** and **3e**, we have included indexing of the FFT patterns along with representative d-spacing values.

[Revision]

Revised Manuscript (Methods)

For the HRTEM measurements, the samples were loaded onto a lacey Cu grid in an Ar-filled glove box, and HRTEM images with FFT patterns were obtained using a JEM-ARM 300F NEOARM (JEOL). **Imaging was conducted at a reduced accelerating voltage of 160 keV, rather than the standard 300 keV, to mitigate degradation during measurement, and beam exposure was further minimized by performing search and focus procedures on separate regions and limiting the area of interest to a single-frame acquisition without additional alignment or prolonged exposure.**

Revised Figure

Fig. 2e HRTEM image with FFT patterns of 0.8Li₂O–ZrCl₄.

Fig. 3e HRTEM image with FFT patterns of 0.8Li₂S–ZrCl₄.

Reviewer #2

This manuscript reported the improvement of the ionic transport properties of halide based solid electrolytes by incorporating divalent anions such as oxygen and sulfur. Authors have used comprehensive characterization and computational techniques to understand the underlying mechanism for the improved transport property. The key takeaway from the work is that, while both oxygen and sulfur can help improve the ionic conductivity of halides, they work in a very different mechanism: oxygen promotes the formation of nanocomposite and sulfur drives the formation of cubic closed packed anion framework. The findings are quite novel, and the insights can help develop new halide based solid electrolytes. Below are some comments that need to be addressed before I can fully recommend its publication.

Response to comment: We are thankful to the reviewer for a thorough review of our manuscript and the valuable and constructive comments.

1. More clarifications are needed to explain the ionic transport mechanism of oxygen substituted halides. Authors stated that the substitution optimizes the interfacial oxygen incorporation which increased the ionic conductivity. The formation of space charge layer has been used as a way to improve the interfacial ionic conductivity of solid electrolytes. What is the difference between the current mechanism and enhancement of interfacial ionic transport by space charge layer? Many halide electrolytes, if not all, have exhibited improved ionic conductivity with the decrease in crystallinity due to the local order/disorder of cations in the lattice. Refs. 32 and 33 shows that amorphous Li-Zr-Cl-O has a very high ionic conductivity. Is that possible the increase in ionic conductivity is due to the formation of amorphous Li-Zr-Cl-O phase, rather than the stated mechanism? What is the exact role of ZrO₂ or Li₂O?

Response to comment 1: We appreciate the reviewer's insightful question about ionic transport mechanism of oxygen substituted structures.

1. Space charge layer effect

The core difference between our interfacial conduction mechanism and the conventional space charge layer (SCL) concept lies in the magnitude of the effect, the physical origin of the interface, and the mechanistic clarity. In previously reported systems utilizing space charge layer (SCL) effects (typically formed at the interface between two different materials due to lithium chemical potential mismatch) ionic conductivity enhancements have been modest, generally in the range of 10^{-6} to 10^{-4} S cm⁻¹ (e.g., LiF/silica, LiBH₄-LiI/Al₂O₃, and PAN-LiClO₄/LLTO nanowires)^{R14-18}. These systems often rely on ex-situ mixing of nanofillers, which not only increases processing complexity and cost, but also results in limited interfacial contact and phase modification due to poor dispersion or phase separation.

To best our knowledge, excluding Zr-based systems, no SCL-based interfacial system to date has achieved room-temperature ionic conductivity exceeding 1 mS cm⁻¹. In contrast to space charge layer effect, our nanocomposite systems are clearly attributed to the formation of an oxygen-substituted hcp phase at the interfaces between in situ formed ZrO₂, Li₂ZrCl₆, and residual Li₂O during the synthesis process. As a result, ionic conductivities are close to 2 mS cm⁻¹, which represents a substantial leap beyond prior SCL-based studies. These components create a large number of well-connected interfaces, which in turn induce framework modifications within the original Li₂ZrCl₆ lattice, ultimately enhancing ionic transport.

The formation of a high-conductivity interfacial phase in our system is not speculative. Through careful structural analysis based on synchrotron ($\lambda = 0.1665$ Å), we have clearly identified the formation of an oxygen-substituted hcp crystalline phase (Synchrotron XRD in **Fig. 1c**, PDF analysis in **Fig. 2a**, EXAFS data with Zr-Cl bond length elongation in **Fig. 2c** and **Supplementary Fig. 5**) with in-situ generated ZrO₂ nanograins (PDF analysis in **Fig. 2a**, EXAFS data with Zr-Zr peak broadening in **Fig. 2c** and **Supplementary Fig. 5**). This compositional and crystallographic distinct interphase allows us to unambiguously attribute enhanced conductivity to oxygen-incorporated lattice, rather than to electrostatic modulation as proposed in traditional SCL theories.

2. Cation disorder

We agree with the reviewer's observation that many halide electrolytes often exhibit the local cationic disorder within the lattice, particularly at the M2/M3 sites. In our system as well, such disorder can exist in lattice. To investigate its structural and ionic transport implications, we constructed computational models incorporating cationic disorder. Specifically, we generated two types of disordered configurations: (1) 1 side-disordered (1SD), where the cation disorder occurs asymmetrically near one side of the M2/M3 sites, and (2) Fully disordered (FD), where the M3 sites are entirely filled with metal cations, representing a high degree of disorder (**Supplementary Fig. 27**). These models allowed us to examine the potential influence of site-specific disorder on Li⁺ transport and lattice stability in a controlled and systematic method.

(Added in revision) Supplementary Fig. 27. Crystal structure depending on metal ordering in hcp structure ($\text{Li}_{2.5}\text{ZrCl}_{5.5}\text{O}_{0.5}$) and ccp structure ($\text{Li}_{2.25}\text{ZrCl}_{5.75}\text{S}_{0.25}$). The red circle shows a disordered metal site in lattice structure. 1 side-disorder (1SD) means cation disorder occurs asymmetrically near one side of the M2/M3 sites, and M3-site fully disordered (FD) indicates that the M3 sites are entirely filled with metal cations.

In response, we performed additional first-principles investigations using the same DFT parameters described in the *Methods* section. To evaluate the impact of cationic disorder on lattice structure and Li-ion transport, we generated disordered configurations by defining all possible Li sites and anionic substitution sites (3 of 36 for O^{2-} and 2 of 48 for S^{2-}) through enumeration of distinct configurations. The most energetically favorable models were then selected for further analysis. Lattice parameters and lattice volumes of disordered structures are presented in **Supplementary Table 24**.

Structure	Composition	a (Å)	b (Å)	c (Å)	α (°)	β (°)	γ (°)	Volume (Å ³)
1SD	$\text{Li}_{12}\text{Zr}_6\text{Cl}_{36}$	11.035	11.035	11.792	90	90	60	1243.554
hcp	$\text{Li}_{15}\text{Zr}_6\text{Cl}_{33}\text{O}_3$	10.988	10.888	12.210	90.282	89.957	60.831	1275.517
FD	$\text{Li}_{12}\text{Zr}_6\text{Cl}_{36}$	11.019	11.019	11.576	90	90	60	1217.247
hcp	$\text{Li}_{15}\text{Zr}_6\text{Cl}_{33}\text{O}_3$	11.082	10.849	11.8120	89.438	90.887	61.258	1245.564
1SD	$\text{Li}_{16}\text{Zr}_8\text{Cl}_{48}$	12.795	11.083	12.338	90.093	91.672	89.993	1748.906
ccp	$\text{Li}_{18}\text{Zr}_8\text{S}_2\text{Cl}_{46}$	12.793	11.095	12.387	89.642	90.80928	89.814	1758.048
FD	$\text{Li}_{16}\text{Zr}_8\text{Cl}_{48}$	12.744	11.055	11.995	90.014	90.80654	89.994	1689.794

ccp	$\text{Li}_{18}\text{Zr}_8\text{S}_2\text{Cl}_{46}$	12.783	11.011	12.093	90.003	90.97785	90.013	1701.934
-----	---	--------	--------	--------	--------	----------	--------	----------

(Added in revision) Supplementary Table 24. Lattice parameters and lattice volume of crystal structure of 1SD-hcp (Li_2ZrCl_6 and $\text{Li}_{2.5}\text{ZrCl}_{5.5}\text{O}_{0.5}$), FD-hcp (Li_2ZrCl_6 and $\text{Li}_{2.5}\text{ZrCl}_{5.5}\text{O}_{0.5}$), 1SD-ccp (Li_2ZrCl_6 and $\text{Li}_{2.25}\text{ZrCl}_{5.75}\text{S}_{0.25}$) and FD-ccp (Li_2ZrCl_6 and $\text{Li}_{2.25}\text{ZrCl}_{5.75}\text{S}_{0.25}$). 1 side-disorder (1SD) means cation disorder occurs asymmetrically near one side of the M2/M3 sites, and M3-site fully disordered (FD) indicates that the M3 sites are entirely filled with metal cations.

Response Fig. 7. Local structural motifs of clustered divalent anion-substituted polyhedra in 1SD- $\text{Li}_{2.5}\text{ZrCl}_{5.5}\text{O}_{0.5}$, FD- $\text{Li}_{2.5}\text{ZrCl}_{5.5}\text{O}_{0.5}$, 1SD- $\text{Li}_{2.25}\text{ZrCl}_{5.75}\text{S}_{0.25}$ and FD- $\text{Li}_{2.25}\text{ZrCl}_{5.75}\text{S}_{0.25}$.

Compared to the pristine bare ccp-LZC and hcp-LZC structures, all disordered structures exhibited lattice volume expansion, particularly along the c-axis. This can be attributed to the incorporation behavior of divalent anions, as shown in **Response Fig. 7**. Similar to the bare structure, the divalent anions are incorporated into the framework in a clustered form to effectively compensate for the charge of lithium, which serves as a fundamental driving force for structural expansion and distortion.

(Added in revision) Supplementary Fig. 28. Arrhenius plots obtained from ab initio molecular dynamics (AIMD) simulations of ordered-hcp (black), 1SD-hcp (green) and FD-hcp (purple) structures in Li_2ZrCl_6 and $\text{Li}_{2.5}\text{ZrCl}_{5.5}\text{O}_{0.5}$ compositions (left). Arrhenius plots of the AIMD simulation of ordered-ccp (black), 1SD-ccp (green) and FD-ccp (purple) in Li_2ZrCl_6 and $\text{Li}_{2.25}\text{ZrCl}_{5.75}\text{S}_{0.25}$ compositions (right).

Notably, despite the presence of cation disorder, the phenomenological effect of lattice modulation by divalent anions remains consistent, mirroring the structural characteristics beneficial to conduction that were observed in the ordered structures. To directly evaluate the impact of disorder on Li-ion mobility, we performed AIMD simulations for all disordered models, and in every case, the Li^+ diffusivity was found to be higher than that of the bare (M2 site fully occupied) counterparts (**Supplementary Fig. 29**).

(Added in revision) Supplementary Fig. 29. Distribution of continuous symmetry measured (CSM) value of Li octahedral site in 1SD-hcp (Li₂ZrCl₆ and Li_{2.5}ZrCl_{5.5}O_{0.5}), FD-hcp (Li₂ZrCl₆ and Li_{2.5}ZrCl_{5.5}O_{0.5}), 1SD-ccp (Li₂ZrCl₆ and Li_{2.25}ZrCl_{5.75}S_{0.25}) and FD-ccp (Li₂ZrCl₆ and Li_{2.25}ZrCl_{5.75}S_{0.25})

(Added in revision) Supplementary Fig. 30. COHP curves with averaged ICOHP values for Li-Cl bonds in 1SD-hcp (Li_2ZrCl_6 and $\text{Li}_{2.5}\text{ZrCl}_{5.5}\text{O}_{0.5}$), FD-hcp (Li_2ZrCl_6 and $\text{Li}_{2.5}\text{ZrCl}_{5.5}\text{O}_{0.5}$), 1SD-ccp (Li_2ZrCl_6 and $\text{Li}_{2.25}\text{ZrCl}_{5.75}\text{S}_{0.25}$) and FD-ccp (Li_2ZrCl_6 and $\text{Li}_{2.25}\text{ZrCl}_{5.75}\text{S}_{0.25}$).

(Added in revision) Supplementary Fig. 31. Histograms for distribution of ICOHP values for Li-Cl bonds in 1SD-hcp (Li_2ZrCl_6 and $\text{Li}_{2.5}\text{ZrCl}_{5.5}\text{O}_{0.5}$), FD-hcp (Li_2ZrCl_6 and $\text{Li}_{2.5}\text{ZrCl}_{5.5}\text{O}_{0.5}$), 1SD-ccp (Li_2ZrCl_6 and $\text{Li}_{2.25}\text{ZrCl}_{5.75}\text{S}_{0.25}$) and FD-ccp (Li_2ZrCl_6 and $\text{Li}_{2.25}\text{ZrCl}_{5.75}\text{S}_{0.25}$)

This trend is consistent across both O²⁻ and S²⁻ substituted ordered configurations and can be attributed to increased electrostatic repulsion from the divalent anions, like the expansion observed in the ordered substitutional models. To further assess the local Li site environment, we analyzed the continuous symmetry measure (CSM) values and found that the Li sites in disordered structures experienced greater distortion (**Supplementary Fig. 29**), supporting the idea that disorder promotes channel widening and diversification of Li–Cl bonding, both of which are conducive to fast ionic transport. (**Supplementary Fig. 30,31**)

Notably, despite the presence of cation disorder, the phenomenological effect of lattice modulation by divalent anions remains consistent, mirroring the structural characteristics beneficial to conduction that were observed in the ordered structures. To directly evaluate the impact of disorder on Li-ion mobility, we performed AIMD simulations for all disordered models, and in every case, the Li⁺ diffusivity was found to be higher than that of the bare (M2 site fully occupied) counterparts.

Regardless of M2/M3 cationic site disorder, the same underlying structural principles (lattice expansion, increased site distortion, weakened Li–Cl coordination and energy landscape flattening) are active and beneficial for enhancing ionic conductivity in divalent anion substituted structures. Accordingly, we have added the corresponding results and extended the discussion in the revised manuscript (**Supplementary Fig. 27-31**). Additionally, during the phase fraction analysis based on PDF fitting, we carefully considered Zr-site disorder and implemented the most appropriate disorder ratio (about 50% in hcp and about 15% in ccp) in the structural models to achieve reliable fitting and accurate phase quantification (**Supplementary Table 1-14**). We appreciate the reviewer's comment, which prompted us to explore this aspect more thoroughly.

3. Amorphous phase

Although the synthesized phase may vary depending on experimental conditions, accurate phase identification and a clear elucidation of how the observed phase contributes to enhanced material properties are essential. Li-Zr-Cl-O compositions reported in *References* 32 and 33 are interpreted as have high ionic conductivity primarily due to their high amorphous content. We also observed broad, featureless patterns in our samples when measured using conventional lab-based XRD ($\lambda = 1.54 \text{ \AA}$), particularly in low-Li₂O compositions such as 0.8Li₂O–ZrCl₄, which could give the impression of an amorphous structure (**Fig. 1a**). However, such patterns alone are insufficient to definitively determine the presence of an amorphous phase.

So, to differentiate whether the enhanced conductivity in our system originates from true amorphization or rather from nanoscale crystalline features and interfacial mechanisms, we employed high-energy synchrotron XRD ($\lambda = 0.1665 \text{ \AA}$). This technique offers significantly enhanced resolution and sensitivity toward detecting short-range order. While lab-based XRD suggested poorly resolved patterns, high-energy synchrotron measurements clearly revealed the dominant presence of a hexagonal close-packed (hcp) phase in 0.8Li₂O–ZrCl₄. Although variations in synthesis conditions may indeed lead to more disordered structures or partial amorphization, our results indicate that in our experimental setting, the primary phase is crystalline.

This highlights the importance of high-intensity and short-wavelength beam source when evaluating materials with potentially nanocrystalline domains, as small crystalline grain signals may be undetectable due to the peak broadening governed by the Scherrer equation ($\tau = \frac{K\lambda}{\beta \cos\theta}$, where τ : detectable grain size, K : shape factor, λ : wavelength, β : line broadening at FWHM, θ : Bragg angle).

Additionally, to assess the presence and structural impact of amorphous phase in our system, we performed detailed pair distribution function (PDF) analyses and examined both local and intermediate-range orders. The results collectively indicate that while short-range ordered ZrO₂ and ZrS₂ phases account for approximately 30% of the structure, amorphization within the conductive hcp and ccp frameworks is not observed (**Supplementary Table 4,9** and **Supplementary Table 3,4,10,11**).

First, no attenuation of the peak corresponding to hcp and ccp phases was observed within the PDF $G(r)$ range of 1.5-6 \AA for the compositions $x\text{Li}_2\text{O}-\text{ZrCl}_4$ and $x\text{Li}_2\text{S}-\text{ZrCl}_4$ ($x = 0.6, 0.8, 1.0$) (**Fig. 2a, 3a**). Given that $\sim 6 \text{ \AA}$ corresponds approximately to the lattice parameter (interlayer distance) along the c -axis of

Li_2ZrCl_6 (6.030 Å for hcp and 6.376 Å for ccp), attenuation of this peak would indicate disruption of long-range periodicity associated with structural amorphization. The absence of such attenuation in this region thus suggests the preservation of both hcp and ccp structural motifs without substantial amorphous phase formation. Thus, we conclude that there is a negligible amount of amorphous phase formation in these materials at compositions corresponding to the highest ionic conductivity ($x = 0.6, 0.8, 1.0$).

Furthermore, the ionic conductivity trends across compositions support this interpretation. Samples exhibiting more pronounced structural disorder or lower crystallinity (such as $1.0\text{Li}_2\text{O}-\text{ZrCl}_4$ and $1.2\text{Li}_2\text{O}-\text{ZrCl}_4$) showed reduced conductivity compared to more crystalline compositions like $0.8\text{Li}_2\text{O}-\text{ZrCl}_4$ (**Fig. 1b,c**). Therefore, while partial amorphous phases may exist, they are unlikely to be the dominant contributors to the enhanced ionic transport in our system.

Taken together, although the presence of amorphous domains cannot be completely excluded depending on synthesis conditions, our combined XRD, PDF, and conductivity analyses suggest that the high ionic conductivity observed in our system does not arise from amorphization. Instead, it correlates strongly with the presence of a structurally and compositionally optimized crystalline hcp phase. We attribute this improved performance to a well-defined oxygen-substituted lattice that facilitates Li^+ conduction through lattice expansion, Li-site distortion, and diversified Li-Cl interactions mechanisms supported by both experimental characterization and first-principles simulations.

4. Role of oxygen sources (ZrO_2 and Li_2O)

ZrO_2 acts as a structural driver for interfacial framework modulation. During mechanochemical processing, nanoscale ZrO_2 domains are formed in situ and create a large number of well-connected interfaces with Li_2ZrCl_6 . These interfaces facilitate the formation of a new oxygen-substituted crystalline phase, which exhibits enhanced Li^+ mobility. To investigate whether such interfacial modulation by ZrO_2 is thermodynamically driven, we constructed interface models and examined the spontaneous anion substitution behavior at the $\text{ZrO}_2-\text{Li}_2\text{ZrCl}_6$ interface by DFT calculations (**Response Fig 8**). Specifically, we evaluated all Miller index surfaces with absolute values ≤ 1 and selected three low-surface-energy planes for each material (Li_2ZrCl_6 : 100, 011, 101 and ZrO_2 : 100, 010, 111). Among these, the interface with the lowest work of adhesion ($W_a = \gamma_{\text{LZC}+\text{ZrO}_2} - \gamma_{\text{LZC}} - \gamma_{\text{ZrO}_2}$), where γ is surface energy) was identified as the most stable interfacial model and used for further modeling.

Response Fig. 8. Interfacial slab model construction of Li_2ZrCl_6 and ZrO_2

Response Fig. 9. DFT calculated structural energy and thermodynamic driving force of interfacial anion substitution between Li_2ZrCl_6 and ZrO_2

We confirmed that the anion exchange at the interface is thermodynamically favorable, indicating a strong spontaneous driving force for oxygen incorporation into the halide framework (**Response Fig. 9**). This suggests that oxygen substitution is sufficiently accessible within the hcp structure at the interface with ZrO_2 . Furthermore, STEM-EELS (**Response Fig. 1**) mapping provided direct experimental evidence that oxygen-containing species are indeed distributed along the interface with a compositional gradient in oxygen content, supporting the occurrence of interphase formation driven by ZrO_2 .

Response Fig. 1. STEM image and EELS (Li, Zr, Cl, O, S) elemental mapping (Li, Zr, Cl, O, S) of $0.8\text{Li}_2\text{O}-\text{ZrCl}_4$ and $0.8\text{Li}_2\text{S}-\text{ZrCl}_4$.

Additionally, Li_2O (as a binary compound) can serve as a source of both Li and O within the hcp structure. Its role can be discussed more concretely through the comparative analysis provided in **Supplementary Table 18**, where experimental ionic conductivities are compared with the corresponding reaction equations of various compositions. Among them, the composition containing residual Li_2O ($0.8\text{Li}_2\text{O}-\text{ZrCl}_4$) exhibits significantly higher ionic conductivity than systems involving ZrO_2 alone or combinations such as $\text{LiCl}-\text{ZrO}_2-\text{Li}_2\text{ZrCl}_6$. This trend strongly suggests that Li_2O also contributes to interfacial lattice modulation, much like ZrO_2 , facilitating the formation of oxygen-substituted hcp phases.

The enhanced conductivity observed in our system (1.8 mS cm^{-1}) compared to previously reported $\text{ZrO}_2-2\text{LiCl}-\text{Li}_2\text{ZrCl}_6$ systems³¹ (1.3 mS cm^{-1}) may stem from this effect. Thus, in the $0.8\text{Li}_2\text{O}-\text{ZrCl}_4$ composition, residual Li_2O likely remains embedded at the interface, where it creates a local oxygen concentration gradient that promotes interfacial oxygen substitution and stabilizes the conductive hcp lattice. This suggests the dual role of Li_2O in both supplying Li^+ and enabling favorable oxygen-driven interfacial hcp phase formation.

5. Advanced discussion about ionic conduction mechanisms

To provide a more comprehensive interpretation of our impedance data, we have adopted the analytical framework proposed by Li et al.^{R8} which enables the decoupling of critical parameters governing ionic conductivity in solid-state electrolytes. Following this approach, we systematically extracted and analyzed the following key parameters from our experimental data as presented in **Supplementary Fig. 18** and **Supplementary Table 20**.

(Added in revision) Supplementary Fig. 18. Arrhenius plot of hopping frequency, carrier concentration factor, migration entropy of hcp-Li₂ZrCl₆, 0.8Li₂O–ZrCl₄, ccp-Li₂ZrCl₆ and 0.8Li₂S–ZrCl₄.

(Added in revision) Supplementary Table 20. The calculated values related to ionic conductivity (pre-factor σ_0 , activation energy E_a , migration energy E_m , mobile carrier formation energy E_f , hopping frequency ν , carrier concentration factor C , migration entropy ΔS_m) of hcp-Li₂ZrCl₆, 0.8Li₂O–ZrCl₄, ccp-Li₂ZrCl₆ and 0.8Li₂S–ZrCl₄.

	hcp-Li ₂ ZrCl ₆	0.8Li ₂ O–ZrCl ₄	ccp-Li ₂ ZrCl ₆	0.8Li ₂ S–ZrCl ₄
σ (25 °C)	3.7×10^{-4} S cm ⁻¹	1.78×10^{-3} S cm ⁻¹	4.3×10^{-6} S cm ⁻¹	1.01×10^{-3} S cm ⁻¹
σ_0	77848 S cm ⁻¹ K ⁻¹	29836 S cm ⁻¹ K ⁻¹	99451 S cm ⁻¹ K ⁻¹	69324 S cm ⁻¹ K ⁻¹
E_a	346.39 meV	280.96 meV	466.75 meV	317.22 meV
E_m	333.11 meV	276.03 meV	445.53 meV	315.04 meV
E_f	13.28 meV	4.93 meV	21.22 meV	2.18 meV
ν	1.28×10^6 Hz at 0 °C	5.17×10^6 Hz at 0 °C	4.30×10^4 Hz at 0 °C	3.14×10^6 Hz at 0 °C
C	4.36×10^{-8} S cm ⁻¹ Hz ⁻¹ K	4.60×10^{-8} S cm ⁻¹ Hz ⁻¹ K	1.15×10^{-8} S cm ⁻¹ Hz ⁻¹ K	2.41×10^{-8} S cm ⁻¹ Hz ⁻¹ K
ΔS_m	1.56×10^{-4} eV K ⁻¹	2.46×10^{-4} eV K ⁻¹	4.42×10^{-5} eV K ⁻¹	1.38×10^{-4} eV K ⁻¹

In the case of 0.8Li₂O–ZrCl₄, the migration energy ($E_m = 276.03$ meV) is notably lower than that of hcp-LZC ($E_m = 333.11$ meV), indicating a reduction in the energy barrier for Li⁺ migration. In addition, the mobile carrier formation energy ($E_f = 4.93$ meV) is smaller than that of hcp-LZC ($E_f = 13.28$ meV), suggesting that Li⁺ carriers are more readily formed in the O-substituted structure. However, the carrier concentration factor in the O-substituted structure ($C = 4.60 \times 10^{-8}$ S cm⁻¹ Hz⁻¹ K⁻¹) remains comparable to that of hcp-LZC ($C = 4.36 \times 10^{-8}$ S cm⁻¹ Hz⁻¹ K⁻¹), with only a slight increase, this marginal change suggests that the total number of detectable mobile carriers is not drastically altered. Indeed, this implies that although the total number of Li⁺ ions may have increased due to O²⁻ substitution and associated charge compensation, a slight fraction of these additional Li⁺ ions are situated in strongly coordinating environments near oxygen. These Li–O coordinated species experience deeper local potential wells and reduced mobility compared to Li⁺ coordinated by Cl⁻, effectively making them non-detectable on the timescale probed by conductivity measurements. That is, although the formation of mobile carriers has become more favorable (lower E_f), the number of lithium ions participating in conduction remains comparable to that of hcp-LZC or is only slightly

increased (comparable C).

While the amount of lithium participating in long-range transport does not substantially differ, the significant increase in both hopping frequency and migration entropy indicates enhanced lattice distortion within the structure. This structural perturbation is further supported by the Raman spectra (**Fig. 2d**), which displays slightly broader features compared to hcp-LZC, reflecting increased dynamic disorder. It is evident that the observed improvement in ionic conductivity is primarily driven by structural regulation effects, particularly the reduction in migration energy. These energetic advantages are accompanied by a higher hopping frequency ($\nu = 5.17 \times 10^6$ Hz at 0 °C) and greater migration entropy ($\Delta S_m = 2.46 \times 10^{-4}$ eV K⁻¹), both of which contribute to faster and more thermodynamically favorable ion migration. These findings collectively demonstrate that oxygen incorporation fundamentally reshapes the local lattice environment by introducing significant structural distortion, particularly at Li sites. This distortion lowers the migration barrier and enhances dynamic lattice fluctuations, ultimately serving as the primary driver for the observed improvement in ionic conductivity. Thus, the enhanced transport properties in the O-substituted system arise not from an increase in mobile carrier concentration, but from structurally induced distortions that boost ion mobility.

In case of sulfur, migration energy in 0.8Li₂S–ZrCl₄ ($E_m = 315.04$ meV) is significantly lower than that of ccp-LZC ($E_m = 445.53$ meV), indicating that S²⁻ substitution significantly flattens the energy landscape for Li⁺ migration. More notably, the formation energy of mobile carriers is drastically reduced by an order of magnitude (2.18 meV vs. 21.22 meV), which strongly suggests a fundamentally altered environment for carrier generation compared to oxygen-substituted systems. This reduction is especially meaningful in the context of ccp-type Zr-based frameworks, where a small cationic radius typically results in extremely low Li-ion conductivity^{R9}; thus, the observed low E_f implies not only the formation of mobile Li ions in such typically inactive lattices, but also a fundamental modification of the lattice framework that enables and stabilizes their presence. Furthermore, the hopping frequency increases by nearly two orders of magnitude ($\sim 10^6$ Hz vs. $\sim 10^4$ Hz), directly contributing to the enhanced dynamic behavior of Li ions. This is accompanied by a more than two-fold increase in the carrier concentration factor ($C = 2.41 \times 10^{-8}$ S cm⁻¹ Hz K⁻¹), highlighting the synergistic effect of faster ion dynamics.

In contrast to the oxygen-substituted structure, where the carrier concentration factor remains nearly unchanged, the more than two-fold increase observed here highlights not only the reduced energy required for mobile carrier formation, but also the substantially increased number of carriers actively participating in conduction. Crucially, the migration entropy (ΔS_m) also increases significantly, by approximately 3–4 times (1.38×10^{-4} eV K⁻¹), reflecting increased vibrational freedom and dynamic lattice flexibility. Collectively, these results indicate that the lattice itself undergoes a profound transformation, fundamentally altering the characteristics of ionic transport within the structure. This dynamic lattice behavior is closely linked to the structural changes induced by sulfur substitution. In the case of LZC, the small ionic radius of Zr⁴⁺ leads to the formation of a ccp-type framework, within which Li⁺ migration along the ab-plane is inherently sluggish due to tight interlayer spacing and the absence of structural pillars in ab plane of Li layer.^{R10-12} To overcome this, we introduced S²⁻ into the framework, where the increased repulsion between sulfur atoms leads to an expansion of the interlayer distance.

(Added in revision) Supplementary Fig. 24. Li probability density at 600 K in ~ 300 ps (isosurface value $P = P_{\max}/100$) of $\text{ccp-Li}_2\text{ZrCl}_6$ (top-left) and $\text{ccp-Li}_{2.25}\text{ZrCl}_{5.75}\text{S}_{0.25}$ (top-right). Mean square displacement (MSD) of AIMD simulations at 600 K in ~ 300 ps of $\text{ccp-Li}_2\text{ZrCl}_6$ (down-left) and $\text{ccp-Li}_{2.25}\text{ZrCl}_{5.75}\text{S}_{0.25}$ (down-right). Sulfur incorporation induces lattice expansion and structural distortion in the ccp framework, activating three-dimensional ionic conduction otherwise limited to the ab plane.

This structural relaxation facilitates Li^+ migration along the ab-plane, as clearly evidenced by the enhanced in-plane mean square displacement (MSD) profiles (**Supplementary Fig. 24**). Since ab-plane transport dominates ionic conductivity in the ccp-type LZC structure, this targeted expansion directly contributes to the significant enhancement in overall ionic conductivity. Collectively, the marked reductions in both E_m and E_f , along with enhanced hopping dynamics and entropy-driven mobility, serve as key factors in transforming a conduction inactive phase ($\sim 10^{-6} \text{ S cm}^{-1}$) into a highly conductive one ($\sim 10^{-3} \text{ S cm}^{-1}$). These findings highlight the pivotal role of lattice dynamics and structural regulations in enabling high ionic conductivity. To further support this interpretation, we have included temperature-dependent EIS data and the derived kinetic parameters from impedance analysis in **Supplementary Fig. 18** and **Supplementary Table 20**. A detailed discussion of these results is provided in **Supplementary Text 6**.

[Revision]

Revised Manuscript

- **Line 77:** However, the challenge lies in the unclear understanding of divalent anion incorporation in Zr-based halide SEs, with ongoing debates on whether the synthesized compounds exist in an amorphous state, adopt a ccp structure, or form a nanocrystalline phase.^{34,35} **Although the synthesized phase may vary depending on experimental conditions, accurate phase identification**

and a clear elucidation of how the observed phase contributes to enhanced material properties are essential.

- **Line 242:** These findings are further corroborated by hopping rate-based conduction parameter analysis (**Supplementary Fig. 18 and Supplementary Table 20**), where we decoupled key descriptors governing Li^+ transport.³⁹ Notably, both $0.8\text{Li}_2\text{O}-\text{ZrCl}_4$ and $0.8\text{Li}_2\text{S}-\text{ZrCl}_4$ exhibited significantly reduced migration energies and elevated hopping frequencies compared to both hcp- and ccp-type Li_2ZrCl_6 . Moreover, the higher carrier concentration factor (C) and migration entropy (ΔS_m) of $0.8\text{Li}_2\text{A}-\text{ZrCl}_4$ suggest enhanced configurational dynamics and a flatter energy landscape (**Supplementary Text 6**).
- **Line 276:** Indeed, the probability density at 600 K shows a wider diffusion pathway and larger regions of Li probability density in the divalent anion substituted structure (**Fig. 4e,f**). A more noteworthy observation is that sulfur substitution leads to enhanced lithium-ion conduction along the ab plane, corresponding to the Li layers, compared to ccp-LZC (**Fig. 4f, dashed line**). In ccp framework, the lattice spacing of the ab plane significantly influences the activation of lithium-ion conduction.^{28,40} Given that the ccp-LZC exhibits limited Li^+ transport along this plane, the lattice expansion and increased Li concentration induced by S^{2-} substitution allow the structure to reach a critical threshold necessary for enabling conduction in this previously suppressed direction (**Supplementary Fig. 24**). This highlights a key phenomenological role of sulfur substitution in facilitating multi-dimensional conduction pathways, which is experimentally supported by over two orders of magnitude increase in Li^+ conductivity.
- **Line 303:** Comprehensively, the fundamental cause of the enhanced lithium-ion conduction is that the structural variation induces flattening of the energy landscape in site-to-site conduction for Li^+ migration.⁴² This effect stems from the incorporation of clustered divalent anions, which modulate lattice framework by widening conduction channels, weakening Li-Cl interactions, and diversifying the energy states of Li sites. Both divalent anion incorporation strategies retain their conductivity-enhancing effects despite of intrinsic cationic disorder (**Supplementary Fig. 27-31 and Supplementary Table 24**).

Revised Supplementary Information (Figure)

(Added in revision) Supplementary Fig. 18. Arrhenius plot of hopping frequency, carrier concentration factor, migration entropy of hcp- Li_2ZrCl_6 , $0.8\text{Li}_2\text{O}-\text{ZrCl}_4$, ccp- Li_2ZrCl_6 and $0.8\text{Li}_2\text{S}-\text{ZrCl}_4$.

(Added in revision) Supplementary Fig. 24. Li probability density at 600 K in ~ 300 ps (isosurface value $P = P_{\text{max}}/100$) of ccp- Li_2ZrCl_6 (top-left) and ccp- $\text{Li}_{2.25}\text{ZrCl}_{5.75}\text{S}_{0.25}$ (top-right). Mean square displacement (MSD) of AIMD simulations at 600 K in ~ 300 ps of ccp- Li_2ZrCl_6 (down-left) and ccp- $\text{Li}_{2.25}\text{ZrCl}_{5.75}\text{S}_{0.25}$ (down-right). Sulfur incorporation induces lattice expansion and structural distortion in the ccp framework, activating three-dimensional ionic conduction otherwise limited to the ab plane.

(Added in revision) Supplementary Fig. 27. Crystal structure depending on metal ordering in hcp structure ($\text{Li}_{2.5}\text{ZrCl}_{5.5}\text{O}_{0.5}$) and ccp structure ($\text{Li}_{2.25}\text{ZrCl}_{5.75}\text{S}_{0.25}$). The red circle shows a disordered metal site in lattice structure. 1 side-disorder (1SD) means cation disorder occurs asymmetrically near one side of the M2/M3 sites, and M3-site fully disordered (FD) indicates that the M3 sites are entirely filled with metal cations.

(Added in revision) Supplementary Fig. 28. Arrhenius plots obtained from ab initio molecular dynamics (AIMD) simulations of ordered-hcp (black), 1SD-hcp (green) and FD-hcp (purple) structures in Li_2ZrCl_6 and $\text{Li}_{2.5}\text{ZrCl}_{5.5}\text{O}_{0.5}$ compositions (left). Arrhenius plots of the AIMD simulation of ordered-ccp (black), 1SD-ccp (green) and FD-ccp (purple) in Li_2ZrCl_6 and $\text{Li}_{2.25}\text{ZrCl}_{5.75}\text{S}_{0.25}$ compositions (right).

(Added in revision) Supplementary Fig. 29. Distribution of continuous symmetry measured (CSM) value of Li octahedral site in 1SD-hcp (Li_2ZrCl_6 and $\text{Li}_{2.5}\text{ZrCl}_{5.5}\text{O}_{0.5}$), FD-hcp (Li_2ZrCl_6 and $\text{Li}_{2.5}\text{ZrCl}_{5.5}\text{O}_{0.5}$), 1SD-ccp (Li_2ZrCl_6 and $\text{Li}_{2.25}\text{ZrCl}_{5.75}\text{S}_{0.25}$) and FD-ccp (Li_2ZrCl_6 and $\text{Li}_{2.25}\text{ZrCl}_{5.75}\text{S}_{0.25}$)

(Added in revision) Supplementary Fig. 30. COHP curves with averaged ICOHP values for Li-Cl bonds in 1SD-hcp (Li_2ZrCl_6 and $\text{Li}_{2.5}\text{ZrCl}_{5.5}\text{O}_{0.5}$), FD-hcp (Li_2ZrCl_6 and $\text{Li}_{2.5}\text{ZrCl}_{5.5}\text{O}_{0.5}$), 1SD-ccp (Li_2ZrCl_6 and $\text{Li}_{2.25}\text{ZrCl}_{5.75}\text{S}_{0.25}$) and FD-ccp (Li_2ZrCl_6 and $\text{Li}_{2.25}\text{ZrCl}_{5.75}\text{S}_{0.25}$).

(Added in revision) Supplementary Fig. 31. Histograms for distribution of ICOHP values for Li-Cl bonds in 1SD-hcp (Li_2ZrCl_6 and $\text{Li}_{2.5}\text{ZrCl}_{5.5}\text{O}_{0.5}$), FD-hcp (Li_2ZrCl_6 and $\text{Li}_{2.5}\text{ZrCl}_{5.5}\text{O}_{0.5}$), 1SD-ccp (Li_2ZrCl_6 and $\text{Li}_{2.25}\text{ZrCl}_{5.75}\text{S}_{0.25}$) and FD-ccp (Li_2ZrCl_6 and $\text{Li}_{2.25}\text{ZrCl}_{5.75}\text{S}_{0.25}$)

Revised Supplementary Information (Table)

(Added in revision) Supplementary Table 20. The calculated values related to ionic conductivity (pre-factor σ_0 , activation energy E_a , migration energy E_m , mobile carrier formation energy E_f , hopping frequency ν , carrier concentration factor C , migration entropy ΔS_m) of hcp- Li_2ZrCl_6 , $0.8\text{Li}_2\text{O}-\text{ZrCl}_4$, ccp- Li_2ZrCl_6 and $0.8\text{Li}_2\text{S}-\text{ZrCl}_4$.

Revised Supplementary Information (Text)

(Added in revision) Supplementary text 3. Amorphous phase

The peaks corresponding to the hcp and ccp phases were well maintained within the PDF $G(r)$ range of 1.5–6 Å for the compositions $x\text{Li}_2\text{O}-\text{ZrCl}_4$ and $x\text{Li}_2\text{S}-\text{ZrCl}_4$ ($x = 0.6, 0.8, 1.0$) (**Fig. 2a, 3a**). Given that ~6 Å corresponds approximately to the lattice parameter (interlayer distance) along the c-axis of Li_2ZrCl_6 (6.030 Å for hcp and 6.376 Å for ccp), the sustained intensity of this peak suggests that the long-range periodicity associated with these lattice frameworks is preserved. The consistent peak behavior in this region thus supports the retention of hcp and ccp structural motifs without meaningful evidence of amorphous phase development. Thus, we conclude that there is a negligible amount of amorphous phase formation in these materials at compositions corresponding to the highest ionic conductivity ($x = 0.6, 0.8, 1.0$). While structural amorphization in Zr-based oxyhalide has been proposed in prior studies as a mechanism for enhancing ionic conductivity, particularly within each respective systems, our compositions and results suggest a different origin for the observed conductivity. Samples with lower crystallinity (such as $1.0\text{Li}_2\text{O}-\text{ZrCl}_4$ and $1.2\text{Li}_2\text{O}-\text{ZrCl}_4$) exhibited reduced ionic conductivity compared to the more crystalline $0.8\text{Li}_2\text{O}-\text{ZrCl}_4$ composition (**Fig. 1b,c**). This behavior shows that amorphization was not linked to improved ionic transport, suggesting instead that the high ionic conductivity in our system originates from an optimized crystalline structure rather than from any amorphous contribution. Thus, we conclude that there is only negligible amount of amorphous phase in these materials at compositions corresponding to the highest ionic conductivity ($x = 0.6, 0.8, 1.0$).

(Added in revision) Supplementary Text 6. Ionic conduction properties

The enhanced ionic conductivity in $0.8\text{Li}_2\text{O}-\text{ZrCl}_4$ primarily originates from a significant reduction in migration energy rather than a substantial increase in mobile carrier concentration. Although the formation of mobile carriers becomes more favorable due to the lower formation energy, the overall carrier concentration remains comparable to hcp-LZC, likely due to partial immobilization of Li^+ near strongly coordinating O^{2-} sites. Instead, the key contributing factors are the increased hopping frequency and migration entropy, reflecting enhanced lattice distortion and dynamic disorder. This indicates a substantial distortion of the lattice and hopping site. In sulfur incorporation, the enhanced ionic conductivity in $0.8\text{Li}_2\text{S}-\text{ZrCl}_4$ stems from both a markedly reduced migration energy and a drastically lower mobile carrier formation energy compared to ccp-LZC. Unlike the oxygen-substituted system, sulfur substitution not only facilitates easier carrier formation but also significantly increases the number of mobile Li^+ ions participating in conduction. This is further supported by the substantial rise in hopping frequency, carrier concentration factor, and migration entropy, all of which indicate a pronounced enhancement in ion dynamics. These

improvements collectively reflect a fundamental transformation of the ccp lattice, wherein sulfur substitution induces interlayer expansion and dynamic lattice flexibility, effectively activating a framework that is otherwise unfavorable for Li⁺ migration.

2. Body centered anion framework has been proposed by Ceder et al, as a basic structural unit for superionic sulfide based solid electrolytes. Authors mentioned sulfur-incorporated halide also benefit from the formation of cubic closed packed anion framework. There seems to be some consistency here and warrants more discussion comparing the ionic transport mechanism between sulfides and halides.

Response to comment 2: We appreciate the reviewer's insightful comment highlighting the connection between sulfide and halide-based superionic conductors in terms of anion framework geometry and ionic transport. The work by Ceder et al.^{R19} proposed that body-centered or closely packed anion frameworks in sulfide-based electrolytes are key for facilitating efficient Li⁺ conduction pathways. Their work also emphasizes that increasing the lattice volume per anion helps reduce the activation barrier for ion migration by flattening the energy landscape. Also, they show insights to cubic close-packed (ccp [fcc]) and hexagonal close-packed (hcp) anion frameworks need increase of lattice volume per anion for relieving the activation energy for Li-ion migration as shown in **Response Fig. 10**.

Response Fig. 10. Li-ion migration pathways in bcc/fcc/hcp-type lattices and activation barrier for Li-ion migration versus lattice volume from ref. R19

The suggested idea can be directly applicable to halide systems as well. We observe that an increase in lattice volume per anion leads to a reduction in the activation barrier (**Fig. 4a-c** and **Fig. 5a**), which parallels the mechanism observed in sulfide-based systems. Our strategy that introduces divalent anions with the same anion counts while increasing electrostatic repulsion within the lattice effectively expands the lattice volume. This demonstrates that the fundamental design principle observed in sulfide systems can also be applied to halide-based materials, providing an insight for enhancing ionic conductivity through anion framework modulation.

In such sulfide ccp (fcc) frameworks, Li⁺ diffusion follows dominant T–O–T (tetrahedral–octahedral–tetrahedral) hopping pathways. At larger lattice volumes, octahedral sites become more unstable, reducing site energy differences and thus promoting lower migration barriers. As the reviewer correctly pointed out, S incorporation into halide frameworks allows the structural regulations of the ccp lattice, analogous to the superionic sulfide structures.

In halide ccp frameworks, unlike sulfide-based systems, Li⁺ ions predominantly occupy energetically favorable octahedral sites. As a result, long-range Li-ion migration typically proceeds through

tetrahedral intermediates, following an octahedral–tetrahedral–octahedral (O–T–O) pathway.²⁸ Nevertheless, by expanding the lattice volume, the relative stability of octahedral sites and transition tetrahedral site can be reduced, thereby decreasing the migration barrier and effectively enabling faster Li-ion conduction.

Our sulfur-substituted halide compositions exhibit the same behavior: lattice expansion induced by S substitution flattens the energy landscape and reduces the migration barrier, even though the chemical nature of the anion differs. This clearly shows that the phenomenological design principle proposed for sulfides (modulating the anion framework to enable facile Li⁺ transport) can be extended to halide-based materials through the incorporation of divalent anion itself.

Response Fig. 11. Distortion of the lithium environment leading to the destabilization of lithium sites and reducing the energy gap to the transition state (left) with Calculated lithium-ion kinetically resolved activation energies (E_{KRA}) in different CSM value (right) from ref. R20

A subsequent study from the same group^{R20} proposed that, in oxide-based electrolytes, frameworks composed of corner-sharing polyhedra can enhance Li-ion conduction. A key mechanism was identified: structural distortion at the Li site raises its local energy, which lowers the activation barrier for Li-ion migration. In our system, such Li site distortion is similarly observed and quantified through CSM analysis (**Fig. 5b**).

The incorporation of heterogeneous divalent anions (e.g., O or S) in halide lattice introduces asymmetry in the local coordination as shown in **Response Fig. 11**, thereby destabilizing Li sites and promoting faster diffusion. This suggests that structural feature driven Li conduction enhancement strategies proven in oxide systems can be effectively extended to halide systems via hetero-anion substitution. This clearly shows that the phenomenological design principle proposed for oxides (modulating Li coordination environment by distortion) can be extended to halide-based materials through the incorporation of oxygen and sulfur anion itself.

(Added in revision) Fig. 5b CSM value of hcp- Li_2ZrCl_6 , hcp- $\text{Li}_{2.5}\text{ZrCl}_{5.5}\text{O}_{0.5}$, ccp- Li_2ZrCl_6 and ccp- $\text{Li}_{2.25}\text{ZrCl}_{5.75}\text{S}_{0.25}$. A minimum of 0 corresponds to a perfectly symmetric coordination environment and the maximum of 66.7 corresponds to infinite elongation along one direction.

In summary, the key structural requirement for enhancing ionic transport in sulfide-based electrolytes has been to destabilize the octahedral sites by increasing the lattice volume, thereby inducing bcc-like conduction behavior. In our halide-based system, we achieved a similar outcome through the lattice expansion induced by the incorporation of divalent anions, which effectively reduced the activation energy. This effect is particularly pronounced in halides incorporating sulfur, where the originally inactive ccp-type lattice undergoes significant lattice expansion, dramatically lowering the activation energy and enabling the formation of a conductive framework which is a key mechanism underlying the enhanced ionic transport.

On the other hand, for oxide-based electrolytes, the ionic conduction enhancement mechanism relies on inducing Li site distortion, often realized through corner-sharing polyhedral frameworks. In our case, oxygen substitution introduces significant site distortion within the halide lattice due to the size mismatch between hetero anions, as evidenced by the elevated CSM values. This distortion weakens the bonding strength, as reflected by the reduced ICOHP values in **Fig. 5c,d**, which in turn induces a flattening of the energy landscape that facilitates enhanced ionic conduction.

So, we confirm that the phenomenological principles previously identified in sulfide and oxide frameworks can be effectively extended to halide systems via the incorporation of divalent anions. This effect is particularly evident in the case of oxygen incorporation, where the substantial ionic radius difference between O^{2-} and the original Cl^- induces significant distortion of the Li sites. In conclusion, this observation shows that the distinct mechanisms understood for each system (whether it be volume-driven site destabilization or local structural distortion) can coexist and be systematically harnessed within a single halide framework. We believe this transferable understanding of ionic conduction mechanisms across anion chemistries should be explicitly emphasized in the revised manuscript.

Accordingly, this point has been incorporated into the main text to provide a more comprehensive understanding, and the CSM values previously listed in the Supplementary Table have been visualized and newly included in the main figure (**Fig. 5b**). In addition, the discussion was expanded to explicitly contrast the effects of sulfur and oxygen substitution, based on the resulting structural and conduction phenomena, thereby elucidating how the previously understood phenomenological mechanisms are applicable to halide systems.

- **Line 309:** Interestingly, the lattice volume expansion, which underlies the design principles of sulfide SEs, and the Li site distortion, which is central to ionic conduction enhancement in oxide SEs, can both be achieved through the incorporation of divalent anions in monovalent halide lattices.^{31,43} However, though based on this fundamental principle, the dominant mechanisms affecting the structure differ depending on the divalent anion type. Oxygen in hcp lattice causes strong local distortions due to its size mismatch with Cl⁻, which significantly alters Li-site coordination and facilitates migration. Sulfur incorporation in ccp lattice follows the same mechanism but is particularly effective in activating transport along the ab-plane in the ccp-LZC structure. This is achieved by expanding interlayer spacing and slightly increasing Li content, which together enable conduction in an otherwise inactive dimension. A comparative summary of these mechanisms and their associated structural, morphological, and electrochemical effects is presented in **Scheme 1 and Supplementary Table 25**.

Revised Manuscript (Figure)

- (Added in revision) **Fig. 5b** CSM value of hcp-Li₂ZrCl₆, hcp-Li_{2.5}ZrCl_{5.5}O_{0.5}, ccp-Li₂ZrCl₆ and ccp-Li_{2.25}ZrCl_{5.75}S_{0.25}

3. *The computational study utilized a model structure of Li_{2.5}ZrCl_{5.5}O_{0.5} with hcp-LZC. However, if the ionic transport mechanism is based on the formation of nanocomposite? How can this computational study, based on single-crystalline Li_{2.5}ZrCl_{5.5}O_{0.5}, help reveal the mechanism?*

Response to comment 3: We appreciate the reviewer's insightful comment regarding the potential role of nanocomposite formation in the ionic transport mechanism. As noted in our response to Comment 1, both thermodynamic driving forces and experimental observations strongly support the formation of oxygen-substituted domains through interfacial phase formation. Specifically, the observed increase in Zr–Cl bond lengths and Zr–Zr distances from EXAFS analysis, together with the oxygen concentration gradients identified via STEM-EELS mapping (**Response Fig. 1** in response to comment 1), collectively confirm the incorporation of oxygen and the structural impact of such substitution.

(Added in revision) Supplementary Fig. 16. Nyquist plots of EIS results for hcp-Li₂ZrCl₆, 0.8Li₂O-ZrCl₄, ccp-Li₂ZrCl₆ and 0.8Li₂S-ZrCl₄ measured at various temperatures ranging from -20°C to 60°C .

We have also included low-temperature Nyquist plots in the revised manuscript (**Supplementary Fig. 16**). These data clearly show a single semicircle, supporting the interpretation that the observed ionic transport is dominated by a single, percolating phase with negligible contribution from grain boundaries or secondary phase. To further assess the possible presence of other resistive components affecting ion transport, we performed equivalent circuit fitting of the -20°C EIS data using both single-phase and multi-component models (**Supplementary Fig. 17**).

(Added in revision) Supplementary Fig. 17. Comparison of Nyquist plot with equivalent circuit fitting using single-phase and multi-component models for $0.8\text{Li}_2\text{O-ZrCl}_4$ (top) and $0.8\text{Li}_2\text{S-ZrCl}_4$ (bottom) at 20°C .

The single-phase model yielded a significantly better fit, whereas the resistance value associated with the secondary component in the multi-component model converged to a negligible value ($R_2 \approx 4.0 \times 10^{-35} \Omega$). This result strongly indicates the absence of secondary resistive phases within the resolution of the EIS analysis. Instead, the Nyquist plots of $0.8\text{Li}_2\text{A-ZrCl}_4$ ($A = \text{O}, \text{S}$) display a consistent shape across temperatures, closely resembling those observed for both hcp-LZC and ccp-LZC. Considering that ZrO_2 does not contain Li^+ and Li_2O possesses extremely low ionic conductivity ($\sim 10^{-12} \text{ S cm}^{-1}$)^{R7}, their contribution to overall ion transport is negligible.

The notable first-shell peak shift observed in the PDF analysis (**Fig. 2a**) confirms the partial substitution of anions, while the elongation of the Zr-Cl bond length, as captured in the EXAFS data (**Fig. 2c**), further supports the structural modification. These experimental observations consistently indicate that the oxygen substitution occurs in small amounts, predominantly within the nanocrystalline hcp phase. Confirming that the oxygen-substituted phase is extensively formed at the interfaces and dominates ionic transport, we constructed our computational model accordingly. In particular, $\text{Li}_{2.5}\text{ZrCl}_{5.5}\text{O}_{0.5}$ reproduces the $\sim 0.05 \text{ \AA}$ backward shift in the Zr-Cl peak observed in EXAFS (**Fig. 2c**), as also reflected in the simulated RDF profiles (**Supplementary Fig. 21**).

More importantly, oxygen-substituted compositions (from $x = 0.166$ to $x = 0.833$ in $\text{Li}_{2+x}\text{ZrCl}_{6+x}\text{O}_x$), regardless of the precise level of substitution, exhibit consistent phenomenological features. These include enhanced Li-ion diffusivity (**Fig. 4b**), elongation of the Zr-Cl bond length (**Fig. 2c** and **Supplementary Fig. 21**), and lattice expansion due to increased anion-anion repulsion compared to the bare hcp-LZC structure

(Supplementary Table 15). To clarify this point, we further examined whether both the weakening of Li–Cl bonding and the diversification of site energies are consistently present across compositions.

Response Fig. 12. Histograms for distribution (cutoff = 0.1) of ICOHP values and average ICOHP values for Li-Cl bonds in $\text{Li}_{2+x}\text{ZrCl}_{6-x}\text{O}_x$ ($x = 0.166, 0.333, 0.667, 0.833$).

Compared to the original LZC structure, every oxygen substituted structure exhibits a similar degree of bond weakening and energy distribution broadening like $\text{Li}_{2.5}\text{ZrCl}_{5.5}\text{O}_{0.5}$, confirming that these effects are intrinsic to the oxygen-substituted structure. However, among each model structure in $\text{Li}_{2+x}\text{ZrCl}_{6-x}\text{O}_x$ (from $x = 0.166$ to $x = 0.833$), this specific composition was selected because it best matches the experimental EXAFS features among all investigated compositions.

This suggests that the selected model reliably captures the essential features of the ion transport mechanism, even if nanoscale phase heterogeneity exists. Although the computational study is based on a specific compositional model, it encapsulates the key structural and energetic characteristics common to oxygen-substituted halide frameworks. Furthermore, this composition was identified as the closest match to experimental observations, reinforcing its validity in providing meaningful insights into the underlying transport mechanism.

[Revision]

Revised Supplementary Information (Figure)

(Added in revision) Supplementary Fig. 16. Nyquist plots of EIS results for hcp- Li_2ZrCl_6 , $0.8\text{Li}_2\text{O}-\text{ZrCl}_4$, ccp- Li_2ZrCl_6 and $0.8\text{Li}_2\text{S}-\text{ZrCl}_4$ measured at various temperatures ranging from $-20\text{ }^\circ\text{C}$ to $60\text{ }^\circ\text{C}$.

(Added in revision) Supplementary Fig. 17. Comparison of Nyquist plot with equivalent circuit fitting using single-phase and multi-component models for $0.8\text{Li}_2\text{O}-\text{ZrCl}_4$ (top) and $0.8\text{Li}_2\text{S}-\text{ZrCl}_4$ (bottom) at $-20\text{ }^\circ\text{C}$.

4, Electrochemical stability. Authors stated that all the halide electrolytes are not stable below 2.0 V due to the Zr reduction. The results are also shown in Figure S18. More details are needed here to indicate how the CV tests were performed. If the cell was polarized to a higher voltage first, then the reduction peak can be from the oxidized product during the CV test, instead of the intrinsic anodic stability of solid electrolytes. Authors are suggested to test the linear sweep of two cells: one to high voltages, and the other to low voltages to understand the electrochemical stability of the electrolytes.

Response to comment 4: We appreciate the reviewer's comment and the opportunity to clarify the electrochemical stability analysis. We acknowledge the limitations of using CV alone for determining the electrochemical stability window (ESW), as oxidative decomposition products can, in certain cases, undergo reductive decomposition and be mistakenly interpreted in reverse. As reviewer suggested, to address this concern, we also performed linear sweep voltammetry (LSV) measurements, which provide a more direct assessment of the onset potentials for oxidation and reduction (**Supplementary Fig. 34**).

(Added in revision) Supplementary Fig. 34. Linear sweep voltammetry (LSV) curves for hcp- Li_2ZrCl_6 (left), $0.8\text{Li}_2\text{O-ZrCl}_4$ (middle), $0.8\text{Li}_2\text{S-ZrCl}_4$ (right) in (SE-carbon) |SE|LPSC|(Li-In) cells at 0.1 mV s^{-1} and $25 \text{ }^\circ\text{C}$. The weight ratio of SE:carbon is 7:3.

The LSV measurements were conducted without prior polarization to high voltages, ensuring that the observed reduction peak originates from the intrinsic reduction of Zr^{4+} , rather than from any oxidized intermediates. The results from the LSV are consistent with those obtained from cyclic voltammetry (CV), as shown in **Supplementary Fig. 33**.

(Added in revision) Supplementary Fig. 35. Intrinsic stability window and phase equilibria of hcp- Li_2ZrCl_6 , $\text{Li}_{2.5}\text{ZrCl}_{5.5}\text{O}_{0.5}$ and $\text{Li}_{2.25}\text{ZrCl}_{5.75}\text{S}_{0.25}$.

Moreover, the experimentally observed reduction onset is similar with the thermodynamic stability window predicted by first-principles calculations in **Supplementary Fig. 35**, further confirming that the stability limit near $2.0 \text{ V vs. Li/Li}^+$ is intrinsic to the Zr-based system. Based on our computational results, none of the oxidation products predicted by DFT are thermodynamically susceptible to reductive decomposition within the stable voltage range of solid electrolytes. This is consistent with the similarity observed between the CV and LSV profiles. However, to address potential concerns regarding this interpretation, we will include both the LSV data and the corresponding DFT-predicted decomposition phases in the Supplementary Information to further substantiate our conclusion. We sincerely appreciate the reviewer's thoughtful comment addressing a potentially ambiguous point.

Revised Manuscript

Additionally, like Li_2ZrCl_6 , $0.8\text{Li}_2\text{O}-\text{ZrCl}_4$ and $0.8\text{Li}_2\text{S}-\text{ZrCl}_4$ exhibited a reduction limit near 2 V (vs. Li/Li^+) due to the same central metal (Zr) element, as confirmed by LSV and the intrinsic stability window (Supplementary Fig. 33-35).

Revised Supplementary Information (Figure)

(Added in revision) Supplementary Fig. 34. Linear sweep voltammetry (LSV) curves for hcp- Li_2ZrCl_6 (left), $0.8\text{Li}_2\text{O}-\text{ZrCl}_4$ (middle), $0.8\text{Li}_2\text{S}-\text{ZrCl}_4$ (right) in (SE-carbon) |SE|LPSC|(Li-In) cells at 0.1 mV s^{-1} and 25°C . The weight ratio of SE:carbon is 7:3.

(Added in revision) Supplementary Fig. 35. Intrinsic stability window and phase equilibria of hcp- Li_2ZrCl_6 , $\text{Li}_{2.5}\text{ZrCl}_{5.5}\text{O}_{0.5}$ and $\text{Li}_{2.25}\text{ZrCl}_{5.75}\text{S}_{0.25}$.

Reviewer #3

The authors investigate how divalent anions (O/Cl and S/Cl) modulate the anion sublattice and associated superionic conductivity within Zr-based halide solid electrolytes, aiming to resolve ambiguities concerning their structure and diffusion mechanisms. Given the growing body of research in oxychloride solid electrolytes, the topic of this manuscript is timely and significant. However, the current manuscript lacks detailed analysis of the structural characterization results, and its arguments regarding the diffusion mechanism require further clarification, particularly how they differ from findings in previous reports on zirconium-based oxyhalides (*Nat Commun* 14, 2459 (2023), *Nat Commun* 14, 3807 (2023), and *J. Am. Chem. Soc.* 2024, 146, 5, 2977–2985). A major revision is required before considering it for publication in *Nature Communications*.

Response to comment: We appreciate the reviewer's insightful comments regarding the need for more rigorous analysis of the structural characterization and a clearer distinction of our proposed diffusion mechanism from prior reports on zirconium-based oxyhalides.

Prior studies have reported various structural motifs of Zr-based oxyhalide (including ccp monoclinic³⁴, hcp trigonal³¹, ccp-based amorphous³² and hcp-based amorphous³³). However, these structural interpretations remain somewhat ambiguous due to the inherent complexity introduced by nanoscale phase domains resulting from oxygen incorporation. This issue is particularly pronounced in zirconium-based systems, which intrinsically exhibit a strong tendency toward nanoparticulate formation.^{R21}

Through this revision, we conducted additional structural analyses by employing pair distribution function (PDF) and extended X-ray absorption fine structure (EXAFS) fitting to quantify phase ratios more accurately. Furthermore, Raman spectroscopy was utilized to verify the preservation of polyhedral units in both hcp and ccp frameworks, thereby enhancing confidence in our structural interpretation. Electrochemical impedance spectroscopy (EIS) fitting confirmed that the observed ionic conduction originates primarily from single-phase hcp or ccp domains, not from interfacial or amorphous components. By isolating key parameters such as hopping frequency and migration entropy, we clarified the fundamental conduction mechanism, which was directly correlated with intrinsic lattice distortions. This enables a more mechanistic understanding and sets our study distinctly apart from earlier work.

Consequently, accurate structural interpretation requires not only conventional crystallographic techniques but also careful consideration of local and medium-range order within nanoscale domains. Our work aims to address this by providing a more systematic framework for analyzing structural-diffusional relationships, specifically focusing on the effect of oxygen and sulfur substitution on lattice distortions and site environment. The following discussion highlights the distinctions between amorphous zirconium oxyhalides and lattice-based counterparts.

1. Comparison of our lattice-based 0.8Li₂O–ZrCl₄ with previously reported amorphous Zr-Based oxyhalides (Li_{1.75}ZrCl_{4.75}O_{0.5} and Li₃ZrCl₄O_{1.5})

Li-Zr-Cl-O compositions reported in *References* 32 and 33 are interpreted as have high ionic conductivity primarily due to their high amorphous content. Indeed, we also observed XRD-amorphous-like features in our samples when measured using lab-based XRD ($\lambda = 1.54 \text{ \AA}$) similar with two references. Samples with lower Li₂O content, such as 0.8Li₂O–ZrCl₄, exhibited broad diffraction peaks, giving the impression of being amorphous, which could have led to an interpretation that the enhanced ionic conductivity originates from amorphization (**Fig. 1a**).

However, to distinguish whether the enhanced ionic conductivity in our system arises from amorphization or a specific nanocrystalline based on interfacial mechanism, we employed high-energy synchrotron XRD ($\lambda = 0.1665 \text{ \AA}$), which offers much higher resolution and sensitivity to detect nanoscale ordering. Lab source XRD patterns appeared broad and poorly resolved XRD pattern, but under these conditions, we clearly identified the primary phase as the hcp structure in 0.8Li₂O–ZrCl₄. Although the synthesized phase may vary depending on experimental conditions, accurate phase identification and a clear elucidation of how the observed phase contributes to enhanced material properties are essential.

This highlights the importance of high-intensity and short-wavelength beam source when evaluating materials with potentially nanocrystalline domains, as small crystalline grain signals may be undetectable due to the peak broadening governed by the Scherrer equation ($\tau = \frac{K\lambda}{\beta \cos\theta}$, where τ : **detectable grain size**, K : shape factor, λ : **wavelength**, β : line broadening at FWHM, θ : Bragg angle).

Moreover, where ionic conductivity increases with higher amorphous content, we observed that samples with lower crystallinity (such as $1.0\text{Li}_2\text{O-ZrCl}_4$ and $1.2\text{Li}_2\text{O-ZrCl}_4$) showed reduced ionic conductivity compared to $0.8\text{Li}_2\text{O-ZrCl}_4$ (**Fig. 1b,c**). This trend clearly indicates that, in contrast to previously reported studies where enhanced ionic transport was attributed to amorphization, the improved conductivity in our system arises from a crystalline phase, not from the amorphous component. We attribute the improved conductivity to the formation of a structurally and compositionally well-defined oxygen-substituted hcp phase, which supports efficient Li-ion conduction through lattice expansion, increased Li site distortion, and diversified Li-Cl bonding environments. These changes are supported by both experimental characterization and first-principles simulations.

Additionally, to assess the presence and structural impact of amorphous phase in our system, we performed detailed pair distribution function (PDF) analyses and examined both local and intermediate-range orders. The results collectively indicate that while short-range ordered ZrO_2 and ZrS_2 phases account for approximately 30% of the structure, amorphization within the conductive hcp and ccp frameworks is not observed (**Supplementary Fig. 4,9** and **Supplementary Table 3,4,10,11**).

First, no attenuation of the peak corresponding to hcp and ccp phases was observed within the PDF $G(r)$ range of 1.5-6 Å for the compositions $x\text{Li}_2\text{O-ZrCl}_4$ and $x\text{Li}_2\text{S-ZrCl}_4$ ($x = 0.6, 0.8, 1.0$) (**Fig. 2a, 3a**). Given that 6 Å corresponds approximately to the lattice parameter along the c-axis of Li_2ZrCl_6 , peak intensity reduction should be observed if amorphization occurs which disrupt lattice periodicity. However, such intensity attenuation is completely absent in both $x\text{Li}_2\text{O-ZrCl}_4$ and $x\text{Li}_2\text{S-ZrCl}_4$ compositions within 1.5-6 Å. Thus, we conclude that there is an only negligible amount of amorphous phase formation in these materials at compositions corresponding to the highest ionic conductivity ($x = 0.6, 0.8, 1.0$).

Second, in our case, the Raman spectra (now included as main data in the revised manuscript) show that the characteristic vibrational modes of Zr-Cl (≈ 325 and ≈ 161 cm^{-1}) in isolated octahedron remain clearly identifiable across all compositions (hcp-LZC, $0.8\text{Li}_2\text{O-ZrCl}_4$, ccp-LZC and $0.8\text{Li}_2\text{S-ZrCl}_4$). Also, in contrast to ZrCl_4 , the signatures of bridging octahedra (≈ 410 and ≈ 145 cm^{-1}) disappeared, and two distinct peaks at ≈ 325 and ≈ 161 cm^{-1} emerged, corresponding to the A_{1g} stretching and F_{2g} bending modes, respectively, which are characteristic of elpasolite-type compounds such as $\text{Cs}_2\text{LiYCl}_6$ and $\text{Cs}_2\text{NaYCl}_6$.^{R3} These modes closely resemble those observed in closed packed structures (hcp and ccp), suggesting that the main phase with polyhedral framework is structurally preserved despite the introduction of divalent anions. This observation indicates that the divalent anion incorporation does not lead to significant amorphization but instead maintains a close-packed structural environment that retains the key bonding features in main phase.

(Added in revision) Fig. 2d Raman spectra of ZrCl_4 , hcp- Li_2ZrCl_6 (hcp-LZC) and $0.8\text{Li}_2\text{O}-\text{ZrCl}_4$, **(left)** and **Fig. 3d** Raman spectra for ZrCl_4 , ccp- Li_2ZrCl_6 (ccp-LZC) and $0.8\text{Li}_2\text{S}-\text{ZrCl}_4$. **(right)**

2. Comparison of our lattice-based $0.8\text{Li}_2\text{O}-\text{ZrCl}_4$ with hcp trigonal $\text{ZrO}_2-\text{Li}_2\text{ZrCl}_6$ & ccp monoclinic $\text{Li}_{3.1}\text{ZrCl}_{4.9}\text{O}_{1.1}$

In our study, we found that the zirconium-based oxyhalide phase can be interpreted as ccp-type monoclinic, but it can be mistakenly assigned due to overlapping features with LiCl (**Fig. 1c**). However, our analysis clearly distinguishes this phase from LiCl, while also revealing that the formation of LiCl is thermodynamically possible under certain compositional conditions. More critically, the observed enhancement in ionic conductivity is not attributed to the presence of LiCl, but rather to the formation of an oxygen-substituted hcp-type structure driven by the incorporation of ZrO_2 . This structural motif corresponds well with previously reported hcp-trigonal frameworks ($\text{ZrO}_2-\text{Li}_2\text{ZrCl}_6$) and supports our interpretation.³¹ What sets our work apart is the identification of a composition that maximizes oxygen substitution within the lattice (with Li_2O) and elucidation of the underlying mechanism, which directly correlates with improved ionic conductivity. This observation highlights the pivotal role of oxygen incorporation in altering the local structural environment and enhancing transport properties.

Most importantly, our findings demonstrate that both lattice expansion and the accompanying lattice distortion, particularly the perturbation of Li sites that leads to energy landscape flattening, collectively contribute to the enhancement of ionic conductivity. In addition, sulfur substitution also induces noticeable lattice changes, further supporting the conclusion that the introduction of divalent anions plays a critical role in enhancing ionic conductivity in Zr-based halide systems. This enhancement is achieved by structurally modifying the lattice to form a more conductive framework. This provides a meaningful insight into how halide systems can simultaneously embody design principles from both sulfide (via bcc-like lattice expansion) and oxide (via Li-site distortion) electrolytes through divalent anion substitution.^{R19,20}

Oxygen and sulfur respectively induce Li-site distortion and energy landscape flattening within hcp and ccp lattices, thereby activating ionic conduction pathways. Oxygen incorporation, due to its large ionic radius mismatch with Cl^- , causes substantial lattice strain, weakens Li-Cl interactions, and diversifies the potential energy of Li sites, collectively enhancing dynamic Li-ion transport. In contrast, sulfur substitution induces relatively milder distortion but plays a critical role in enabling ion conduction within the originally non-conductive ccp lattice by expanding the lattice along the c-axis and activating conduction along the ab plane, ultimately surpassing the structural threshold required for efficient ionic conduction. Through this revision, we have clarified the above key aspects, thereby establishing a clear distinction from previous studies.

[Revision]

Revised Manuscript

- **Line 77:** However, the challenge lies in the unclear understanding of divalent anion incorporation in Zr-based halide SEs, with ongoing debates on whether the synthesized compounds exist in an amorphous state, adopt a ccp structure, or form a nanocrystalline phase.^{34,35} **Although the synthesized phase may vary depending on experimental conditions, accurate phase identification and a clear elucidation of how the observed phase contributes to enhanced material properties are essential.**

Revised Figure

(Added in revision) Fig. 2d Raman spectra for ZrCl_4 , hcp-LZC, and $0.8\text{Li}_2\text{O}-\text{ZrCl}_4$.

(Added in revision) Fig. 3d Raman spectra for ZrCl_4 , ccp-LZC, and $0.8\text{Li}_2\text{S}-\text{ZrCl}_4$.

Revised Supplementary Information (Text)

(Added in revision) Supplementary text 3. Amorphous phase

The peaks corresponding to the hcp and ccp phases were well maintained within the PDF $G(r)$ range of 1.5-6 Å for the compositions $x\text{Li}_2\text{O}-\text{ZrCl}_4$ and $x\text{Li}_2\text{S}-\text{ZrCl}_4$ ($x = 0.6, 0.8, 1.0$) (Fig. 2a, 3a). Given that ~6 Å corresponds approximately to the lattice parameter (interlayer distance) along the c-axis of Li_2ZrCl_6 (6.030 Å for hcp and 6.376 Å for ccp), the sustained intensity of this peak suggests that the long-range periodicity associated with these lattice frameworks is preserved. The consistent peak behavior in this region thus supports the retention of hcp and ccp structural motifs without meaningful evidence of amorphous phase development. Thus, we conclude that there is an only negligible amount of amorphous phase formation in these materials at compositions corresponding to the highest ionic conductivity ($x = 0.6, 0.8, 1.0$). While structural amorphization in Zr-based oxyhalide has been proposed in prior studies as a mechanism for enhancing ionic conductivity, particularly within each respective systems, our compositions and results suggest a different origin for the observed conductivity. Samples with lower crystallinity (such as $1.0\text{Li}_2\text{O}-\text{ZrCl}_4$ and $1.2\text{Li}_2\text{O}-\text{ZrCl}_4$) exhibited reduced ionic conductivity compared to the more crystalline $0.8\text{Li}_2\text{O}-\text{ZrCl}_4$ composition (Fig. 1b,c). This behavior shows that amorphization was not linked to improved ionic transport, suggesting instead that the high ionic conductivity in our system originates from an optimized crystalline structure rather than from any amorphous contribution. Thus, we conclude that there is only negligible amount of amorphous phase in these materials at compositions corresponding to the highest ionic conductivity ($x = 0.6, 0.8, 1.0$).

1. The discussion of the synchrotron XRD, PDF, and EXAFS data is primarily qualitative. Quantitative analyses, such as results from XRD refinement and PDF/EXAFS fitting, are essential for substantiating the structural claims and are currently missing from the manuscript.

Response to comment 1: We are thankful to reviewer's careful comment. Although the synthesized samples exhibit crystallinity, they possess nanocrystalline features that result in significant background signals and peak broadening in the XRD patterns, introducing substantial uncertainty in phase quantification and the potential for misleading interpretations. These limitations significantly hinder reliable phase fraction quantification using XRD. Additionally, residual phases such as ZrO_2 and ZrS_2 , which exhibit only short-range ordering, coexist within the samples and remain undetectable even when using high-energy X-rays.

Therefore, to obtain more reliable estimates of the phase fractions, we employed pair distribution function (PDF) analysis and extended X-ray absorption fine structure (EXAFS) fitting, both of which are less sensitive to long-range order and thus more suitable for accurate structural quantification in such systems. Similarly, refinements for compositions beyond $x = 1.0$ in the $x\text{Li}_2\text{O}-\text{ZrCl}_4$ and $x\text{Li}_2\text{S}-\text{ZrCl}_4$ systems were also not pursued, as significant reductions in first-shell intensity (Zr-Cl and Zr-S pairs in hcp and ccp phases, respectively) indicated the formation of the potential non-ionic conductive by-products (LiCl). In response to the reviewer's suggestion, we have now performed quantitative PDF and EXAFS analyses to estimate the respective phase fractions with improved reliability.

We performed PDF refinements to identify the dominant phase and quantify phase fraction in $x\text{Li}_2\text{O}-\text{ZrCl}_4$ and $x\text{Li}_2\text{S}-\text{ZrCl}_4$ system (Supplementary Fig. 4,9 and Supplementary Tables 1-6, 8-13). Given that overwhelming scattering power from Zr in ZrO_2 , ZrS_2 , and the hcp, ccp phases compared to that of Li_2O and LiCl , the latter two were excluded in fitting to reduce parameter thereby securing fitting reliability. Additionally, anion exchange was not considered due to the following reasons. First, a slight first shell intensity reduction was observed in $x\text{Li}_2\text{O}-\text{ZrCl}_4$ ($x=0.6, 0.8, 1.0$) due to anion exchange (oxygen substitution) in hcp- Li_2ZrCl_6 (Fig. 2a). However, as this reduction is marginal because of the slight amount of oxygen substitution level, the fitting for hcp- Li_2ZrCl_6 was performed considering only the Zr-Cl coordination and excluding small amount of Zr-O bond in hcp- Li_2ZrCl_6 . Second, the isoelectronic feature of Cl and S (18 electrons) make it heavily challenging to refine substitution fractions due to similar X-ray scattering power.

We also conducted EXAFS refinement to figure out Zr-Zr distance in ZrO₂/ZrS₂ and hcp/ccp lattice element selectively. (**Supplementary Fig. 6,11** and **Supplementary Tables 7,14**) We constructed EXAFS fitting model for ZrO₂ which nine coordinated Zr-Zr distances were divided into three subsets to accurately capture Zr-Zr correlations in ZrO₂. The site disorder at M2-M3 in the hcp phase was fixed at 50% based on our long-range region (15-30 Å) PDF fitting results, which demonstrate experimental observation for M2-M3 site disorder is near 0.5 (**Supplementary Tables 2,4 and 6**). The phase fraction results obtained from EXAFS analyses align well with those from PDF refinements, further reinforcing the reliability of our fitting methodology.

Although Li₂O was excluded from the refinements due to low scattering factors, the obtained fractions of ZrO₂ vs. hcp phase (33:66 \cong 1:2) and ZrS₂ vs. ccp phase (7:16 \cong 3:7), as shown in **Response Table 1**, closely matched computational predictions ((interfacial substitution in hcp) 2Li₂O + 5ZrO₂ + 10Li₂ZrCl₆ for 0.8Li₂O–ZrCl₄ and (bulk substitution in ccp) 7ZrS₂ + 16Li_{2.25}ZrCl_{5.75}S_{0.25} for 0.8Li₂S–ZrCl₄ in **Supplementary Table 18,19**). This agreement strongly suggests the negligible unidentified phases or additional features that could be attributed to alternative components in the xLi₂O–ZrCl₄ and xLi₂S–ZrCl₄ systems. Therefore, we suggest that the phase fractions of Li₂O, ZrO₂ and ZrS₂, as well as the amount of O and S substitution in the hcp and ccp phases closely align with computationally predicted values. This agreement between the reaction phases and their ratios, as experimentally observed and computationally predicted based on reaction energy calculations, reinforces the reliability of phase fraction estimations from both theoretical and experimental perspectives.

Composition	ZrO ₂ (PDF) (%)	hcp (PDF) (%)	ZrO ₂ (EXAFS) (%)	hcp (EXAFS) (%)
0.6Li ₂ O–ZrCl ₄	35.4	64.6	29.9	70.0
0.8Li₂O–ZrCl₄	34.9	65.1	33.12	67.9
1.0Li ₂ O–ZrCl ₄	50.1	49.9	48.2	51.8
Composition	ZrS ₂ (PDF) (%)	ccp (PDF) (%)	ZrS ₂ (EXAFS) (%)	ccp (EXAFS) (%)
0.6Li ₂ S–ZrCl ₄	35.8	64.2	33.3	66.7
0.8Li₂S–ZrCl₄	31.9	68.1	29.9	70.1
1.0Li ₂ S–ZrCl ₄	28.6	71.4	23.4	76.56

Response Table 1. Phase fraction of PDF and EXAFS fitting in xLi₂O–ZrCl₄ (x = 0.6, 0.8, 1.0) and xLi₂S–ZrCl₄ (x = 0.6, 0.8, 1.0)

(Added in revision) Supplementary Fig. 4. Observed and calculated PDF short-range region (1.5-4.1 Å, top) and long-range region (15-30 Å, bottom) fitting curves for 0.6Li₂O–ZrCl₄, 0.8Li₂O–ZrCl₄ and 1.0Li₂O–ZrCl₄. The detailed results are summarized in **Supplementary Table 1-6**.

(Added in revision) Supplementary Fig. 6. Zr K-edge EXAFS results. The real/imaginary part of FT (left) and $k^3 \chi(k)$ (right) based on R-space curve fitting of x Li₂O–ZrCl₄ ($x = 0.6, 0.8, 1.0$). The detailed results are summarized in **Supplementary Table 7**.

(Added in revision) Supplementary Fig. 9. Observed and calculated PDF short-range region (1.5-4.1 Å, top) and long-range region (15-30 Å, bottom) fitting curves for 0.6Li₂S-ZrCl₄, 0.8Li₂S-ZrCl₄ and 1.0Li₂S-ZrCl₄. The detailed results are summarized in **Supplementary Table 8-13**.

(Added in revision) Supplementary Fig. 11. Zr K-edge EXAFS results. The real/imaginary part of FT (left) and $k^3\chi(k)$ (right) based on R-space curve fitting of $x\text{Li}_2\text{S-ZrCl}_4$ ($x = 0.6, 0.8, 1.0$). The detailed results are summarized in **Supplementary Table 14**.

Following the reviewer's valuable recommendations, we have included detailed descriptions of our quantitative analysis and phase fraction determination in the manuscript and **Supplementary Texts**. The *Methods* section has also been updated accordingly.

[Revision]

Revised manuscript

- **Line 151:** Additionally, the presence of a Zr–O pair in monoclinic ZrO_2 (**Supplementary Fig. 3** and **Supplementary Text 1**) at 2.1 Å in all compositions suggests the formation of nanosized ZrO_2 domains. Additionally, PDF fitting was conducted to estimate the phase fractions of ZrO_2 and the hcp phase (**Supplementary Fig. 4** and **Supplementary Table 1-6**). For $x = 0.6$ and 0.8, the long-range region (15–30 Å), where only crystalline phases contribute, was well-reproduced using a single hcp model with ~50% M2–M3 site disorder, confirming the coexistence of crystalline hcp and short range ordered ZrO_2 domains. In contrast, for $x = 1.0$, the fitting yielded a substantially higher R_w value, suggesting progressive structural distortion of the hcp framework due to oxygen incorporation and the onset of phase transformation (**Supplementary Table 6**).
- **Line 164:** Furthermore, the extended X-ray absorption fine structure (EXAFS) spectra in **Fig. 2c** show overall positive peak shifts toward a longer region for the Zr–Cl bond in the hcp phase

around 2.0 Å, as well as broadening of the Zr-Zr bonding peaks from 3.0 to 3.4 Å compared to bare hcp-LZC and ZrO₂, respectively (**Supplementary Fig. 5**). Also, our EXAFS fitting results clearly show such elongation (**Supplementary Fig. 6**).

- **Line 171:** Conversely, the substitution of Cl in ZrO₂ increases the Zr-Zr distance and local distortion due to the larger anion radius of Cl, suggesting anion exchange between ZrO₂ and Li₂ZrCl₆, consistent with our previous study.³¹ Additionally, phase fraction for ZrO₂ in xLi₂O-ZrCl₄ (x = 0.6-1.0) are well aligned with our PDF fitting results, further supporting reliability of our refinement (**Supplementary Table 7** and **Supplementary Text 2**).
- **Line 190:** Notably, small peaks at ~4.98 Å, attributed to ZrS₂, suggest the presence of ZrS₂ with only short-range ordering. The PDF fitting results and phase fraction of ZrS₂ for xLi₂S-ZrCl₄ (x = 0.6-1.0) are shown in **Supplementary Fig. 9** and **Supplementary Table 8-13**.
- **Line 195:** The EXAFS results in **Fig. 3c** display a noticeable positive peak shift upon increasing Li₂S, with low fractions of Li₂S (x ≤ 1.2) showing a similar tendency of divalent-anion-driven bond elongation observed in xLi₂O-ZrCl₄, but within the ccp structure (**Supplementary Fig. 10**) and our EXAFS fitting further support such elongation (**Supplementary Fig. 11**). Additionally, the phase fraction of ZrS₂ refined by EXAFS fitting demonstrates similarity to those predicted by PDF, analogous to the case of O-substituted hcp phases. (**Supplementary Table 14** and **Supplementary Text 3**).

Revised Manuscript (Methods)

XANES and EXAFS data were processed using the Demeter software package. The extracted EXAFS signal, $k^3\chi(k)$, was Fourier transformed over a k -range of 3.1-12.8 Å⁻¹ for xLi₂O-ZrCl₄ and 3.8-11.2 Å⁻¹ for xLi₂S-ZrCl₄ to obtain magnitude plots of the EXAFS spectra. EXAFS are fitted over a r -range of 1.35-3.5 Å for xLi₂O-ZrCl₄ and 1.7-3.7 Å for xLi₂S-ZrCl₄. Detailed fitting procedures for PDF and EXAFS are shown in **Supplementary Text 2**.

Revised Supplementary Information (Figure)

(Added in revision) Supplementary Fig. 4. Observed and calculated PDF short-range region (1.5-4.1 Å, top) and long-range region (15-30 Å, bottom) fitting curves for 0.6Li₂O-ZrCl₄, 0.8Li₂O-ZrCl₄ and 1.0Li₂O-ZrCl₄. The detailed results are summarized in **Supplementary Table 1-6**.

(Added in revision) Supplementary Fig. 6. Zr K-edge EXAFS results. The real/imaginary part of FT (left) and $k^3\chi(k)$ (right) based on R-space curve fitting of xLi₂O-ZrCl₄ (x = 0.6, 0.8, 1.0). The detailed results are summarized in **Supplementary Table 7**.

(Added in revision) Supplementary Fig. 9. Observed and calculated PDF short-range region (1.5-4.1 Å, top) and long-range region (15-30 Å, bottom) fitting curves for 0.6Li₂S-ZrCl₄, 0.8Li₂S-ZrCl₄ and 1.0Li₂S-ZrCl₄. The detailed results are summarized in **Supplementary Table 8-13**.

(Added in revision) Supplementary Fig. 11. Zr K-edge EXAFS results. The real/imaginary part of FT (right) and $k^3\chi(k)$ (left) based on R-space curve fitting of xLi₂S-ZrCl₄ (x = 0.6, 0.8, 1.0). The detailed results are summarized in **Supplementary Table 14**.

Revised Supplementary Information (Table)

(Added in revision) Supplementary Table 1-6. PDF fitting results of xLi₂O-ZrCl₄ (x=0.6, 0.8, 1.0)

(Added in revision) Supplementary Table 7. EXAFS fitting results of xLi₂O-ZrCl₄ (x=0.6, 0.8, 1.0) corresponding to Supplementary Fig. 6.

(Added in revision) Supplementary Table 8-13. PDF fitting results of xLi₂S-ZrCl₄ (x=0.6, 0.8, 1.0)

(Added in revision) Supplementary Table 14. EXAFS fitting results of xLi₂S-ZrCl₄ (x=0.6, 0.8, 1.0) corresponding to Supplementary Fig. 11.

Revised Supplementary Information (Text)

(Added in revision) Supplementary Text 2. Phase ratio of PDF and EXAFS fitting model for Supplementary Fig. 4,6,9,11 and Supplementary Table 1-14.

Given that overwhelming scattering power from Zr in ZrO_2 , ZrS_2 , and the hcp, ccp phases compared to that of Li_2O and $LiCl$, the latter two were excluded in PDF and EXAFS fitting to reduce parameter thereby securing fitting reliability. Additionally, anion exchange was not considered in PDF and EXAFS refinement due to the following reasons:

3. A slight first shell intensity reduction was observed in $xLi_2O-ZrCl_4$ ($x=0.6, 0.8, 1.0$) due to anion exchange in hcp- Li_2ZrCl_6 . (**Fig. 2a**) However, as this reduction is marginal, the fitting for hcp- Li_2ZrCl_6 was performed considering only the Zr-Cl coordination and excluding small amount of Zr-O bond.
4. The isoelectronic feature of Cl and S (18 electrons) make it heavily challenging to refine substitution fraction due to similar X-ray scattering power.

We constructed EXAFS fitting model for ZrO_2 which nine coordinated Zr-Zr distances were divided into three subsets to capture Zr-Zr correlations in amorphous ZrO_2 . Continually, site disorder at M2-M3 in the hcp phase was fixed at 50% based on our PDF fitting results, which demonstrate M2-M3 site disorder is near 0.5. (**Supplementary Tables 2, 4, 6**)

Although Li_2O and Li_2S were excluded from the refinements, the obtained fractions of ZrO_2 vs. hcp phase and ZrS_2 vs. ccp phase by both PDF and EXAFS fitting closely matched with computational predictions (**Supplementary Tables 1-14**) This agreement strongly suggests the absence of residual precursor or unreacted Li_2ZrCl_6 phases in the $xLi_2O-ZrCl_4$ and $xLi_2S-ZrCl_4$ systems. Therefore, we suggest that the phase fractions of Li_2O and Li_2S , as well as the amount of O and S substitution in the hcp and ccp phases, closely align with computationally predicted values.

2. The authors claim that “the Li_2O promotes the nanostructuring of an oxygen-substituted hcp anionic sublattice, generating a nanocomposite state with oxide compounds (Li_2O and ZrO_2). In contrast, the sulfur source (Li_2S) drives the formation of a sulfur-substituted ccp anionic sublattice.” This argument is not fully persuasive due to the absence of quantitative structural analysis (as noted in comments 1). What is the ratio of Li_2O and ZrO_2 in the structure? How many O and S are substituted into the hcp or ccp lattice?

Response to comment 2: We appreciate the reviewer’s critical insight regarding the need for quantitative structural evidence supporting our claim.

As detailed in our **Response to Comment 1**, we conducted quantitative phase analysis using PDF and EXAFS fitting. These analyses revealed the relative phase fractions of ZrO_2 and the hcp phase (~ 1:2) for the $0.8Li_2O-ZrCl_4$, and ZrS_2 and the ccp phase (~ 3:7) for the $0.8Li_2S-ZrCl_4$. Additionally, STEM-EELS analysis (**Response Fig. 1**) revealed that, in the case of oxygen, the interface region exhibits gradient in oxygen concentration. In contrast, sulfur was found to be uniformly distributed throughout the bulk, indicating a homogeneous incorporation into the lattice.

Response Fig. 1. STEM image and EELS (Li, Zr, Cl, O, S) elemental mapping (Li, Zr, Cl, O, S) of $0.8Li_2O-ZrCl_4$ and $0.8Li_2S-ZrCl_4$.

In the previous version of the manuscript, we proposed the most thermodynamically favorable phase ratios based on first-principles calculations. Although Li_2O was excluded from the refinements due to low scattering factors, the obtained fractions (by PDF and EXAFS) of ZrO_2 vs. hcp phase ($33:66 \cong 1:2$) and ZrS_2 vs. ccp phase ($7:16 \cong 3:7$) closely matched computational predictions ((interfacial substitution in hcp) $2\text{Li}_2\text{O} + 5\text{ZrO}_2 + 10\text{Li}_2\text{ZrCl}_6$ for $0.8\text{Li}_2\text{O}-\text{ZrCl}_4$ and (bulk substitution in ccp) $7\text{ZrS}_2 + 16\text{Li}_{2.25}\text{ZrCl}_{5.75}\text{S}_{0.25}$ for $0.8\text{Li}_2\text{S}-\text{ZrCl}_4$ in **Supplementary Table 18,19**). This indicates that the phase formed at the interface corresponds to an oxygen-substituted hcp structure (**Supplementary Text 4**). In contrast, because sulfur is incorporated into the bulk rather than the interface, the phase ratio between ZrO_2 and hcp in the oxygen-substituted system ($\sim 1:2$) differs from that of ZrS_2 and ccp in the sulfur-substituted counterpart ($\sim 3:7$), despite using the same amount of lithium source ($0.8\text{Li}_2\text{A}-\text{ZrCl}_4$).

Supplementary Table 18. Various chemical reaction equations and reaction energy (E_{Rxn}) with experimental ionic conductivity of oxygen-source component.

Index	Reaction Equations	E_{Rxn} (eV/atom)	σ_{Li^+} (Exp.) (mS cm^{-1})
1	$12\text{Li}_2\text{O} + 15\text{ZrCl}_4$ $\rightarrow 2\text{Li}_2\text{O} + 5\text{ZrO}_2 + 10\text{Li}_2\text{ZrCl}_6$	-0.219	1.78
2	$10\text{Li}_2\text{O} + 11\text{ZrCl}_4$ $\rightarrow 3\text{ZrO}_2 + 8\text{Li}_{2.5}\text{ZrCl}_{5.5}\text{O}_{0.5}$	-0.212	1.42

Supplementary Text 4. Reaction 1 ($x = 0.8$, interfacial anion substitution) vs. Reaction 2 ($x \approx 0.9$, doping all oxygen into the bulk without Li_2O)

The basis for judging that Reaction 2 is impossible and Reaction 1 is possible is as follows. First, Li_2O was observed in the synchrotron XRD pattern (Figure 1c), and the presence of Li_2O indicates that it is difficult for the structure to accommodate oxygen. Second, the E_{hull} in Supplementary Table 16 is relatively high, making direct substitution difficult in environments without oxygen concentration, such as at the interface. Third, if full substitution had occurred in the bulk, the highest conductivity would have been observed at $x = 0.9$; however, this is not the case. This implies that the composition where Li_2O and ZrO_2 most readily form an hcp- Li_2ZrCl_6 phase with oxygen-substituted interfaces is $x=0.8$.

Supplementary Table 19. Various chemical reaction equations and reaction energy (E_{Rxn}) with experimental ionic conductivity of sulfur-source component.

Index	Reaction Equations	E_{Rxn} (eV/atom)	σ_{Li^+} (Exp.) (mS cm^{-1})
8	$18\text{Li}_2\text{S} + 23\text{ZrCl}_4$ $\rightarrow 7\text{ZrS}_2 + 16\text{Li}_{2.25}\text{ZrCl}_{5.75}\text{S}_{0.25}$	-0.112	1.01

These phase ratios, initially estimated through thermodynamic modeling in our previous study, were experimentally confirmed to be in close agreement with the relative phase fractions derived from PDF and EXAFS fitting. Consequently, the presence of Li₂O observed in the XRD patterns is now understood to exist at approximately one-fifth the quantity of the hcp phase within the nanocomposite, further validating the reliability of our compositional estimation.

While the sulfur content can be more directly inferred from diffraction-based phase identification due to its homogeneous bulk incorporation, oxygen-substituted structures are inherently more challenging to quantify due to their nanocrystalline and short-range ordered nature (**Response Fig. 1** and **Fig. 2e**). In such systems, a gradient in oxygen content and local variations are inevitable. Nevertheless, multiple structural fingerprints from experimental EXAFS and PDF data provide compelling indirect evidence of oxygen substitution.

Notably, EXAFS analysis of the 0.8Li₂O–ZrCl₄ sample revealed a backward shift of ~0.05 Å in the Zr–Cl peak (**Fig. 2c**), consistent with the expected structural perturbation from partial oxygen substitution. This shift was accurately reproduced by our computational model, Li_{2.5}ZrCl_{5.5}O_{0.5}, which also demonstrated excellent agreement with the simulated radial distribution function (RDF) profiles (**Supplementary Fig. 21**). Hence, Li_{2.5}ZrCl_{5.5}O_{0.5} serves as a representative structural model, effectively capturing the experimentally observed features and providing a reliable basis for understanding the local environment and ionic transport characteristics. This aspect is discussed in detail in **Supplementary Text 4,5**, and we revise the text to enhance clarity and improve the reader’s understanding of this point.

[Revision]

Revised Supplementary Information (Text)

Supplementary Text 5. Reaction 7 ($x \approx 0.67$, interfacial substitution) vs. Reaction 8 ($x \approx 0.78$, conditions required for doping all sulfur into the bulk)

Unlike oxygen, sulfur appears to undergo bulk substitution. As explained in the text, no ccp phase was observed among the Zr-based solid electrolytes synthesized via ball milling, and the E_{bulk} is not significantly high, with no Li₂S detected. Additionally, the domain size is sufficiently large for the crystal structure to be identified using a lab-source XRD in Figure 1d. The anionic sizes of Cl and S are also similar ($\text{Cl}^- = 1.81 \text{ \AA}$ and $\text{S}^{2-} = 1.84 \text{ \AA}$). If Li₂S were excessive, it would have been detected, but it was not. This result suggests that the phase is likely to have generally formed in the bulk near $x = 0.8$ (~18/23), making this assumption reasonable. **In contrast to 0.8Li₂O–ZrCl₄, because sulfur is incorporated into the bulk rather than the interface, the phase ratio between ZrO₂ and hcp in the oxygen-substituted system (~1:2) differs from that of ZrS₂ and ccp in the sulfur-substituted counterpart (~3:7), despite using the same amount of lithium source (0.8Li₂A–ZrCl₄). This ratio, confirmed by both computational predictions and experimental observations using PDF and EXAFS, further clarifies the difference in phase formation between oxygen- and sulfur-substituted systems.**

3. It appears that O-substitution might favor the formation of nanoscopic oxide domains, which could potentially facilitate interfacial Li-ion diffusion. Could this explain why the Li₂O-ZrCl₄ derived material exhibits higher ionic conductivity than the Li₂S-ZrCl₄ derived material? The reviewer suggested the author provide more discussion on it.

Response to comment 3: We appreciate the reviewer’s insightful question regarding whether the enhanced ionic conductivity in the $0.8\text{Li}_2\text{O}-\text{ZrCl}_4$ may be attributed to nanoscopic interfacial effects, and why this system exhibits higher conductivity compared to the $0.8\text{Li}_2\text{S}-\text{ZrCl}_4$.

The phenomenon mentioned by the reviewer, namely the formation of nanoscopic domains due to oxygen incorporation, is indeed valid. It is presumed that oxygen-substituted hcp phases are present across many interfacial regions, establishing interconnected domains that facilitate enhanced ionic conduction. In contrast, the sulfur-substituted system forms a lattice within larger domains that are detectable even by lab-based XRD sources (**Fig. 1d**).

Fundamentally, the two materials start with different structural origins. The pristine hcp- Li_2ZrCl_6 exhibits an ionic conductivity of approximately 0.4 mS cm^{-1} at RT, and oxygen substitution results in about a four-fold improvement within the same structural framework (1.8 mS cm^{-1} at RT). In comparison, the sulfur-substituted material exhibited more than a 200-fold increase in ionic conductivity compared to pristine ccp-LZC. This result indicates that the ccp phase, which originally showed poor conductivity ($4.3 \times 10^{-3} \text{ mS cm}^{-1}$), has been transformed into a conductive structure (1.01 mS cm^{-1}).

In the case of oxygen, due to the high spontaneity of ZrO_2 formation, direct incorporation into the bulk lattice is limited. Instead, the oxygen-substituted hcp phase tends to form in nanoscopic regions where an oxygen concentration gradient arises due to the preferential formation of ZrO_2 and residual Li_2O (**Response Fig. 1 in Response to comment 2**).

More importantly, the oxygen-substituted hcp structure exhibits a significantly greater degree of Li site distortion, which plays a key role in enabling enhanced ionic mobility (**Fig. 5b**). The difference in ionic radius between S^{2-} (184 pm) and Cl^- (181 pm) is slight, whereas a much larger disparity exists with O^{2-} (140 pm). This substantial size mismatch in the case of oxygen induces greater lattice distortion, which in turn leads to more pronounced distortion of the lithium sites. The resulting highly distorted coordination environment around Li^+ plays a significant role in lowering the kinetically resolved activation energy for ion migration.^{R20}

(Added in revision) Fig. 5b CSM value of hcp- Li_2ZrCl_6 , hcp- $\text{Li}_{2.5}\text{ZrCl}_{5.5}\text{O}_{0.5}$, ccp- Li_2ZrCl_6 and ccp- $\text{Li}_{2.25}\text{ZrCl}_{5.75}\text{S}_{0.25}$. A minimum of 0 corresponds to a perfectly symmetric coordination environment and the maximum of 66.7 corresponds to infinite elongation along one direction.

On the other hand, in the case of sulfur, incorporation occurs throughout the bulk during ball milling, inducing a phase transition from hcp to ccp and resulting in a structurally distinct lattice. In the case of ccp-LZC, the single central metal Zr leads to the formation of a low conductive ccp framework, within which Li^+ migration along the ab-plane is inherently sluggish due to tight interlayer spacing and the absence of structural pillars in ab plane of Li layer.^{R10-12}

To overcome this, we introduced sulfur into the framework, where the increased repulsion between sulfur atoms leads to an expansion of the interlayer distance. This structural modulation facilitates Li^+ migration along the ab-plane, as clearly evidenced by the enhanced in-plane mean square displacement (MSD) profiles (**Supplementary Fig. 24**). Since ab-plane transport dominates ionic conductivity in the ccp-type LZC structure, this targeted expansion directly contributes to the significant enhancement in overall ionic conductivity. Collectively, the marked reductions in both E_m and E_f , along with enhanced hopping dynamics and entropy-driven mobility, serve as key factors in transforming a conduction inactive phase ($\sim 10^{-6} \text{ S cm}^{-1}$) into a highly conductive one ($\sim 10^{-3} \text{ S cm}^{-1}$).

(Added in revision) Supplementary Fig. 24. Li probability density at 600 K in ~ 300 ps (isosurface value $P = P_{\text{max}}/100$) of $\text{ccp-Li}_2\text{ZrCl}_6$ (top-left) and $\text{ccp-Li}_{2.25}\text{ZrCl}_{5.75}\text{S}_{0.25}$ (top-right). Mean square displacement (MSD) of AIMD simulations at 600 K in ~ 300 ps of $\text{ccp-Li}_2\text{ZrCl}_6$ (down-left) and $\text{ccp-Li}_{2.25}\text{ZrCl}_{5.75}\text{S}_{0.25}$ (down-right). Sulfur incorporation induces lattice expansion and structural distortion in the ccp framework, activating three-dimensional ionic conduction otherwise limited to the ab plane.

To provide a more comprehensive interpretation of our impedance data, we have adopted the analytical framework proposed by Li et al.^{R8} which enables the decoupling of critical parameters governing ionic conductivity in solid-state electrolytes. Following this approach, we systematically extracted and analyzed the following key parameters from our experimental data as presented in **Supplementary Fig. 18** and **Supplementary Table 20**.

(Added in revision) Supplementary Fig. 18. Arrhenius plot of hopping frequency, carrier concentration factor, migration entropy of hcp-Li₂ZrCl₆, 0.8Li₂O–ZrCl₄, ccp-Li₂ZrCl₆ and 0.8Li₂S–ZrCl₄.

(Added in revision) Supplementary Table 20. The calculated values related to ionic conductivity (pre-factor σ_0 , activation energy E_a , migration energy E_m , mobile carrier formation energy E_f , hopping frequency ν , carrier concentration factor C , migration entropy ΔS_m) of hcp-Li₂ZrCl₆, 0.8Li₂O–ZrCl₄, ccp-Li₂ZrCl₆ and 0.8Li₂S–ZrCl₄.

	hcp-Li ₂ ZrCl ₆	0.8Li ₂ O–ZrCl ₄ ,	ccp-Li ₂ ZrCl ₆	0.8Li ₂ S–ZrCl ₄
σ (25 °C)	$3.7 \times 10^{-4} \text{ S cm}^{-1}$	$1.78 \times 10^{-3} \text{ S cm}^{-1}$	$4.3 \times 10^{-6} \text{ S cm}^{-1}$	$1.01 \times 10^{-3} \text{ S cm}^{-1}$
σ_0	$77848 \text{ S cm}^{-1} \text{ K}^{-1}$	$29836 \text{ S cm}^{-1} \text{ K}^{-1}$	$99451 \text{ S cm}^{-1} \text{ K}^{-1}$	$69324 \text{ S cm}^{-1} \text{ K}^{-1}$
E_a	346.39 meV	280.96 meV	466.75 meV	317.22 meV
E_m	333.11 meV	276.03 meV	445.53 meV	315.04 meV
E_f	13.28 meV	4.93 meV	21.22 meV	2.18 meV
ν	$1.28 \times 10^6 \text{ Hz at } 0^\circ\text{C}$	$5.17 \times 10^6 \text{ Hz at } 0^\circ\text{C}$	$4.30 \times 10^4 \text{ Hz at } 0^\circ\text{C}$	$3.14 \times 10^6 \text{ Hz at } 0^\circ\text{C}$
C	$4.36 \times 10^{-8} \text{ S cm}^{-1} \text{ Hz}^{-1} \text{ K}$	$4.60 \times 10^{-8} \text{ S cm}^{-1} \text{ Hz}^{-1} \text{ K}$	$1.15 \times 10^{-8} \text{ S cm}^{-1} \text{ Hz}^{-1} \text{ K}$	$2.41 \times 10^{-8} \text{ S cm}^{-1} \text{ Hz}^{-1} \text{ K}$
ΔS_m	$1.56 \times 10^{-4} \text{ eV K}^{-1}$	$2.46 \times 10^{-4} \text{ eV K}^{-1}$	$4.42 \times 10^{-5} \text{ eV K}^{-1}$	$1.38 \times 10^{-4} \text{ eV K}^{-1}$

In both 0.8Li₂O–ZrCl₄ and 0.8Li₂S–ZrCl₄ systems, significant enhancements in Li⁺ transport are observed due to reductions in migration energy (E_m) and mobile carrier formation energy (E_f), along with increases in hopping frequency (ν) and migration entropy (ΔS_m). In the O-substituted structure, the improvement in conductivity arises primarily from lattice distortion and energy landscape flattening, rather than a substantial increase in carrier concentration, which remains nearly comparable to that of hcp-LZC. This is attributed to the local immobilization of Li⁺ by strong Li–O bonds, despite increased Li content from charge compensation.

In contrast, the sulfur substituted system shows not only similar dynamic advantages but also a significant increase in carrier concentration, suggesting a synergistic effect between enhanced mobility and carrier population. These results demonstrate that the incorporation of divalent anions induces distinct structural and dynamic modifications that are critical for enabling high ionic conductivity in Zr-based halide system.

Our analysis suggests that the superior ionic conductivity of the $0.8\text{Li}_2\text{O}-\text{ZrCl}_4$, compared to its $0.8\text{Li}_2\text{S}-\text{ZrCl}_4$, is more convincingly attributed to intrinsic lattice structural differences. Specifically, although both $0.8\text{Li}_2\text{O}-\text{ZrCl}_4$ and $0.8\text{Li}_2\text{S}-\text{ZrCl}_4$ systems exhibit sufficiently low carrier formation energies (E_f) (indicating thermodynamically favorable generation of mobile carriers) the O-substituted hcp phase outperforms the S-substituted ccp phase across multiple dynamic parameters. The hopping frequency (ν) in the oxygen structure is approximately 1.65 times higher, the carrier concentration factor (C) increases by a factor of 1.91 times higher, and the migration entropy (ΔS_m) is 1.79 times greater than in the sulfur-substituted counterpart. These enhancements indicate that the O-substituted lattice possesses more favorable vibrational freedom and a more energetically flattened landscape for Li^+ migration.

The structural differences between the two lattices are also supported by computational analysis. As shown in **Fig. 4b and 4c**, the oxygen-substituted structure exhibits a higher absolute diffusivity on the order of $10^{-4} \text{ cm}^2 \text{ s}^{-1}$, whereas the sulfur-substituted structure shows a diffusivity around $10^{-6} \text{ cm}^2 \text{ s}^{-1}$. This enhancement can be attributed to the atomic arrangement of more electronegative oxygen atoms in closer proximity within the lattice (3.29\AA for oxygen vs. 3.77\AA for sulfur), which induces greater distortion and stronger repulsive interactions. Consistent with this, the integrated crystal orbital Hamilton population (ICOHP) values indicate that the Li–Cl interactions are slightly weaker in the oxygen-substituted structure (-0.3700) compared to the sulfur-substituted counterpart (-0.3749), suggesting more facile Li^+ migration in the presence of oxygen.

These results collectively demonstrate that the superior conductivity in the oxygen-substituted system arises not from nanoscopic effect alone, but from intrinsic differences in lattice structure, dynamics, and site distortion. The smaller ionic radius and higher electronegativity of oxygen, when incorporated into the closely packed hcp lattice, lead to stronger Coulombic repulsion and greater lattice distortion due to the reduced inter-anion spacing. While both O^{2-} and S^{2-} substitutions improve transport properties, the enhanced vibrational freedom with pronounced lattice distortion and Li site distortion in the oxygen-substituted hcp phase play the dominant role in enabling higher ionic conductivity.

[Revision]

Revised Manuscript

- Line 234:** The Arrhenius plots (**Fig. 4a**) and Nyquist plots (**Supplementary Fig. 15,16**) of $0.8\text{Li}_2\text{A}-\text{ZrCl}_4$ indicate enhanced Li^+ ionic conductivity (σ_{Li^+}) of 1.78 mS cm^{-1} ($0.8\text{Li}_2\text{O}-\text{ZrCl}_4$) and 1.01 mS cm^{-1} ($0.8\text{Li}_2\text{S}-\text{ZrCl}_4$) at 25° with reduced activation energy (E_a) of 0.281 eV ($0.8\text{Li}_2\text{O}-\text{ZrCl}_4$) and 0.317 eV ($0.8\text{Li}_2\text{S}-\text{ZrCl}_4$) compared with those of Li_2ZrCl_6 (hcp: $\sigma_{\text{Li}^+} = 0.37 \text{ mS cm}^{-1}$ at 25°C with $E_a = 0.346 \text{ eV}$ and ccp: $\sigma_{\text{Li}^+} = 4.3 \times 10^{-3} \text{ mS cm}^{-1}$ at 25°C with $E_a = 0.467 \text{ eV}$). Also, EIS fitting of $0.8\text{Li}_2\text{A}-\text{ZrCl}_4$ confirms that the observed ionic conduction originates from a single-phase component, corresponding to the oxygen substituted hcp phase and the sulfur substituted ccp phase, as previously confirmed by structural analysis. (**Supplementary Fig. 17**). These findings are further corroborated by hopping rate-based conduction parameter analysis (**Supplementary Fig. 18 and Supplementary Table 20**), where we decoupled key descriptors governing Li^+ transport.³⁹ Notably, both $0.8\text{Li}_2\text{O}-\text{ZrCl}_4$ and $0.8\text{Li}_2\text{S}-\text{ZrCl}_4$ exhibited significantly reduced migration energies and elevated hopping frequencies compared to both hcp- and ccp-type Li_2ZrCl_6 . Moreover, the higher carrier concentration factor (C) and migration entropy (ΔS_m) of $0.8\text{Li}_2\text{A}-\text{ZrCl}_4$ suggest enhanced configurational dynamics and a flatter energy landscape (**Supplementary Text 6**).
- Line 276:** Indeed, the probability density at 600 K shows a wider diffusion pathway and larger regions of Li probability density in the divalent anion substituted structure (**Fig. 4e,f**). A more noteworthy observation is that sulfur substitution leads to enhanced lithium-ion conduction along the ab plane, corresponding to the Li layers, compared to ccp-LZC (**Fig. 4f, dashed line**). In ccp framework, the lattice spacing of the ab plane significantly influences the activation of lithium-ion

conduction.^{28,40} Given that the ccp-LZC exhibits limited Li⁺ transport along this plane, the lattice expansion and increased Li concentration induced by S²⁻ substitution allow the structure to reach a critical threshold necessary for enabling conduction in this previously suppressed direction (**Supplementary Fig. 24**). This highlights a key phenomenological role of sulfur substitution in facilitating multi-dimensional conduction pathways, which is experimentally supported by over two orders of magnitude increase in Li⁺ conductivity.

- **Line 287:** We performed topological analysis to investigate the channel size of the lithium-ion migration pathways due to lattice changes and observed that both divalent anions enlarged the lithium-transport channel size in each hcp and ccp structure (**Fig. 5a** and **Supplementary Fig. 25**). To further understand the structural origin of this enhancement, we performed continuous symmetry measure (CSM) analysis based on the principle that lithium-site distortion raises site energy and reduces the energy gap to the transition state, thus promoting ionic conductivity (**Fig. 5b**).⁴¹ The results reveal that the LZCS shows a modest increase in distortion relative to the original ccp phase, and the LZCO exhibits significantly higher distortion compared to the hcp framework (**Supplementary Fig. 26** and **Supplementary Table 23**). This distortion, induced by divalent anion substitution, also alters the bonding environment in the lattice.
- **Line 312:** However, though based on this fundamental principle, the dominant mechanisms affecting the structure differ depending on the divalent anion type. Oxygen in hcp lattice causes strong local distortions due to its size mismatch with Cl⁻, which significantly alters Li-site coordination and facilitates migration. Sulfur incorporation in ccp lattice follows the same mechanism but is particularly effective in activating transport along the ab-plane in the ccp-LZC structure. This is achieved by expanding interlayer spacing and slightly increasing Li content, which together enable conduction in an otherwise inactive dimension. A comparative summary of these mechanisms and their associated structural, morphological, and electrochemical effects is presented in **Scheme 1** and **Supplementary Table 25**.

Revised Manuscript (Figure)

(Added in revision) **Fig. 5b** CSM value of hcp-Li₂ZrCl₆, hcp-Li_{2.5}ZrCl_{5.5}O_{0.5}, ccp-Li₂ZrCl₆ and ccp-Li_{2.25}ZrCl_{5.75}S_{0.25}

(Added in revision) **Scheme 1**. Divalent anion driven framework regulations

Revised Supplementary Information (Figure)

(Added in revision) **Supplementary Fig. 16**. Nyquist plots of EIS results for hcp-Li₂ZrCl₆, 0.8Li₂O–ZrCl₄, ccp-Li₂ZrCl₆ and 0.8Li₂S–ZrCl₄ measured at various temperatures ranging from –20 °C to 60 °C.

(Added in revision) **Supplementary Fig. 18**. Arrhenius plot of hopping frequency, carrier concentration factor, migration entropy of hcp-Li₂ZrCl₆, 0.8Li₂O–ZrCl₄, ccp-Li₂ZrCl₆ and 0.8Li₂S–ZrCl₄.

(Added in revision) **Supplementary Fig. 24**. Li probability density at 600 K in ~300 ps (isosurface value $P = P_{\max}/100$) of ccp-Li₂ZrCl₆ (top-left) and ccp-Li_{2.25}ZrCl_{5.75}S_{0.25} (top-right). Mean square displacement (MSD) of AIMD simulations at 600 K in ~300 ps of ccp-Li₂ZrCl₆ (down-left) and ccp-Li_{2.25}ZrCl_{5.75}S_{0.25} (down-right). Sulfur incorporation induces lattice expansion and structural distortion in the ccp framework, activating three-dimensional ionic conduction otherwise limited to the ab plane.

Revised Supplementary Information (Table)

(Added in revision) **Supplementary Table 20**. The calculated values related to ionic conductivity (pre-factor σ_0 , activation energy E_a , migration energy E_m , mobile carrier formation energy E_f , hopping

frequency ν , carrier concentration factor C , migration entropy ΔS_m) of hcp-Li₂ZrCl₆, 0.8Li₂O–ZrCl₄, ccp-Li₂ZrCl₆ and 0.8Li₂S–ZrCl₄.

Revised Supplementary Information (Text)

(Added in revision) Supplementary Text 6. Ionic conduction properties

The enhanced ionic conductivity in 0.8Li₂O–ZrCl₄ primarily originates from a significant reduction in migration energy rather than a substantial increase in mobile carrier concentration. Although the formation of mobile carriers becomes more favorable due to the lower formation energy, the overall carrier concentration remains comparable to hcp-LZC, likely due to partial immobilization of Li⁺ near strongly coordinating O²⁻ sites. Instead, the key contributing factors are the increased hopping frequency and migration entropy, reflecting enhanced lattice distortion and dynamic disorder. This indicates a substantial distortion of the lattice and hopping site. In sulfur incorporation, the enhanced ionic conductivity in 0.8Li₂S–ZrCl₄ stems from both a markedly reduced migration energy and a drastically lower mobile carrier formation energy compared to ccp-LZC. Unlike the oxygen-substituted system, sulfur substitution not only facilitates easier carrier formation but also significantly increases the number of mobile Li⁺ ions participating in conduction. This is further supported by the substantial rise in hopping frequency, carrier concentration factor, and migration entropy, all of which indicate a pronounced enhancement in ion dynamics. These improvements collectively reflect a fundamental transformation of the ccp lattice, wherein sulfur substitution induces interlayer expansion and dynamic lattice flexibility, effectively activating a framework that is otherwise unfavorable for Li⁺ migration.

4. The manuscript presents simulations comparing pristine and substituted samples. The reviewer suggested expanding the discussion to elaborate specifically on the differences between O- and S-substitution effects. For example, the topological analysis reveals an expanded transport channel in the S-substituted sample compared to the O-substituted one, but why does the O-substituted sample present higher ionic conductivity?

Response to comment 4: We thank the reviewer for raising this important point. We acknowledge that while our original discussion addressed the common structural features induced by O²⁻ and S²⁻ substitution, the explicit comparison between their respective effects was insufficient and has been revised accordingly.

First, in the analysis of Li-ion transport channels using Zeo⁺⁺, it is critical to recognize that direct comparison of absolute channel sizes across different crystal structure types, such as ccp and hcp, may result in interpretational bias. This limitation arises from the structure-sensitive nature of the Voronoi decomposition algorithm, which depends heavily on the symmetry, unit cell construction, and periodic boundary conditions (PBC) of the input structure.^{R21,22} In identical composition, it can yield different accessible volumes based on input cell configuration and symmetry of crystal. Due to the requirement of lattice parameters exceeding 10 Å in all directions for reliable Li⁺ diffusion sampling, we employed the smallest possible supercell models for each phase. For hcp-LZC, this resulted in a Li₁₂Zr₆Cl₃₆ composition (from Li₆Zr₃Cl₁₈ to 1x1x2 supercell), while for ccp-LZC, the minimal supercell configuration was Li₁₆Zr₈Cl₄₈ (from Li₈Zr₄Cl₂₄ to 1x1x2 supercell). The number of atoms contained in the model structure is different and it can mean that the definition of channels and pockets (resulting network connectivity) varies across distinct topological classes in different supercell compositions.^{R23}

Although the relative increase in lattice volume is nearly comparable in both cases (approximately 1.6%), the implications for ionic conductivity differ depending on the initial structure. In the case of sulfur, the original ccp structure possessed inherently poor ionic conductivity, and the induced structural expansion appears to have surpassed the threshold required to activate ion transport in ccp arrangement. In contrast, the hcp Li₂ZrCl₆ already exhibited moderate conductivity, and the oxygen substitution further enhanced its transport performance. This suggests that the substitution effect activates ion conduction differently depending on the baseline transport activity of the lattice. While the lithium transport channel size clearly plays a

significant role in enhancing ionic conductivity within the same lattice framework, broader structural features must be considered when comparing across different crystal systems.

In addition, as shown in **Supplementary Fig. 25**, the channel size increases consistently with increasing oxygen substitutions for $x=0$ to $x=0.833$. Although this reflects an expanded lattice and a larger topological space for Li-ion migration, the Li diffusivity does not show a monotonic increase from $x=0$ to $x=0.833$ and maximized in $x=0.5$ (**Fig. 4b**). Similar features are observed in the sulfur-substituted system: although the $x=0.75$ and $x=0.875$ compositions exhibit wider transport channels than pristine ccp-LZC, their ionic conductivities are lower (**Fig. 4c, Supplementary Fig. 25**). These results indicate that, while channel size generally has a favorable impact on ionic transport, it is not the sole determining factor. A deeper discussion is required to account for other dominant mechanisms that govern conductivity differences between O^{2-} - and S^{2-} -substituted structures. The most critical distinction lies in the absolute binding environment of the Li sites. Due to the substantial difference in ionic radius between oxygen (140 pm) and chlorine (181 pm), oxygen incorporation induces more severe lattice distortion.

Response Fig. 13. Distortion index in bond length of $ZrCl_{6-x}A_x$ ($A=O, S$) polyhedral unit in hcp- $Li_{2.5}ZrCl_{5.5}O_{0.5}$ and ccp- $Li_{2.25}ZrCl_{5.75}S_{0.25}$ (Distortion index = $\frac{1}{n} \sum_{i=1}^n \frac{|d_i - d_{avg}|}{d_{avg}}$)

Response Fig. 14. Histograms for distribution of ICOHP values for Li-Cl bonds in hcp- $\text{Li}_{2.5}\text{ZrCl}_{5.5}\text{O}_{0.5}$ and ccp- $\text{Li}_{2.25}\text{ZrCl}_{5.75}\text{S}_{0.25}$ (interval width = 0.025)

As shown in **Response Fig. 13** and **Response Fig. 14**, oxygen in hcp lattice leads to more distorted polyhedral environment which drives weaker Li-Cl binding interactions. It drives more pronounced distortion of the Li coordination environment (**Fig. 5b**), both of which play a pivotal role in lowering the migration energy and facilitating Li-ion mobility. The smaller ionic radius and higher electronegativity of oxygen, when incorporated into the densely packed hcp framework, lead to stronger Coulombic repulsion between anions, greater local lattice distortion, and pronounced off-centering of Li sites. These effects collectively flatten the energy landscape and reduce the activation barrier for Li^+ migration.

While S^{2-} substitution expands the conduction channel in the ccp lattice by facilitating Li^+ migration along the ab-plane through structurally modulated interlayer expansion, the relatively modest distortion and lower Li-site asymmetry result in a less flattened energy landscape compared to that of the oxygen-substituted structure. This structural characteristic also contributed to the experimentally observed improvements in activation energy, migration energy, hopping frequency, and migration entropy, albeit to a lesser extent than in the oxygen-substituted structure.

These underlying effects explain why the oxygen-based structures consistently exhibit more favorable values across key transport parameters. Overall, the key features derived from both experimental and computational analyses in $0.8\text{Li}_2\text{O}-\text{ZrCl}_4$ and $0.8\text{Li}_2\text{S}-\text{ZrCl}_4$ are summarized in **Supplementary Table 25** and **Scheme 1**. These underlying effects reflect the fundamentally different structural formation mechanisms and resulting phases, thereby explaining why the oxygen-substituted lattice consistently exhibits more favorable values across major ion transport parameters. We sincerely thank the reviewer for encouraging a deeper comparative discussion between oxygen and sulfur substitution. This perspective greatly contributed to a more comprehensive interpretation of how each anion distinctly affects phase evolution and transport behavior at both structural and mechanistic levels.

(Added in revision) Supplementary Table 25. Key features derived from both experimental and computational analyses in $0.8\text{Li}_2\text{O}-\text{ZrCl}_4$ and $0.8\text{Li}_2\text{S}-\text{ZrCl}_4$. SRO means short-range ordering.

	$0.8\text{Li}_2\text{O}-\text{ZrCl}_4$	$0.8\text{Li}_2\text{S}-\text{ZrCl}_4$
Analysis		
Lab XRD	Weak peak	ccp or LiCl
Synchrotron XRD	hcp trigonal, residual Li_2O	ccp monoclinic

PDF	hcp trigonal, SRO of ZrO ₂	hcp trigonal, SRO of ZrS ₂
XANES	hcp-like Zr environment	ccp-like Zr environment
EXAFS	Zr-Cl bond elongation by O	Zr-Cl bond elongation by S
Raman	ZrX ₆ polyhedral feature	ZrX ₆ polyhedral feature
HRTEM	ZrO ₂ , hcp trigonal	ccp monoclinic
Phase identification		
Domain size	Nano-crystalline	Micro-crystalline
Observed residual component	Li ₂ O, ZrO ₂	ZrS ₂ (PDF)
Divalent-anion substitution	Oxygen gradient based interfacial substitution of hcp trigonal	Bulk substitution with phase transition from hcp trigonal to ccp monoclinic
Conduction Property		
σ (25 °C)	$1.78 \times 10^{-3} \text{ S cm}^{-1}$	$1.01 \times 10^{-3} \text{ S cm}^{-1}$
E_a	280.96 meV	317.22 meV
σ_0	$29836 \text{ S cm}^{-1} \text{ K}^{-1}$	$69324 \text{ S cm}^{-1} \text{ K}^{-1}$
E_m	276.03 meV	315.04 meV
E_f	4.93 meV	2.18 meV
ν	$5.17 \times 10^6 \text{ Hz at } 0 \text{ }^\circ\text{C}$	$3.14 \times 10^6 \text{ Hz at } 0 \text{ }^\circ\text{C}$
C	$4.60 \times 10^{-8} \text{ S cm}^{-1} \text{ Hz}^{-1} \text{ K}$	$2.41 \times 10^{-8} \text{ S cm}^{-1} \text{ Hz}^{-1} \text{ K}$
ΔS_m	$2.46 \times 10^{-4} \text{ eV K}^{-1}$	$1.38 \times 10^{-4} \text{ eV K}^{-1}$
DFT Calculations		
Diffusivity	$3.56 \times 10^{-6} \text{ cm}^2 \text{ s}^{-1}$	$5.49 \times 10^{-7} \text{ cm}^2 \text{ s}^{-1}$
Structural feature	Divalent anion clustering	Divalent anion clustering
Probability density	Enhanced 3D connectivity	Enhanced ab plane conduction
Lattice volume increase	1.58%	1.60%
Avg. polyhedral distortion index	0.047 (More distorted)	0.009
ICOHP_{Avg} value	-0.3700 (More weekends, more diversified)	-0.3749
CSM_{Avg} value	3.431 (More distorted)	0.398
Dominant mechanism	Severe Li site distortion	Activation of ab plane conduction
Common Feature	Divalent anion clustering, charge carrier increase, lattice expansion, lattice distortion, Li site distortion, Li-Cl bond weakening, site energy diversification, energy landscape flattening	

Divalent Anion driven Framework Regulations

Scheme 1. Divalent anion driven framework regulations

[Revision]

Revised Manuscript

- **Abstract:** Synchrotron-based X-ray analyses identify distinct anionic sublattices, and based on the clearly elucidated structure, first-principles calculations reveal that **divalent anions locally cluster within the lattice, inducing structural distortion and Li-site destabilization. These changes widen lithium conduction channels and alter the bonding environment, weakening and diversifying Li–Cl interactions. As a result, the energy landscape for lithium migration is flattened, leading to significantly enhanced ionic conduction.**
- **Line 309:** Interestingly, the lattice volume expansion, which underlies the design principles of sulfide SEs, and the Li site distortion, which is central to ionic conduction enhancement in oxide SEs, can both be achieved through the incorporation of divalent anions in monovalent halide lattices.^{31,43} However, though based on this fundamental principle, the dominant mechanisms affecting the structure differ depending on the divalent anion type. Oxygen in hcp lattice causes strong local distortions due to its size mismatch with Cl⁻, which significantly alters Li-site coordination and facilitates migration. Sulfur incorporation in ccp lattice follows the same mechanism but is particularly effective in activating transport along the ab-plane in the ccp-LZC structure. This is achieved by expanding interlayer spacing and slightly increasing Li content, which together enable conduction in an otherwise inactive dimension. A comparative summary of these mechanisms and their associated structural, morphological, and electrochemical effects is presented in **Scheme 1 and Supplementary Table 25.**
- **Discussion:** We demonstrated that divalent anions play a crucial role in regulating the framework of Zr-based SEs, facilitating the formation of superionic conductive lattices. Specifically, the oxygen source (Li₂O) promotes the nanostructuring of an oxygen-substituted hcp anionic sublattice. In contrast, the sulfur source (Li₂S) drives the formation of a sulfur-substituted ccp anionic sublattice. **Divalent anions introduced into the hcp and ccp lattice tend to form clusters, which induce lattice expansion and consequently enlarge the available topological space for lithium conduction. Moreover, Li site distortion, triggered by divalent-anion incorporation, results in weakened and diversified Li–Cl bonding environments, which contribute to the flattening of the energy landscape for ion migration. Oxygen and sulfur both contribute to enhanced ionic conduction through a common underlying mechanism involving lattice modulation induced by divalent anion. Oxygen primarily inducing severe distortion of Li sites and sulfur more effectively promoting ab-plane conduction.** Therefore, in optimized compositions of 0.8Li₂A–ZrCl₄ (A = O and S), improved Li⁺ conduction compared with LZC results in ionic conductivities of 1.78 and 1.01 mS cm⁻¹ at 25 °C in 0.8Li₂O–ZrCl₄ and 0.8Li₂S–ZrCl₄, respectively. Irrespective of the specific anion lattice type (hcp or ccp), framework regulation through the incorporation of divalent anions serves as a universal strategy to flatten the energy

landscape and enhance ionic conduction. In this mechanism, local hetero-anion clustering, induced by higher valent or aliovalent anions incorporated into the lattice, modulates the conduction framework. The advanced electrochemical properties of the optimized SEs demonstrate improved cycling stability and rate performance in ASSBs. These results establish a generalized design principle for enhancing ionic conductivity in halide SEs through anion-induced lattice regulation, which enables the development of practical, high-performance ASSBs.

Revised Manuscript (Figure)

(Added in revision) Scheme 1. Divalent anion driven framework regulation

Revised Supplementary Information (Figure)

(Added in revision) Supplementary Table 25. Key features derived from both experimental and computational analyses in $0.8\text{Li}_2\text{O-ZrCl}_4$ and $0.8\text{Li}_2\text{S-ZrCl}_4$. SRO means short-range ordering.

5. What is the electronic conductivity of O- and S- substituted electrolytes? This may vary with the divalent anion composition and affect battery stability.

Response to Comment 5: We appreciate the reviewer's insightful question regarding the electronic conductivity of the O- and S-substituted electrolytes and their potential impact on battery stability. To address this, we conducted DC polarization measurements (**Supplementary Fig. 36**) on solid electrolytes ($0.8\text{Li}_2\text{O-ZrCl}_4$, $0.8\text{Li}_2\text{S-ZrCl}_4$ and $\text{hcp-Li}_2\text{ZrCl}_6$) by assembling ionic blocking configurations (SUS|SE|SUS).

(Added in revision) Supplementary Fig. 36. DC measurement for electronic conductivity of $0.8\text{Li}_2\text{O-ZrCl}_4$, $0.8\text{Li}_2\text{S-ZrCl}_4$ and $\text{hcp-Li}_2\text{ZrCl}_6$.

The results revealed that the electronic conductivity in all cases was on the order of $10^{-10} \text{ S cm}^{-1}$, indicating highly insulating behavior. These values are consistent with those reported for halide-based solid electrolytes in previous studies²¹⁻²³ and suggest that electronic conduction is sufficiently negligible. In Zr-based halide electrolytes, as a d^0 transition metal, Zr^{4+} inherently lacks the d-electrons necessary for

establishing electronic conduction. Also, Zr^{4+} ions form six-coordinate polyhedral structures within the crystal lattice, thereby maintaining structural integrity (**Fig. 4d**). This also implies that electron transport through metal-to-metal linkages is fundamentally suppressed and it exhibits intrinsically low electronic conductivity, consequently. As a result, such structural characteristics inhibit polaron conduction and help preserve the electronically insulating nature of the material.^{R26} Therefore, we conclude that variations in divalent anion in lattice do not significantly affect the electronic conductivity within the compositional range explored in this work.

Supplementary Table 26. DC measurement result and parameters for electronic conductivity of $0.8Li_2O-ZrCl_4$, $0.8Li_2S-ZrCl_4$ and $hcp-Li_2ZrCl_6$.

Sample	D (cm)	A (cm ²)	I (A)	V (V)	R (Ω)	t (cm)	σ_e (S cm ⁻¹)	σ_e - (Avg) (S cm ⁻¹)
0.8Li₂O-ZrCl₄	1.3	1.327	2.07×10^{-9}	0.1	4.83×10^7	0.037	5.77×10^{-10}	4.60×10^{-10}
			3.22×10^{-9}	0.2	6.21×10^7		4.49×10^{-10}	
			3.81×10^{-9}	0.3	7.87×10^7		3.54×10^{-10}	
0.8Li₂S-ZrCl₄			1.08×10^{-9}	0.1	9.26×10^7	0.045	3.66×10^{-10}	3.32×10^{-10}
			2.03×10^{-9}	0.2	9.85×10^7		3.44×10^{-10}	
			2.53×10^{-9}	0.3	1.19×10^8		2.86×10^{-10}	
hcp-Li₂ZrCl₆			7.22×10^{-9}	0.1	1.39×10^8	0.048	2.61×10^{-10}	2.72×10^{-10}
			1.60×10^{-9}	0.2	1.25×10^8		2.89×10^{-10}	
			2.21×10^{-9}	0.3	1.36×10^8		2.66×10^{-10}	

[Revision]

Revised Manuscript

- **Line 333:** Furthermore, the electronic conductivity of Li_2ZrCl_6 , $0.8Li_2O-ZrCl_4$, and $0.8Li_2S-ZrCl_4$ was confirmed to be as low as $\sim 10^{-10}$ S cm⁻¹ (**Supplementary Fig. 36** and **Supplementary Table 26**).

Revised Supplementary Information (Figure)

(Added in revision) **Supplementary Fig. 36.** DC measurement for electronic conductivity of $0.8Li_2O-ZrCl_4$, $0.8Li_2S-ZrCl_4$ and $hcp-Li_2ZrCl_6$.

Revised Supplementary Information (Table)

(Added in revision) **Supplementary Table 26.** DC measurement result and parameters for electronic conductivity of $0.8Li_2O-ZrCl_4$, $0.8Li_2S-ZrCl_4$ and $hcp-Li_2ZrCl_6$.

6. Standard error bars should be added for the ionic conductivity measurement.

Response to Comment 6: We thank the reviewer for their valuable suggestion regarding the inclusion of standard error bars in the ionic conductivity measurements. In response, we have included three additional measurements for each composition, containing both thick and thin pellet samples, to ensure reproducibility

and capture measurement variation. The average values of ionic conductivity and their corresponding deviations are now presented in **Response Table 2** (oxygen) and **Response Table 3** (sulfur).

Response Table 2. Summary of Experimental Li^+ ionic conductivity (σ_{Li^+}) measurements across multiple pellets of varying thickness for each sample ($x\text{Li}_2\text{O-ZrCl}_4$, $x=0.2, 0.4, 0.6, 0.8, 1.0, 1.2, 1.6, 2.0$), including calculated average values.

Sample	Diameter (cm)	Area (cm ²)	Resistance (Ω)	Thickness (cm)	σ_{Li^+} (S cm ⁻¹)	σ_{Li^+} (Avg) (S cm ⁻¹)
0.2Li ₂ O-ZrCl ₄	1.3	1.327	1856.84	0.062	2.52×10^{-5}	2.30×10^{-5}
			2679.57	0.085	2.39×10^{-5}	
			1358.25	0.036	2.00×10^{-5}	
0.4Li ₂ O-ZrCl ₄			121.44	0.092	5.71×10^{-4}	5.85×10^{-4}
			133.89	0.106	5.96×10^{-4}	
			70.37	0.055	5.89×10^{-4}	
0.6Li ₂ O-ZrCl ₄			48.12	0.096	1.47×10^{-3}	1.43×10^{-3}
			52.63	0.101	1.45×10^{-3}	
			29.11	0.054	1.40×10^{-3}	
0.8Li ₂ O-ZrCl ₄			36.97	0.089	1.81×10^{-3}	1.80×10^{-3}
			48.61	0.114	1.77×10^{-3}	
			16.92	0.041	1.83×10^{-3}	
1.0Li ₂ O-ZrCl ₄	43.89	0.076	1.30×10^{-3}	1.33×10^{-3}		
	68.02	0.12	1.33×10^{-3}			
	21.48	0.039	1.37×10^{-3}			
1.2Li ₂ O-ZrCl ₄	109.85	0.066	4.53×10^{-4}	4.39×10^{-4}		
	131.98	0.082	4.68×10^{-4}			
	43.75	0.023	3.96×10^{-4}			
1.6Li ₂ O-ZrCl ₄	19874.94	0.076	2.88×10^{-6}	2.34×10^{-6}		
	25497.21	0.082	2.42×10^{-6}			
	14565.26	0.033	1.71×10^{-6}			
2.0Li ₂ O-ZrCl ₄	277894.56	0.092	2.49×10^{-7}	2.48×10^{-7}		
	326372.49	0.103	2.38×10^{-7}			
	162067.14	0.055	1.56×10^{-7}			

Response Table 3. Summary of Experimental Li^+ ionic conductivity (σ_{Li^+}) measurements across multiple pellets of varying thickness for each sample ($x\text{Li}_2\text{S-ZrCl}_4$, $x=0.2, 0.4, 0.6, 0.8, 1.0, 1.2, 1.6, 2.0$), including calculated average values.

Sample	Diameter (cm)	Area (cm ²)	Resistance (Ω)	Thickness (cm)	σ_{Li^+} (S cm ⁻¹)	σ_{Li^+} (Avg) (S cm ⁻¹)
0.2Li ₂ S-ZrCl ₄	1.3	1.327	47578.98	0.092	1.46×10^{-6}	1.37×10^{-6}
			50480.52	0.093	1.39×10^{-6}	
			26724.84	0.045	1.27×10^{-6}	
0.4Li ₂ S-ZrCl ₄			2501.93	0.069	2.08×10^{-5}	1.96×10^{-6}
			4988.86	0.147	2.22×10^{-5}	
			1849.84	0.039	1.59×10^{-5}	

0.6Li₂S-ZrCl₄	81.49	0.098	9.06×10^{-4}	9.67×10^{-4}
	104.05	0.137	9.92×10^{-4}	
	36.25	0.048	9.98×10^{-4}	
0.8Li₂S-ZrCl₄	49.82	0.071	1.07×10^{-3}	1.04×10^{-3}
	72.42	0.096	9.99×10^{-4}	
	28.25	0.038	1.01×10^{-3}	
1.0Li₂S-ZrCl₄	91.18	0.091	7.52×10^{-4}	7.14×10^{-4}
	101.90	0.099	7.32×10^{-4}	
	43.52	0.038	6.58×10^{-4}	
1.2Li₂S-ZrCl₄	282.11	0.086	2.30×10^{-4}	2.27×10^{-4}
	342.52	0.117	2.57×10^{-4}	
	202.75	0.052	1.93×10^{-4}	
1.6Li₂S-ZrCl₄	21485.96	0.09	3.16×10^{-6}	2.75×10^{-6}
	37948.60	0.142	2.82×10^{-6}	
	14851.42	0.045	2.28×10^{-6}	
2.0Li₂S-ZrCl₄	127496.30	0.101	5.97×10^{-7}	5.71×10^{-7}
	176372.49	0.133	5.68×10^{-7}	
	50430.22	0.044	6.57×10^{-7}	

By encompassing this measurement, error bars have been incorporated into the main conductivity plot (Fig. 1b and Fig. 1e). However, we acknowledge that due to the logarithmic scale of the y-axis, error bars for compositions with higher conductivity may not be clearly visible. To address this, we have added a supplemental figure (Supplementary Fig. 2) that focuses on the high-conductivity range, where the error bars are more clearly distinguishable. Once again, we sincerely appreciate the reviewer's thoughtful comments.

(Added in revision) Supplementary Fig. 2. Ionic conductivities of xLi₂O-ZrCl₄ and xLi₂S-ZrCl₄.

[Revision]

Revised Manuscript (Methods)

EIS data were recorded for cells under an external pressure of 35 MPa at open-circuit voltage with an amplitude of 14.2 mV and a frequency range from 7 MHz to 10 mHz using a VSP-300 (Bio-Logic). Ten data points in each decade in frequency were recorded. **Ionic conductivity values were**

obtained from four separate measurements to ensure reproducibility, and the variability among these measurements is indicated as error bars in the corresponding figures.

Revised Supplementary Information (Figure)

(Added in revision) Supplementary Fig. 2. Ionic conductivities of $x\text{Li}_2\text{O-ZrCl}_4$ and $x\text{Li}_2\text{S-ZrCl}_4$.

7. For FFT patterns in the HRTEM figures, the scale bar and the diffraction pattern should be indicated.

Response to comment 7: We thank the reviewer for the helpful suggestion. As reviewer suggested, we have added scale bars and indicated the diffraction patterns in the FFT images of the HRTEM figures.

Response Fig. 6. HRTEM image and FFT patterns of $0.8\text{Li}_2\text{O-ZrCl}_4$ and $0.8\text{Li}_2\text{S-ZrCl}_4$.

Additionally, in response to a related comment concerning beam sensitivity by another reviewer, we have clarified the detailed HRTEM imaging conditions in the revised manuscript, including the reduced accelerating voltage, low-dose acquisition strategy, and the precautions taken to minimize sample degradation. These clarifications aim to ensure transparency in our imaging methodology and support the reliability of the presented data.

[Revision]

Revised Manuscript (Methods)

For the HRTEM measurements, the samples were loaded onto a lacey Cu grid in an Ar-filled glove box, and HRTEM images with FFT patterns were obtained using a JEM-ARM 300F NEOARM (JEOL). Imaging was conducted at a reduced accelerating voltage of 160 keV, rather than the standard 300 keV, to mitigate degradation during measurement, and beam exposure was further minimized by performing search and focus procedures on separate regions and limiting the area of interest to a single-frame acquisition without additional alignment or prolonged exposure.

Revised Figure

Fig. 2e HRTEM image with FFT patterns of $0.8\text{Li}_2\text{O-ZrCl}_4$.

Fig. 3e HRTEM image with FFT patterns of $0.8\text{Li}_2\text{S}-\text{ZrCl}_4$.

8. In Figure 5d,e, it is hard to read the x-axis for each column, please set a gap between different groups of columns, and there is a typo in Figure 5a: “ $\text{Li}_{2.25}\text{ZrCl}_{5.75}\text{Cl}_{0.25}$ ”.

Response to comment 7: We thank the reviewer for pointing out the issue regarding the readability of the x-axis in **Fig. 5d,e** and Typographical error in **Fig. 5a**. In response, we have adjusted the interval width between different groups of histogram columns to improve clarity. Additionally, we have included a description of the sampling interval used for the data in the figure caption for better interpretation. We have also corrected the typographical error in Figure 5a, where “ $\text{Li}_{2.25}\text{ZrCl}_{5.75}\text{S}_{0.25}$ ” has been revised to the correct composition. We sincerely appreciate the reviewer’s attention to detail in pointing out the typographical issues.

[Revision]

Revised Manuscript (Figure)

- **Fig. 5a** Topological analysis and Li^+ -transport channel size of hcp- Li_2ZrCl_6 , hcp- $\text{Li}_{2.5}\text{ZrCl}_{5.5}\text{O}_{0.5}$, ccp- Li_2ZrCl_6 , and ccp- $\text{Li}_{2.25}\text{ZrCl}_{5.75}\text{S}_{0.25}$
- **Fig. 5e** Histograms for distribution of ICOHP values for Li-Cl bonds (interval width = 0.5) in hcp- $\text{Li}_{2.5}\text{ZrCl}_{5.5}\text{O}_{0.5}$ with hcp- Li_2ZrCl_6 .
- **Fig. 5f** Histograms for distribution of ICOHP values for Li-Cl bonds (interval width = 0.25) in ccp- $\text{Li}_{2.25}\text{ZrCl}_{5.75}\text{S}_{0.25}$ with ccp- Li_2ZrCl_6 (e).

Reviewer #4

As one of the promising Li ionic conductors, lithium transition metal halides exhibit notable electrochemical stability, yet their Li ion conductivity is generally lower than that of their competitors, such as LGPS. O substitutions have been observed to be compelling in improving their Li conductivity, which was attributed to the wider Li ion diffusion channels in these oxyhalides compared to that in pure halides. In this manuscript, the authors expanded the anionic substitution from O to S and reported a positive effect on the ionic conduction for both. Detailed synchrotron characterizations and first-principles simulations were conducted to elucidate the conduction mechanism. The work has proceeded well, and the manuscript is well organized.

Response to comment: We are thankful to the reviewer for a thorough review of our manuscript and the valuable comments.

However, I am not entirely convinced by the significance of the findings of this work as required by premier journals like Nature Communications. One of the major discoveries of this study is the trial of S substitution along with O, which indeed shows positive effects on electrolyte performance. However, the improvement (1.78 mS/cm for O and 1.01 mS/cm for S) is rather limited, especially compared to the previously reported LiMOCl₄ (M=Nb-10.4 mS/cm, Ta-12.4 mS/cm).

Response to comment: We sincerely thank the reviewer for their candid assessment and valuable perspective. We fully understand the reviewer's concern regarding the absolute conductivity values and their comparison with the state-of-the-art LiMOCl₄ systems (M = Nb, Ta).

While recent discussions in halide-based systems have addressed the formation of hcp and ccp phases based on ionic potential and the necessity of phase transitions toward ccp to enhance ionic conductivity, there has been limited consideration of divalent anion incorporation as a structural tuning strategy.^{28,29} Our work highlights the importance of this overlooked factor, demonstrating that even within the same hcp or ccp lattice framework, ionic conductivity can be dramatically altered. This represents a new avenue for structural design and provides direct evidence of how divalent anions induce critical structural effects that govern ion transport.

We would like to respectfully clarify the distinct scientific context and the contribution of our work. While our reported peak ionic conductivities (1.78 mS cm⁻¹ for O-substituted and 1.01 mS cm⁻¹ for S-substituted electrolytes) are lower than those of Nb- and Ta-based halide systems, the primary aim of our work is not solely focused on achieving record-high ionic conductivity values. Instead, our core objective is to systematically elucidate and differentiate the structural and mechanistic effects introduced by divalent anion substitutions (O²⁻ and S²⁻) within structurally distinct halide lattices (hcp and ccp) and to establish clear, transferable structural design principles.

Furthermore, we emphasize the importance and relevance of Zr-based halide systems due to their favorable cost-effectiveness, making them attractive candidates for scalable solid-state battery applications. Previous studies have reported diverse structural motifs in Zr-based oxyhalides, with the resulting phases varying depending on synthesis conditions. However, accurate phase identification remains challenging due to nanoscale domains formed by oxygen incorporation, particularly in Zr systems that intrinsically favor nanoparticulate formation.^{R21} Consequently, accurate structural interpretation requires not only conventional crystallographic techniques but also careful consideration of local and medium-range order within nanoscale domains. Our work aims to address this by providing a more systematic framework for analyzing the structural-diffusional relationships, specifically focusing on the effect of oxygen and sulfur substitution on lattice distortions and site environments.

To the best of our knowledge, this is the first comprehensive investigation directly comparing the impact of O²⁻ and S²⁻ substitution in a Zr-based halide framework, leveraging complementary structural characterization (synchrotron-based XRD, PDF, EXAFS), Raman spectroscopy, and first-principles

calculations (ab-initio molecular dynamics (AIMD) simulations, crystal orbital Hamilton population (COHP) analysis, and continuous symmetry measure (CSM) analysis). This combined experimental-computational approach provides a detailed mechanistic insight into how divalent anions influence ionic conduction through lattice distortion, site destabilization, and channel expansion.

Also, our results reveal a 200-fold enhancement in conductivity within the same structural family (S substitution in Zr-based ccp structure) and a four-fold enhancement via O substitution within the same hcp lattice, which are unprecedented structural responses among closed-packed halide electrolytes. Especially, in sulfur, this is particularly notable given that pristine Zr-based ccp halide structures typically exhibit conductivities around 10^{-6} S cm^{-1} due to their compact Li layers and low intrinsic carrier density. These findings are not only of fundamental interest in halide solid electrolyte but also offer a tunable design pathway for future high-performance and cost-effective halide SEs. While we acknowledge that the conductivity values reported here do not yet reach those of Nb- or Ta-based systems, our work explicitly offers complementary and scalable structural and mechanistic insights. These insights provide a well-defined and broadly applicable design strategy for enhancing ionic conduction in cost-effective Zr-based halide electrolytes and potentially beyond, thereby significantly expanding the chemical and structural design space available for solid-state electrolyte materials.

S substitution seems to be a bad idea for mixed anionic halide electrolyte design, which, to be honest, is not very surprising.

Response to comment: We sincerely appreciate the reviewer's honest assessment and the opportunity to clarify our contribution. We understand the reviewer's concern regarding the perceived limited benefit of sulfur. However, a key contribution of our work lies in demonstrating that sulfur incorporation can activate ionic conductivity in a Zr-based ccp framework, previously considered ionically inactive ($\sim 10^{-6}$ S cm^{-1}).

Through rigorous structural analysis including synchrotron XRD, PDF, and EXAFS (**Fig. 1f**), we clarify that the observed features are not attributable to simple LiCl formation but rather correspond to a distinct ccp phase stabilized by S^{2-} incorporation. This sulfur-driven ccp structure is particularly notable because Zr-based ccp halides have been widely regarded as structurally limited for Li-ion conduction due to inherently narrow interlayer spacings, low intrinsic Li concentration, and absence of structural pillars in the ab-plane. However, our results demonstrate that S^{2-} substitution can overcome these constraints and activate ionic conduction within the Zr-based ccp lattice. Rather than serving merely as compositional doping, sulfur substitution opens a previously inaccessible chemical space by inducing structural reorganization, site distortion, and lattice expansion driven destabilization of Li coordination environment. These effects collectively transform the ccp structure into an ionically conductive phase, thereby establishing a new structural design strategy for halide solid electrolytes.

Through combining AIMD simulations and CSM analysis with COHP analysis, we demonstrate that S substitution not only affects topological parameters but also alters interaction between Li and anion resulting migration energy landscape. This mechanistic detail provides a deeper understanding that has not been quantitatively addressed in previous literature. Sulfur plays a greater role in activating ionic conduction within the originally inactive ccp-type Zr-based framework, primarily through lattice expansion. Unlike the Li_3MX_6 -type ccp structures, the ccp- Li_2ZrCl_6 framework intrinsically lacks sufficient Li ions within the Li layer, resulting in limited conduction along the ab plane due to the absence of structural pillars and narrow interlayer spacing.^{R10-12} This structural constraint renders Li^+ migration highly unfavorable. However, our results indicate that sulfur incorporation can effectively activate ionic transport within this otherwise inactive ab plane.

This is further supported by the mean square displacement (MSD) analysis, which shows that in the ccp-LZC structure, Li^+ migration is highly restricted along the ab plane compared to the c-axis, whereas in $\text{Li}_{2.25}\text{ZrCl}_{5.75}\text{S}_{0.25}$, active and isotropic conduction occurs in all directions (**Supplementary Fig. 24**).

Additionally, the Li probability density significantly increases along the ab plane in the sulfur-substituted structure, further corroborating the enhanced in-plane transport behavior (Fig. 4f). Rather than merely increasing lattice volume, the key contribution of sulfur lies in its ability to open up migration pathways within the ab plane by modulating the lattice topology. This highlights the unique role of sulfur substitution in enabling Li⁺ mobility in Zr-based ccp frameworks. This key distinction was not sufficiently discussed in the original manuscript and has now been explicitly added in the revised version.

(Added in revision) Supplementary Fig. 24. Li probability density at 600 K in ~300 ps (isosurface value $P = P_{\max}/100$) of $\text{ccp-Li}_2\text{ZrCl}_6$ (top-left) and $\text{ccp-Li}_{2.25}\text{ZrCl}_{5.75}\text{S}_{0.25}$ (top-right). Mean square displacement (MSD) of AIMD simulations at 600 K in ~300 ps of $\text{ccp-Li}_2\text{ZrCl}_6$ (down-left) and $\text{ccp-Li}_{2.25}\text{ZrCl}_{5.75}\text{S}_{0.25}$ (down-right). Sulfur incorporation induces lattice expansion and structural distortion in the ccp framework, activating three-dimensional ionic conduction otherwise limited to the ab plane.

To the best of our knowledge, this study also represents one of the first comprehensive investigations into how a Zr-based ccp structure which previously considered inactive due to inherently poor ionic conductivity can become conductive by enabling ab plane conduction. We experimentally and computationally demonstrate that sulfur substitution effectively activates ionic transport in this ccp framework.

Importantly, sulfur is not merely a passive source of lattice expansion as just a slightly larger anion. Rather, divalent anions promote anion clustering by charge compensation of additional Li that actively drives lattice expansion, induces Li-site distortion, and significantly alters the local site energy landscape. This provides direct evidence that, even within a lattice originally composed of monovalent anions, the incorporation of divalent anions without transition of phase can induce favorable structural factors that create a more conducive environment for lithium-ion migration. We have revised the main text accordingly to highlight this point more clearly.

[Revision]

Revised Manuscript

- **Line 276:** Indeed, the probability density at 600 K shows a wider diffusion pathway and larger regions of Li probability density in the divalent anion substituted structure (**Fig. 4e,f**). A more noteworthy observation is that sulfur substitution leads to enhanced lithium-ion conduction along the ab plane, corresponding to the Li layers, compared to ccp-LZC (**Fig. 4f, dashed line**). In ccp framework, the lattice spacing of the ab plane significantly influences the activation of lithium-ion conduction.^{28,40} Given that the ccp-LZC exhibits limited Li⁺ transport along this plane, the lattice expansion and increased Li concentration induced by S²⁻ substitution allow the structure to reach a critical threshold necessary for enabling conduction in this previously suppressed direction (**Supplementary Fig. 24**). This highlights a key phenomenological role of sulfur substitution in facilitating multi-dimensional conduction pathways, which is experimentally supported by over two orders of magnitude increase in Li⁺ conductivity.
- **Line 303:** Comprehensively, the fundamental cause of the enhanced lithium-ion conduction is that the structural variation induces flattening of the energy landscape in site-to-site conduction for Li⁺ migration.⁴² This effect stems from the incorporation of clustered divalent anions, which modulate lattice framework by widening conduction channels, weakening Li–Cl interactions, and diversifying the energy states of Li sites. Both divalent anion incorporation strategies retain their conductivity-enhancing effects despite of intrinsic cationic disorder (**Supplementary Fig. 27-31 and Supplementary Table 24**).

Revised Supplementary Information (Figure)

(Added in revision) Supplementary Fig. 24. Li probability density at 600 K in ~300 ps (isosurface value $P = P_{\max}/100$) of ccp-Li₂ZrCl₆ (top-left) and ccp-Li_{2.25}ZrCl_{5.75}S_{0.25} (top-right). Mean square displacement (MSD) of AIMD simulations at 600 K in ~300 ps of ccp-Li₂ZrCl₆ (down-left) and ccp-Li_{2.25}ZrCl_{5.75}S_{0.25} (down-right). Sulfur incorporation induces lattice expansion and structural distortion in the ccp framework, activating three-dimensional ionic conduction otherwise limited to the ab plane.

Another vital section of this work is clarifying the underlying mechanism of the S/O substituted halide on Li ion conduction, where the authors weren't able to bring any new understandings but reemphasizing the possibility that the incorporated S or O may broaden the Li channel as raised by previous works.

Response to comment: We sincerely thank the reviewer for this thoughtful comment on the mechanistic interpretation. Our mechanistic interpretation goes beyond the conventional notion of channel widening. While previous studies have qualitatively proposed that anion substitution may broaden Li⁺ diffusion channels, our work offers a more comprehensive and quantitative inherent mechanistic explanation. Specifically, we demonstrate that sulfur or oxygen substitution does not merely widen the channels but fundamentally reorganizes the lattice through divalent anion clustering. This clustering emerges as a natural outcome of charge compensation for the additional lithium introduced by substitution (**Supplementary Fig. 23**). In this configuration, divalent anions are positioned in proximity, leading to a maximization of both lattice distortion and volumetric expansion (**Fig. 5a,b**).

Supplementary Fig. 23. Coordination structure of additional Li in the most stable structure of $\text{Li}_{2.125}\text{ZrCl}_{5.875}\text{S}_{0.125}$, $\text{Li}_{2.25}\text{ZrCl}_{5.75}\text{S}_{0.25}$, and $\text{Li}_{2.375}\text{ZrCl}_{5.625}\text{S}_{0.375}$.

Such structural reorganization induces significant distortion at lithium sites. Consequently, the discrete energy states associated with Li occupation become broadened into a flattened energy landscape, thereby lowering the activation barrier for ion migration. While previous studies have qualitatively suggested that anion substitution may broaden the Li^+ diffusion channels, our work provides a systematic elucidation of these linked mechanisms. By combining AIMD simulations and site distortion analysis (CSM) with electronic structure-based bond strength descriptors (ICOHP), we demonstrate that S^{2-} and O^{2-} substitutions contribute enhanced ionic conduction by inducing channel expansion, Li-site distortion, and energy landscape diversification. This multi-faceted analysis offers new mechanistic insight beyond the conventional channel-widening hypothesis.

Furthermore, while both S^{2-} and O^{2-} substitutions share a common mechanistic tendency as divalent anions, contributing to channel expansion, Li-site distortion, and diversification of Li-site energies, we newly clarify that the dominant contribution of conduction enhancement differs between the two. Due to the large ionic radius mismatch with Cl^- , O^{2-} induces more pronounced structural distortion and greater diversification of Li-site energies, which primarily lowers the migration barrier. As shown in **Response Fig. 14**, the polyhedral distortion index exhibits a pronounced increase in the O-substituted structure, whereas S substitution induces comparatively weaker distortion due to the similar ionic radius of S^{2-} (184pm) and Cl^- (181pm). Consequently, the distortion of Li sites is significantly more pronounced in the O-substituted structure than in the S-substituted one, highlighting the stronger structural perturbation introduced by oxygen (Fig. 5b).

Response Fig. 13. Distortion index in bond length of $ZrCl_{6-x}A_x$ ($A=O, S$) polyhedral unit in hcp- $Li_{2.5}ZrCl_{5.5}O_{0.5}$ and ccp- $Li_{2.25}ZrCl_{5.75}S_{0.25}$ (Distortion index = $\frac{1}{n} \sum_{i=1}^n \frac{|d_i - d_{avg}|}{d_{avg}}$)

(Added in revision) Fig. 5b CSM value of hcp- Li_2ZrCl_6 , hcp- $Li_{2.5}ZrCl_{5.5}O_{0.5}$, ccp- Li_2ZrCl_6 and ccp- $Li_{2.25}ZrCl_{5.75}S_{0.25}$. A minimum of 0 corresponds to a perfectly symmetric coordination environment and the maximum of 66.7 corresponds to infinite elongation along one direction.

In contrast, the ccp-type Li_2ZrCl_6 framework inherently suppresses Li^+ conduction along the ab plane due to insufficient Li content, narrow interlayer spacing, and the absence of structural pillars, as discussed in our previous **response to comment**. Sulfur substitution mitigates these limitations by expanding the lattice and altering the local topology, effectively opening previously inactive migration pathways. This structural activation is supported by MSD analysis and Li probability density maps, which show enhanced and isotropic Li^+ transport in the sulfur-substituted structure compared to the pristine phase. To provide a more comprehensive interpretation of our impedance data, we have adopted the analytical framework proposed by Li et al.^{R8} which enables the decoupling of critical parameters governing ionic conductivity in solid-state electrolytes. Following this approach, we systematically extracted and analyzed the following key parameters from our experimental data as presented in **Supplementary Fig. 18 and Supplementary Table 20**.

(Added in revision) **Supplementary Fig. 18.** Arrhenius plot of hopping frequency, carrier concentration factor, migration entropy of hcp-Li₂ZrCl₆, 0.8Li₂O–ZrCl₄, ccp-Li₂ZrCl₆ and 0.8Li₂S–ZrCl₄.

(Added in revision) **Supplementary Table 20.** The calculated values related to ionic conductivity (pre-factor σ_0 , activation energy E_a , migration energy E_m , mobile carrier formation energy E_f , hopping frequency ν , carrier concentration factor C , migration entropy ΔS_m) of hcp-Li₂ZrCl₆, 0.8Li₂O–ZrCl₄, ccp-Li₂ZrCl₆ and 0.8Li₂S–ZrCl₄.

	hcp-Li ₂ ZrCl ₆	0.8Li ₂ O–ZrCl ₄	ccp-Li ₂ ZrCl ₆	0.8Li ₂ S–ZrCl ₄
σ (25 °C)	$3.7 \times 10^{-4} \text{ S cm}^{-1}$	$1.78 \times 10^{-3} \text{ S cm}^{-1}$	$4.3 \times 10^{-6} \text{ S cm}^{-1}$	$1.01 \times 10^{-3} \text{ S cm}^{-1}$
σ_0	$77848 \text{ S cm}^{-1} \text{ K}^{-1}$	$29836 \text{ S cm}^{-1} \text{ K}^{-1}$	$99451 \text{ S cm}^{-1} \text{ K}^{-1}$	$69324 \text{ S cm}^{-1} \text{ K}^{-1}$
E_a	346.39 meV	280.96 meV	466.75 meV	317.22 meV
E_m	333.11 meV	276.03 meV	445.53 meV	315.04 meV
E_f	13.28 meV	4.93 meV	21.22 meV	2.18 meV
ν	$1.28 \times 10^6 \text{ Hz}$ at 0 °C	$5.17 \times 10^6 \text{ Hz}$ at 0 °C	$4.30 \times 10^4 \text{ Hz}$ at 0 °C	$3.14 \times 10^6 \text{ Hz}$ at 0 °C
C	$4.36 \times 10^{-8} \text{ S cm}^{-1} \text{ Hz}^{-1} \text{ K}$	$4.60 \times 10^{-8} \text{ S cm}^{-1} \text{ Hz}^{-1} \text{ K}$	$1.15 \times 10^{-8} \text{ S cm}^{-1} \text{ Hz}^{-1} \text{ K}$	$2.41 \times 10^{-8} \text{ S cm}^{-1} \text{ Hz}^{-1} \text{ K}$
ΔS_m	$1.56 \times 10^{-4} \text{ eV K}^{-1}$	$2.46 \times 10^{-4} \text{ eV K}^{-1}$	$4.42 \times 10^{-5} \text{ eV K}^{-1}$	$1.38 \times 10^{-4} \text{ eV K}^{-1}$

In the case of 0.8Li₂O–ZrCl₄, the migration energy ($E_m = 276.03 \text{ meV}$) is notably lower than that of hcp-LZC ($E_m = 333.11 \text{ meV}$), indicating a reduction in the energy barrier for Li⁺ migration. In addition, the mobile carrier formation energy ($E_f = 4.93 \text{ meV}$) is smaller than that of hcp-LZC ($E_f = 13.28 \text{ meV}$), suggesting that Li⁺ carriers are more readily formed in the O-substituted structure. However, the carrier concentration factor in the O-substituted structure ($C = 4.60 \times 10^{-8} \text{ S cm}^{-1} \text{ Hz}^{-1} \text{ K}^{-1}$) remains comparable to that of hcp-LZC ($C = 4.36 \times 10^{-8} \text{ S cm}^{-1} \text{ Hz}^{-1} \text{ K}^{-1}$), with only a slight increase, this marginal change suggests that the total number of detectable mobile carriers is not drastically altered. Indeed, this implies that although the total number of Li⁺ ions may have increased due to O²⁻ substitution and associated charge compensation, a slight fraction of these additional Li⁺ ions are situated in strongly coordinating environments near oxygen.

These Li–O coordinated species experience deeper local potential wells and reduced mobility compared to Li⁺ coordinated by Cl⁻, effectively making them non-detectable on the timescale probed by conductivity measurements. That is, although the formation of mobile carriers has become more favorable

(lower E_f), the number of lithium ions participating in conduction remains comparable to that of hcp-LZC or is only slightly increased (comparable C). While the amount of lithium participating in long-range transport does not substantially differ, the significant increase in both hopping frequency and migration entropy indicates enhanced lattice distortion within the structure.

This structural perturbation is further supported by the Raman spectra (**Fig. 2d**), which displays slightly broader features compared to hcp-LZC, reflecting increased dynamic disorder. It is evident that the observed improvement in ionic conductivity is primarily driven by structural regulation effects, particularly the reduction in migration energy. These energetic advantages are accompanied by a higher hopping frequency ($\nu = 5.17 \times 10^6$ Hz at 0 °C) and greater migration entropy ($\Delta S_m = 2.46 \times 10^{-4}$ eV K⁻¹), both of which contribute to faster and more thermodynamically favorable ion migration. These findings collectively demonstrate that oxygen incorporation fundamentally reshapes the local lattice environment by introducing significant structural distortion, particularly at Li sites. This distortion lowers the migration barrier and enhances dynamic lattice fluctuations, ultimately serving as the primary driver for the observed improvement in ionic conductivity. Thus, the enhanced transport properties in the O-substituted system arise not from an increase in mobile carrier concentration, but from structurally induced distortions that enhance the ionic mobility.

In case of sulfur, migration energy in 0.8Li₂S–ZrCl₄ ($E_m = 315.04$ meV) is significantly lower than that of ccp-LZC ($E_m = 445.53$ meV), indicating that S²⁻ substitution significantly flattens the energy landscape for Li⁺ migration. More notably, the formation energy of mobile carriers is drastically reduced by an order of magnitude (2.18 meV vs. 21.22 meV), which strongly suggests a fundamentally altered environment for carrier generation compared to oxygen-substituted systems. This reduction is especially meaningful in the context of ccp-type Zr-based frameworks, where a small cationic radius typically results in extremely low Li-ion conductivity^{R9}; thus, the observed low E_f implies not only the formation of mobile Li ions in such typically inactive lattices, but also a fundamental modification of the lattice framework that enables and stabilizes their presence.

Furthermore, the hopping frequency increases by nearly two orders of magnitude ($\sim 10^6$ Hz vs. $\sim 10^4$ Hz), directly contributing to the enhanced dynamic behavior of Li ions. This is accompanied by a more than two-fold increase in the carrier concentration factor ($C = 2.41 \times 10^{-8}$ S cm⁻¹ Hz K⁻¹), highlighting the synergistic effect of faster ion dynamics. In contrast to the oxygen-substituted structure, where the carrier concentration factor remains nearly unchanged, the more than two-fold increase observed here highlights not only the reduced energy required for mobile carrier formation, but also the substantially increased number of carriers actively participating in conduction.

Crucially, the migration entropy (ΔS_m) also increases significantly, by approximately 3–4 times (1.38×10^{-4} eV K⁻¹), reflecting increased vibrational freedom and dynamic lattice flexibility. Collectively, these results indicate that the lattice itself undergoes a profound transformation, fundamentally altering the characteristics of ionic transport within the structure. This dynamic lattice behavior is closely linked to the structural changes induced by sulfur substitution. To overcome the inter-layer distance, we introduced S²⁻ into the framework, where the increased repulsion between sulfur atoms leads to an expansion of the interlayer distance.

To bridge these findings, it becomes evident that although both oxygen and sulfur substitutions improve dynamic transport parameters, they operate through distinct structural mechanisms. Specifically, oxygen substitution mainly enhances the ionic conduction by inducing pronounced Li site distortion, thereby lowering the migration energy without significantly increasing the number of mobile carriers. In contrast, sulfur substitution facilitates a fundamentally different conduction regime by promoting volumetric expansion driven site destabilization, which dramatically increases both the carrier population and their mobility.

Overall, the key features derived from both experimental and computational analyses in 0.8Li₂O–ZrCl₄ and 0.8Li₂S–ZrCl₄ are summarized in **Supplementary Table 25**. These underlying effects reflect the

fundamentally different structural formation mechanisms and resulting phases. We also recognize that a more explicit discussion was needed regarding which theoretically derived factors predominantly govern the transport enhancement, and how the mechanisms differ between oxygen and sulfur substitution. To address this, we have revised the manuscript to provide a clearer mechanistic comparison between the two substitution strategies and to emphasize the new insights gained through this analysis (**Scheme 1**).

(Added in revision) Supplementary Table 25. Key features derived from both experimental and computational analyses in $0.8\text{Li}_2\text{O}-\text{ZrCl}_4$ and $0.8\text{Li}_2\text{S}-\text{ZrCl}_4$. SRO means short-range ordering.

	$0.8\text{Li}_2\text{O}-\text{ZrCl}_4$	$0.8\text{Li}_2\text{S}-\text{ZrCl}_4$
Analysis		
Lab XRD	Weak peak	ccp or LiCl
Synchrotron XRD	hcp trigonal, residual Li_2O	ccp monoclinic
PDF	hcp trigonal, SRO of ZrO_2	hcp trigonal, SRO of ZrS_2
XANES	hcp-like Zr environment	ccp-like Zr environment
EXAFS	Zr-Cl bond elongation by O	Zr-Cl bond elongation by S
Raman	ZrX_6 polyhedral feature	ZrX_6 polyhedral feature
HRTEM	ZrO_2 , hcp trigonal	ccp monoclinic
Phase identification		
Domain size	Nano-crystalline	Micro-crystalline
Observed residual component	Li_2O , ZrO_2	ZrS_2 (PDF)
Divalent-anion substitution	Oxygen gradient based interfacial substitution of hcp trigonal	Bulk substitution with phase transition from hcp trigonal to ccp monoclinic
Conduction Property		
σ (25 °C)	$1.78 \times 10^{-3} \text{ S cm}^{-1}$	$1.01 \times 10^{-3} \text{ S cm}^{-1}$
E_a	280.96 meV	317.22 meV
σ_0	$29836 \text{ S cm}^{-1} \text{ K}^{-1}$	$69324 \text{ S cm}^{-1} \text{ K}^{-1}$
E_m	276.03 meV	315.04 meV
E_f	4.93 meV	2.18 meV
ν	$5.17 \times 10^6 \text{ Hz at } 0 \text{ }^\circ\text{C}$	$3.14 \times 10^6 \text{ Hz at } 0 \text{ }^\circ\text{C}$
C	$4.60 \times 10^{-8} \text{ S cm}^{-1} \text{ Hz}^{-1} \text{ K}$	$2.41 \times 10^{-8} \text{ S cm}^{-1} \text{ Hz}^{-1} \text{ K}$
ΔS_m	$2.46 \times 10^{-4} \text{ eV K}^{-1}$	$1.38 \times 10^{-4} \text{ eV K}^{-1}$
DFT Calculations		
Diffusivity	$3.56 \times 10^{-6} \text{ cm}^2 \text{ s}^{-1}$	$5.49 \times 10^{-7} \text{ cm}^2 \text{ s}^{-1}$
Structural feature	Divalent anion clustering	Divalent anion clustering
Probability density	Enhanced 3D connectivity	Enhanced ab plane conduction
Lattice volume increase	1.58%	1.60%
Avg. polyhedral distortion index	0.047 (More distorted)	0.009
ICOHP_{Avg} value	-0.3700 (More weekends, more diversified)	-0.3749
CSM_{Avg} value	3.431 (More distorted)	0.398
Dominant mechanism	Severe Li site distortion	Activation of ab plane conduction

Common Feature

Divalent anion clustering, charge carrier increase, lattice expansion, lattice distortion, Li site distortion, Li-Cl bond weakening, site energy diversification, energy landscape flattening

Scheme 1. Divalent anion driven framework regulations

[Revision]

Revised Manuscript

- Abstract:** Synchrotron-based X-ray analyses identify distinct anionic sublattices, and based on the clearly elucidated structure, first-principles calculations reveal that **divalent anions locally cluster within the lattice, inducing structural distortion and Li-site destabilization. These changes widen lithium conduction channels and alter the bonding environment, weakening and diversifying Li-Cl interactions. As a result, the energy landscape for lithium migration is flattened, leading to significantly enhanced ionic conduction.**
- Line 234:** The Arrhenius plots (Fig. 4a) and Nyquist plots (Supplementary Fig. 15,16) of $0.8\text{Li}_2\text{A}-\text{ZrCl}_4$ indicate enhanced Li^+ ionic conductivity (σ_{Li^+}) of 1.78 mS cm^{-1} ($0.8\text{Li}_2\text{O}-\text{ZrCl}_4$) and 1.01 mS cm^{-1} ($0.8\text{Li}_2\text{S}-\text{ZrCl}_4$) at 25° with reduced activation energy (E_a) of 0.281 eV ($0.8\text{Li}_2\text{O}-\text{ZrCl}_4$) and 0.317 eV ($0.8\text{Li}_2\text{S}-\text{ZrCl}_4$) compared with those of Li_2ZrCl_6 (hcp: $\sigma_{\text{Li}^+} = 0.37 \text{ mS cm}^{-1}$ at 25°C with $E_a = 0.346 \text{ eV}$ and ccp: $\sigma_{\text{Li}^+} = 4.3 \times 10^{-3} \text{ mS cm}^{-1}$ at 25°C with $E_a = 0.467 \text{ eV}$). Also, EIS fitting of $0.8\text{Li}_2\text{A}-\text{ZrCl}_4$ confirms that the observed ionic conduction originates from a single-phase component, corresponding to the oxygen substituted hcp phase and the sulfur substituted ccp phase, as previously confirmed by structural analysis. (Supplementary Fig. 17). These findings are further corroborated by hopping rate-based conduction parameter analysis (Supplementary Fig. 18 and Supplementary Table 20), where we decoupled key descriptors governing Li^+ transport.³⁹ Notably, both $0.8\text{Li}_2\text{O}-\text{ZrCl}_4$ and $0.8\text{Li}_2\text{S}-\text{ZrCl}_4$ exhibited significantly reduced migration energies and elevated hopping frequencies compared to both hcp- and ccp-type Li_2ZrCl_6 . Moreover, the higher carrier concentration factor (C) and migration entropy (ΔS_m) of

0.8Li₂A–ZrCl₄ suggest enhanced configurational dynamics and a flatter energy landscape (**Supplementary Text 6**).

- **Line 303:** Comprehensively, the fundamental cause of the enhanced lithium-ion conduction is that the structural variation induces flattening of the energy landscape in site-to-site conduction for Li⁺ migration.⁴² This effect stems from the incorporation of clustered divalent anions, which modulate lattice framework by widening conduction channels, weakening Li–Cl interactions, and diversifying the energy states of Li sites.
- **Line 309:** Interestingly, the lattice volume expansion, which underlies the design principles of sulfide SEs, and the Li site distortion, which is central to ionic conduction enhancement in oxide SEs, can both be achieved through the incorporation of divalent anions in monovalent halide lattices.^{31,43} However, though based on this fundamental principle, the dominant mechanisms affecting the structure differ depending on the divalent anion type. Oxygen in hcp lattice causes strong local distortions due to its size mismatch with Cl[–], which significantly alters Li-site coordination and facilitates migration. Sulfur incorporation in ccp lattice follows the same mechanism but is particularly effective in activating transport along the ab-plane in the ccp-LZC structure. This is achieved by expanding interlayer spacing and slightly increasing Li content, which together enable conduction in an otherwise inactive dimension. A comparative summary of these mechanisms and their associated structural, morphological, and electrochemical effects is presented in **Scheme 1 and Supplementary Table 25**.

Revised Manuscript (Figure)

(Added in revision) **Fig. 5b** CSM value of hcp-Li₂ZrCl₆, hcp-Li_{2.5}ZrCl_{5.5}O_{0.5}, ccp-Li₂ZrCl₆ and ccp-Li_{2.25}ZrCl_{5.75}S_{0.25}. A minimum of 0 corresponds to a perfectly symmetric coordination environment and the maximum of 66.7 corresponds to infinite elongation along one direction.

(Added in revision) **Scheme 1**. Divalent anion driven framework regulation

Revised Supplementary Information (Figure)

(Added in revision) **Supplementary Fig. 18**. Arrhenius plot of hopping frequency, carrier concentration factor, migration entropy of hcp-Li₂ZrCl₆, 0.8Li₂O–ZrCl₄, ccp-Li₂ZrCl₆ and 0.8Li₂S–ZrCl₄.

Revised Supplementary Information (Table)

(Added in revision) **Supplementary Table 20**. The calculated values related to ionic conductivity (pre-factor σ_0 , activation energy E_a , migration energy E_m , mobile carrier formation energy E_f , hopping frequency ν , carrier concentration factor C , migration entropy ΔS_m) of hcp-Li₂ZrCl₆, 0.8Li₂O–ZrCl₄, ccp-Li₂ZrCl₆ and 0.8Li₂S–ZrCl₄.

(Added in revision) **Supplementary Table 25**. Key features derived from both experimental and computational analyses in 0.8Li₂O–ZrCl₄ and 0.8Li₂S–ZrCl₄. SRO means short-range ordering.

Revised Supplementary Information (Text)

(Added in revision) **Supplementary Text 6. Ionic conduction properties**

The enhanced ionic conductivity in 0.8Li₂O–ZrCl₄ primarily originates from a significant reduction in migration energy rather than a substantial increase in mobile carrier concentration. Although the formation of mobile carriers becomes more favorable due to the lower formation energy, the overall carrier

concentration remains comparable to hcp-LZC, likely due to partial immobilization of Li^+ near strongly coordinating O^{2-} sites. Instead, the key contributing factors are the increased hopping frequency and migration entropy, reflecting enhanced lattice distortion and dynamic disorder. This indicates a substantial distortion of the lattice and hopping site. In sulfur incorporation, the enhanced ionic conductivity in $0.8 \text{Li}_2\text{S}-\text{ZrCl}_4$ stems from both a markedly reduced migration energy and a drastically lower mobile carrier formation energy compared to ccp-LZC. Unlike the oxygen-substituted system, sulfur substitution not only facilitates easier carrier formation but also significantly increases the number of mobile Li^+ ions participating in conduction. This is further supported by the substantial rise in hopping frequency, carrier concentration factor, and migration entropy, all of which indicate a pronounced enhancement in ion dynamics. These improvements collectively reflect a fundamental transformation of the ccp lattice, wherein sulfur substitution induces interlayer expansion and dynamic lattice flexibility, effectively activating a framework that is otherwise unfavorable for Li^+ migration.

The authors claimed that they would put forward design principles for halide electrolytes, yet I received barely any guidance after a thorough reading of their manuscript. I would recommend its publication in alternative journals unless the authors can provide more evidence.

Response to comment: We thank the reviewer for raising this important point regarding the clarity of the design principles derived from our work. In response, we have revised the manuscript to more explicitly state the structural design guidelines that emerge from our findings. Specifically, our study offers mechanistic insights into how divalent anion substitution (e.g., O^{2-} and S^{2-}) can be strategically employed to regulate Li^+ conduction in halide-based frameworks. The following is a list of key data newly included through the revision process.

1. Raman spectra to confirm retention of polyhedral units: Raman spectroscopy was utilized to verify the preservation of discrete ZrX_6 polyhedral units, supporting the structural integrity of the halide framework.

2. Quantitative analysis of PDF and EXAFS (phase ratio and amorphous phase): Quantitative pair distribution function (PDF) and EXAFS analyses were performed to determine the crystalline-to-amorphous phase ratio and to confirm the existence of amorphous components.

3. Hopping rate-based conduction parameter analysis: Conduction-related parameters, including hopping frequency and migration entropy, were extracted from multi-temperature EIS data.

4. Interpretation of enhanced ab-plane conduction due to sulfur substitution, compared to ccp-LZC: The ab-plane conduction enhancement driven by sulfur incorporation was systematically analyzed in comparison with ccp-LZC, highlighting the structural and dynamic modifications responsible for the improved ionic transport.

5. Continuous symmetry measure (CSM) analysis based on lithium-site distortion: CSM analysis was conducted to quantify the degree of Li-site distortion, elucidating its role in modifying the local conduction environment.

6. Effect of intrinsic cationic disorder on ionic conduction: The impact of intrinsic cationic disorder on lithium-ion conduction was examined, revealing its contribution to structural heterogeneity and energy landscape modulation.

7. Integration of lattice volume expansion and Li-site distortion via divalent anion incorporation: Interestingly, both the lattice volume expansion (commonly exploited in sulfide electrolytes) and the Li-site distortion (central to oxide-based ionic conductivity enhancement) were simultaneously achieved through divalent anion incorporation into monovalent halide lattices.

8. Distinct dominant mechanisms of ionic conduction enhancement by oxygen vs. sulfur: The dominant mechanisms by which oxygen and sulfur enhance ionic conductivity were differentiated, revealing fundamentally different structural and dynamic contributions from each anion species.

These additions collectively expand and clarify the discussion by providing experimental and theoretical insights into the structural and dynamic origins of ionic conduction, highlighting the role of framework regulations, phase quantification, hopping-based transport metrics, anion-induced conduction anisotropy, local site distortion and the design strategy achievable via divalent anion incorporation in halides.

The summary of our findings is summarized as follows:

1. Accurate phase identification is critical in Zr-based halide electrolytes, which often form nanoscale domains upon anion substitution. As these systems require high-resolution structural tools (e.g., synchrotron XRD and PDF analysis) to resolve complex local structures, our work emphasizes the importance of unambiguous phase identification prior to establishing structure–property relationships.
2. When divalent anions (O^{2-} or S^{2-}) are incorporated into halide lattices without disrupting the global lattice symmetry, they tend to cluster to compensate for the increase in local Li concentration, leading to volumetric expansion and pronounced lattice distortion. This induced strain distorts the Li octahedral sites thereby weakening Li–Cl bonding interactions, destabilizing Li sites, and flattening the energy landscape to reduce the activation barrier for Li^+ migration.
3. Oxygen incorporation in hcp lattice, due to the large ionic radius mismatch between O^{2-} and Cl^- , generates substantial local lattice strain and induces severe distortion at Li sites. This not only flattens the energy landscape but also diversifies the potential energy of Li sites, thereby activating overall conduction pathways. The significant site distortion also severely weakens and diversifies Li–Cl interactions and promotes dynamic Li hopping.
4. While the degree of Li-site distortion induced by sulfur substitution is relatively less pronounced than that of oxygen, similar structural effects such as lattice distortion and site energy diversification still occur. However, it plays a crucial role in activating ion conduction within the inherently non-conductive ccp lattice. Specifically, S^{2-} incorporation promotes lattice expansion along the c-axis, effectively unlocking Li-ion conduction along the ab plane. This dimensional activation allows the system to exceed the structural threshold required for dynamic conduction, thereby enabling a dramatic enhancement of ionic conductivity within the same ccp structural framework

These observations can lead to a generalized design rule: by retaining the original lattice symmetry while substituting monovalent anions with divalent anions, local divalent anion clustering is induced, leading to lattice expansion, structural distortion, and ultimately Li-site destabilization. Importantly, we demonstrate that these effects are not limited to a specific crystal structure but consistently arise across different anion lattice types, including both hcp and ccp structures. This convergence of outcomes, despite structural distinctions, highlights high-valent anion doping as a universal strategy to flatten the energy landscape and enhance ionic conduction, offering a transferable design principle for diverse solid-state electrolyte systems.

This design principle, rooted in the fundamental differences between monovalent and higher-valent anions (particularly in their charge state and resulting electrostatic and structural effects) can be further extended to other halide systems such as fluorides and bromides, where the incorporation of divalent anions similarly induces substantial structural modulation without compromising the original phase symmetry. Furthermore, if trivalent anions such as nitrogen can be incorporated into oxide or sulfide lattices without disrupting the overall framework, they may similarly trigger significant lattice distortion, enabling the structural effects identified here to be broadly applied across diverse anionic chemistries.

We sincerely appreciate your candid evaluation, which has significantly contributed to improving the quality of our manuscript. We have clearly clarified the scope of our study in the revised manuscript. In particular, the implications of sulfur substitution and its structural effects have been elaborated in greater detail. Moreover, the underlying mechanism responsible for the enhanced ionic conductivity has been refined to

emphasize that it is not governed by a single factor but rather arises from a collective interplay of multiple structural and chemical phenomena. Through this, our study does not merely propose a new material but instead clarify the phenomenological mechanisms that can occur within maintaining halide frameworks, thereby expanding the chemical design space of halide solid electrolytes.

[Revision]

Revised Manuscript

- **Discussion:** We demonstrated that divalent anions play a crucial role in regulating the framework of Zr-based SEs, facilitating the formation of superionic conductive lattices. Specifically, the oxygen source (Li_2O) promotes the nanostructuring of an oxygen-substituted hcp anionic sublattice. In contrast, the sulfur source (Li_2S) drives the formation of a sulfur-substituted ccp anionic sublattice. **Divalent anions introduced into the hcp and ccp lattice tend to form clusters, which induce lattice expansion and consequently enlarge the available topological space for lithium conduction. Moreover, Li site distortion, triggered by divalent-anion incorporation, results in weakened and diversified Li–Cl bonding environments, which contribute to the flattening of the energy landscape for ion migration. Oxygen and sulfur both contribute to enhanced ionic conduction through a common underlying mechanism involving lattice modulation induced by divalent anion. Oxygen primarily inducing severe distortion of Li sites and sulfur more effectively promoting ab-plane conduction.** Therefore, in optimized compositions of $0.8\text{Li}_2\text{A–ZrCl}_4$ ($\text{A} = \text{O}$ and S), improved Li^+ conduction compared with LZC results in ionic conductivities of 1.78 and 1.01 mS cm^{-1} at $25 \text{ }^\circ\text{C}$ in $0.8\text{Li}_2\text{O–ZrCl}_4$ and $0.8\text{Li}_2\text{S–ZrCl}_4$, respectively. **Irrespective of the specific anion lattice type (hcp or ccp), framework regulation through the incorporation of divalent anions serves as a universal strategy to flatten the energy landscape and enhance ionic conduction. This modulation became possible through the incorporation of oxygen and sulfur into the chloride anion sublattice while maintaining the original lattice structure. In this mechanism, local hetero-anion clustering, induced by the aliovalent incorporation of higher-valent anions into the lattice, modulates the conduction properties.** The advanced electrochemical properties of the optimized SEs demonstrate improved cycling stability and rate performance in ASSBs. **These results establish a generalized design principle for enhancing ionic conductivity in halide SEs through anion-induced lattice regulation, which enables the development of practical, high-performance ASSBs.**

References

- R1. Kwak, H. et al. New cost-effective halide solid electrolytes for all-solid-state batteries: Mechanochemically prepared Fe³⁺-substituted Li₂ZrCl₆. *Adv. Energy Mater.* **11**, 2101609 (2021).
- R2. M Photiadis, G. Vibrational modes and structure of liquid and gaseous zirconium tetrachloride and of molten ZrCl₄-CsCl mixtures." *J. Chem. Soc., Dalton Trans.* **6**, 981-990 (1998).
- R3. Zhang, H. et al. Li-richening strategy in Li₂ZrCl₆ lattice towards enhanced ionic conductivity. *J. Energy Chem.* **79**, 348-356 (2023).
- R4. Quirk, J. A. & James A. D. Design principles for grain boundaries in solid-state lithium-ion conductors. *Adv. Energy Mater.* **13**, 2301114 (2023).
- R5. Rom, C. L. et al. Expanding the Phase Space for Halide-Based Solid Electrolytes: Li-Mg-Zr-Cl Spinel. *Chem. Mater.* **36**, 7283-7291 (2024).
- R6. Tron, A. et al. Insights into the chemical and electrochemical behavior of halide and sulfide electrolytes in all-solid-state batteries. *Energy Adv.* **4**, 518-529 (2025).
- R7. Lorger, S. et al. Transport and charge carrier chemistry in lithium oxide. *J. Electrochem. Soc.* **166**, A2215-A2220 (2019).
- R8. Li, Xiaona, et al. Hopping rate and migration entropy as the origin of superionic conduction within solid-state electrolytes. *J. Am. Chem. Soc.* **145**, 11701-11709 (2023).
- R9. Kwak, H. et al. Emerging halide superionic conductors for all-solid-state batteries: Design, synthesis, and practical applications. *ACS Energy Lett.* **7**, 1776-1805 (2022).
- R10. Wang, Q. et al. Designing lithium halide solid electrolytes. *Nat. Commun.* **15**, 1050 (2024).
- R11. Barker, K. et al. The importance of A-site cation chemistry in superionic halide solid electrolytes. *Nat. Commun.* **15**, 7501, (2024).
- R12. Wang, K. et al. A cost-effective and humidity-tolerant chloride solid electrolyte for lithium batteries. *Nat. Commun.* **12**, 4410, (2021).
- R13. Park, K. -H. et al. Anion engineering for stabilizing Li interstitial sites in halide solid electrolytes for all-solid-state Li batteries. *ACS Appl. Mater. Interfaces.* **15**, 58367-58376 (2023).
- R14. Liang, C. C. Conduction characteristics of the lithium iodide-aluminum oxide solid electrolytes. *J. Electrochem. Soc.* **120**, 1289 (1973).
- R15. Sata, N. et al. Mesoscopic fast ion conduction in nanometre-scale planar heterostructures." *Nature* **408**, 946-949 (2000).
- R16. Li, C., Gu, L., & Maier, J. Enhancement of the Li conductivity in LiF by introducing glass/crystal interfaces. *Adv. Func. Mater.* **22**, 1145-1149 (2012).
- R17. Schirmeisen, A. et al. Fast interfacial ionic conduction in nanostructured glass ceramics. *Phys. Rev. Lett.* **98**, 225901 (2007).
- R18. Jiang, S., & Wagner Jr, J. B. A theoretical model for composite electrolytes—II. Percolation model for ionic conductivity enhancement. *J. Phys. Chem. Solids* **56**, 1113-1124 (1995).
- R19. Wang, Y. et al. Design principles for solid-state lithium superionic conductors. *Nat. Mater.* **14**, 1026-1031 (2015).

- R20. Jun, K. et al. Lithium superionic conductors with corner-sharing frameworks." *Nat. Mater.* **21**, 924-931 (2022).
- R21. Luo, X. et al. Ionic conductivity enhancement of Li_2ZrCl_6 halide electrolytes via mechanochemical synthesis for all-solid-state lithium–metal batteries. *ACS Appl. Mater. Interfaces.* **14**, 49839-49846 (2022).
- R22. He, B. et al. CAVD, towards better characterization of void space for ionic transport analysis. *Sci. Data* **7**, 153 (2020).
- R23. Lucarini, V. From symmetry breaking to Poisson point process in 2D Voronoi tessellations: the generic nature of hexagons. *J. Stat. Phys.* **130**, 1047-1062 (2008).
- R25. Dabo, I. et al. Electrostatics in periodic boundary conditions and real-space corrections. *Phys. Rev. B.* **77**, 115139 (2008).
- R26. Goodenough, J. B. Metallic oxides. *Prog. Solid State Chem.* **5** 145-399 (1971).

RESPONSES TO REVIEWERS' COMMENTS

Dear Reviewers,

We sincerely appreciate your thoughtful and constructive feedback. Your insightful comments have been instrumental in improving the quality and rigor of our work. All changes in the revised documents are clearly marked in red for easy reference. In this response file, our responses to each comment are presented in blue, and the corresponding changes in the manuscript and supplementary information are highlighted in red.

Reviewer #1

I have read carefully the report and the changes introduced to the manuscript. These authors have addressed my initial concerns excellently. The manuscript should be considered by Nature Communications.

Response to comment: We sincerely thank the reviewer for the positive feedback and kind recommendation. We truly appreciate the recognition of our efforts in revising the manuscript.

Reviewer #2

The authors have addressed all of my comments. It can be accepted as is.

Response to comment: We sincerely thank the reviewer for the positive assessment and are grateful for the recommendation to accept the manuscript.

Reviewer #3

The authors have conducted extensive analyses and experiments to address most of the reviewers' concerns. Despite their significant efforts, the structural analysis, especially for Li₂O-ZrCl₄ system, is still not sufficiently clear.

1. The proposed structure of the Li₂O-ZrCl₄ system is not clearly described and is likely to confuse both reviewers and the broader audience of Nature Communications. Specifically, the phrase "an anion-substituted hcp structure at the interface of ZrO₂ (or Li₂O) with hcp-LZC" is ambiguous. Are the authors suggesting that oxygen substitutes only on the outer surface of the hcp-LZC crystallites, or are they referring to a heterogeneous interface between ZrO₂ and hcp-LZC?

Response to comment 1: We appreciate the reviewer's insightful observation regarding the ambiguity in our structural description.

We intend to describe the substitution as occurring near the heterogeneous interface between ZrO₂ and hcp-LZC rather than being limited to the outer surface of hcp-LZC. Importantly, oxygen substitution is neither restricted to the crystal surface (or a simple core-shell structure) nor uniformly distributed in the bulk but instead proceeds through spontaneous interfacial anion exchange at the ZrO₂ (or Li₂O)-hcp-LZC interfaces, resulting in a spatially graded oxygen distribution that underpins the observed conductivity enhancement. This interpretation is strongly supported by STEM-EELS mapping, comparative sulfur substitution experiments, and DFT interface modeling, which collectively establish the interfacial nature of oxygen incorporation and its decisive role in conductivity enhancement.

Critically, the STEM-EELS maps of the 0.8Li₂O-ZrCl₄ system show that oxygen (O) and zirconium (Zr) are concentrated in specific regions (e.g., the yellow dotted line), where the lithium (Li) signal is notably weak (**Response Fig. 1**). This compositional signature, highly indicative of a ZrO₂ phase, provides a clear visual demonstration that the system is a nanocomposite. Most importantly, the elemental maps show that oxygen substitution into the halide framework is not a uniform, bulk process. Instead, oxygen is incorporated

with a distinct concentration gradient, being highly concentrated in the hcp-LZC domains immediately adjacent to these ZrO_2 interfaces. This observation strongly suggests that the ZrO_2 phase acts as a local oxygen source or a reaction site, driving the interfacial anion exchange and the subsequent formation of the spatially graded oxygen-substituted structure.

In contrast, the STEM-EELS maps for the sulfur-substituted $0.8\text{Li}_2\text{S}-\text{ZrCl}_4$ system demonstrate a spatially uniform distribution of sulfur (S) throughout the entire domain. This uniform distribution, labeled as bulk substitution, serves as a critical control experiment. The difference in anion distribution between the oxygen and sulfur systems is pivotal: oxygen incorporation is driven by an interfacial mechanism resulting in a heterogeneous nanocomposite, whereas sulfur incorporation proceeds through a more uniform, bulk substitution. This distinct behavior further validates our argument that the oxygen-substituted system exhibits a unique structural motif characterized by spatially graded anion substitution at interfaces.

Response Fig. 1. STEM image and EELS (Li, Zr, Cl, O, S) elemental mapping (Li, Zr, Cl, O, S) of $0.8\text{Li}_2\text{O}-\text{ZrCl}_4$ (left) and $0.8\text{Li}_2\text{S}-\text{ZrCl}_4$ (right).

To assess this interfacial anion exchange, we constructed interface models and examined the spontaneous anion substitution behavior at the $\text{ZrO}_2-\text{Li}_2\text{ZrCl}_6$ interface by DFT calculations (**Response Fig 2**). Specifically, we evaluated all Miller index surfaces with absolute values ≤ 1 and selected three low-surface-energy planes for each material (Li_2ZrCl_6 : 100, 011, 101 and ZrO_2 : 100, 010, 111). Among these, the interface with the lowest work of adhesion ($W_a = \gamma_{\text{LZC}+\text{ZrO}_2} - \gamma_{\text{LZC}} - \gamma_{\text{ZrO}_2}$), where γ is surface energy) was identified as the most stable interfacial model and used for further modeling. We confirmed that the anion exchange at the interface is thermodynamically favorable, indicating a strong spontaneous driving force for oxygen incorporation into the halide framework (**Response Fig. 3**). This suggests that oxygen substitution is sufficiently accessible within the hcp structure at the interface with ZrO_2 .

Response Fig. 2. Interfacial slab model construction of Li_2ZrCl_6 (LZC) and ZrO_2

Response Fig. 3. DFT calculated structural energy and thermodynamic driving force of interfacial anion substitution between Li_2ZrCl_6 and ZrO_2

The primary origin of the enhanced conductivity is the oxygen-substituted hcp-LZC phase. While EXAFS and PDF $G(r)$ provide an average (ensemble-averaged) feature of the structure and show an overall oxygen incorporation, they are unable to resolve the spatial distribution, and isolating structural information exclusively from O-substituted phases remains heavily challenging. In contrast, our local analysis reveals that this substitution is not uniform. Within the nanocomposite of nanoscopic ZrO_2 and hcp-LZC domains, oxygen substitution occurs with a non-uniform concentration gradient, reaching its highest level at the ZrO_2 interfaces.

Although the degree of oxygen substitution varies spatially, the incorporation of oxygen into the hcp lattice positively contributes to Li^+ conduction, as evidenced by the AIMD results in **Fig. 4b**. Furthermore, the ~ 0.05 Å Zr–Cl bond shift observed in EXAFS (**Fig. 2c**) aligns with the bond elongation in $\text{Li}_{2.5}\text{ZrCl}_{5.5}\text{O}_{0.5}$ (**Supplementary Fig. 22**), reflecting that the spatially non-uniform distribution corresponds to an average substitution level of approximately 0.5 oxygen atoms per formula unit of Li_2ZrCl_6 . The interfacial oxygen substitution induces local lattice distortions and Li-site perturbations that facilitate Li^+ migration and thereby enhance ionic conductivity. This enhancement mechanism is further corroborated by our DFT calculations, which reveal the structural basis underlying the improved conductivity.

To address the ambiguity regarding our structural description, we have revised the manuscript to more precisely define our system. This clarification is crucial as it correctly characterizes the unique substitution mechanism that drives the conductivity enhancement. To provide strong experimental evidence, we have added STEM-EELS data, which clearly demonstrates the heterogeneous distribution of oxygen, to *Supplementary Information*. This data directly supports the proposed non-uniform, interfacial substitution behavior. We sincerely appreciate the reviewer for their insightful comments, which have enabled us to clarify and strengthen our structural description of the manuscript.

[Revision]

Revised Manuscript

Before revised: Overall, the nanocomposite structure, with nanoscopic domain sizes, is attributable to conductivity enhancement by forming an anion-substituted hcp structure at the interface of ZrO_2 (or Li_2O) with hcp-LZC (**Fig. 2e**).

After revised: Overall, within the nanocomposite of nanoscopic ZrO_2 (or Li_2O) and hcp-LZC domains, interfacial oxygen substitution in hcp-LZC occurs with a non-uniform concentration gradient, reaching its higher level at their interfaces of ZrO_2 (or Li_2O) with hcp-LZC (**Fig. 2e, Supplementary Fig. 9**). The primary origin of the enhanced conductivity is these oxygen-substituted hcp-LZC phases.

Revised Supplementary Information (Figure)

2. There appears to be a contradiction between the the experimentally derived structure and the simulated structure. In response to Comment 2, the authors suggest a final product of $2\text{Li}_2\text{O} + 5\text{ZrO}_2 + 10\text{Li}_2\text{ZrCl}_6$ according to the PDF and EXAFS fitting, implying a heterogeneous structure composed of ZrO_2 and hcp-LZC. If this is the case, the reviewer interprets the authors' response as implying that no oxygen was incorporated into the hcp-LZC structure. However, the computational simulations continue to focus solely on O-doped hcp-LZC, which does not align with the proposed experimental outcome.

Response to comment 2: We thank the reviewer for this insightful observation regarding the experimental interpretation and the computational model. We fully understand the concern and appreciate the opportunity to clarify our structural understanding of the $\text{Li}_2\text{O}-\text{ZrCl}_4$ system and its connection to the simulation approach.

1. Phase ratio of $0.8\text{Li}_2\text{O}-\text{ZrCl}_4$ ($2\text{Li}_2\text{O} + 5\text{ZrO}_2 + 10\text{Li}_2\text{ZrCl}_6$)

We acknowledge that the use of $2\text{Li}_2\text{O} + 5\text{ZrO}_2 + 10\text{Li}_2\text{ZrCl}_6$ could give the impression that no oxygen was incorporated into the hcp-LZC structure. The reason for expressing the composition in this manner was to represent the phase fractions rather than to provide the exact final product, and it should not be interpreted as the absence of oxygen incorporation. A more accurate description of the substitution can be written as $2\text{Li}_{2-5x}\text{O}_{1-5x}\text{Cl}_{5x} + 5\text{ZrO}_{2-2y}\text{Cl}_{2y} + 10\text{Li}_{2+x}\text{ZrCl}_{(6-x-y)}\text{O}_{(x+y)}$ ($x+y$ is approximately 0.5, x and y represent the local extent of anion exchange ($\text{O}^{2-} \leftrightarrow \text{Cl}^-$) occurring within Li_2O and ZrO_2 , respectively) Experimentally, the relative phase fractions are close to 1:2 between the $\text{ZrO}_{2-2y}\text{Cl}_{2y}$ phase and $\text{Li}_{2+x}\text{ZrCl}_{(6-x-y)}\text{O}_{(x+y)}$ hcp phase and the simplified ratio ($2\text{Li}_2\text{O} + 5\text{ZrO}_2 + 10\text{Li}_2\text{ZrCl}_6$) was intended to reflect this ratio (2:5:10 of $\text{Li}_{2-5x}\text{O}_{1-5x}\text{Cl}_{5x} : \text{ZrO}_{2-2y}\text{Cl}_{2y} : \text{Li}_{2+x}\text{ZrCl}_{(6-x-y)}\text{O}_{(x+y)}$).

As presented in **Response to Comment 1**, oxygen is incorporated into the structure in the form of a compositional gradient. The hcp phase is preserved through anion exchange between oxygen and chloride during synthesis, accompanied by the formation of a nanocomposite with hcp and oxides (ZrO_2 , Li_2O). The formulation is not intended to convey a single, fixed stoichiometry, but instead to represent a reasonable average compositional ratio within the range where hcp, ZrO_2 , and Li_2O -related phases are structurally maintained and observed to coexist. In this context, the proposed composition serves as a rational approximation that reflects the structural and chemical features of the nanocomposite, accommodating local substitutional gradients while remaining consistent with the dominant phase assemblage confirmed through XRD, PDF, and TEM analyses.

To avoid any potential misinterpretation, we have also added clarifying remarks in the revised manuscript, emphasizing that the phase ratio presented in **Supplementary Table 18** was originally intended to represent the relative amounts of ZrO_2 and hcp-LZC phases, and is now explicitly stated in **Supplementary Text 6**, newly added in this revision, not to imply the absence of oxygen incorporation into the hcp-LZC structure, where it is further clarified with the representative expression $2\text{Li}_{2-5x}\text{O}_{1-5x}\text{Cl}_{5x} + 5\text{ZrO}_{2-2y}\text{Cl}_{2y} + 10\text{Li}_{2+x}\text{ZrCl}_{(6-x-y)}\text{O}_{(x+y)}$.

2. Evidence for anion substitution

Evidence for anion substitution can be found in the Zr-K edge EXAFS data shown in **Fig. 2c** and **Supplementary Fig. 5**. To more clearly highlight the differences, we have compiled the Zr-K edge EXAFS spectra of $0.8\text{Li}_2\text{O}-\text{ZrCl}_4$, Li_2ZrCl_6 , and ZrO_2 in **Response Fig. 4**, allowing for a direct comparison of the local structural environments. Two key features can be identified from the EXAFS data. First, the peak corresponding to the Zr-Cl bond distance exhibits a shift of approximately 0.05 \AA , indicating a subtle change in the local coordination environment. Second, there is a noticeable broadening in the Zr-Zr distance characteristic of ZrO_2 . Both features are closely related to oxygen substitution within the hcp phase, reflecting local structural distortions induced by anion exchange.

Response Fig. 4. Zr K-edge EXAFS fitting data of $0.8\text{Li}_2\text{O}-\text{ZrCl}_4$, hcp- Li_2ZrCl_6 , and ZrO_2 .

The increase in the Zr-Cl bond length is attributed to the substitution of Cl^- by O^{2-} , which leads to the formation of shorter and stronger Zr-O bonds. As Zr-O bonds form, the local coordination environment around Zr becomes more distorted, and the stronger electrostatic interaction with O^{2-} results in a structural rearrangement that pushes the remaining Cl^- ligands slightly farther from the Zr center. This manifests as an increase in the average Zr-Cl bond distance observed in the EXAFS analysis. **Response Fig. 5** presents the distribution and average bond length of Zr-Cl bonds, along with the simulated radial distribution function (RDF), as a function of the oxygen substitution level in hcp-LZC based on DFT calculations. As the degree of O^{2-} substitution increases, the Zr-Cl bond lengths shift toward longer values, and the average bond length increases accordingly. Notably, the composition with $x = 0.5$ ($\text{Li}_{2.5}\text{ZrCl}_{5.5}\text{O}_{0.5}$) shows an average Zr-Cl bond length increase of approximately 0.05 \AA , which closely matches the experimental EXAFS observations.

Response Fig. 5. Histograms (left) and simulated radial distribution function (RDF) for Zr-Cl bonds in hcp- $\text{Li}_{2+x}\text{ZrCl}_{6-x}\text{O}_x$ ($x = 0, 0.167, 0.333, 0.5, 0.667, 0.833, 1$).

Also, the decreased intensity of the Zr-Cl peak in **Response Fig. 4** compared to pristine LZC originates from the partial substitution of Zr-Cl bonds by Zr-O bonds within the hcp phase. Since Zr-O bonds are shorter and possess different scattering amplitudes than Zr-Cl bonds, their incorporation reduces the relative contribution of the Zr-Cl coordination environment to the EXAFS signal. This explains the attenuation of the Zr-Cl peak **observed in Response Fig. 4**.

The broadening observed in the Zr-Zr region near 3.0 Å in the EXAFS spectrum (corresponding to ~3.5 Å in real space after accounting for phase shift) is attributed to anion substitution occurring in the ZrO_2 domains. Specifically, if oxygen substitution occurs in the hcp phase, then a reciprocal effect (chlorine substitution in ZrO_2) should be present. In this case, Zr-Zr distances mediated by larger anions such as Cl^- become longer than those mediated by O^{2-} , leading to an overall broadening of the Zr-Zr peak. As shown in **Response Fig. 6**, increasing the degree of Cl substitution in ZrO_2 results in the replacement of Zr-O-Zr linkages with Zr-Cl-Zr connections, which shifts the Zr-Zr distances to longer values and leads to observable peak broadening by approximately 0.4 Å in the RDF.

Response Fig. 6. Crystal structure and simulated RDF for Zr-Zr distance in $\text{ZrO}_{2-x}\text{Cl}_x$ ($x = 0, 0.25, 0.5, 0.75$).

Furthermore, in the PDF patterns (**Fig. 2a**), oxygen incorporation induces local lattice distortion, which diminishes the periodicity of long-range atomic ordering. This effect manifests as an accelerated attenuation of peaks in the high- r region ($r > 10$ Å). Importantly, the long-range order is not completely lost, but the distortions weaken the coherence of the lattice periodicity, resulting in lower peak intensities at extended distances. If such substitution and the associated distortions had not occurred, the long-range ordering would have followed the same trend as pristine LZC. Based on these observations, we carried out structural modeling in manuscript to describe the oxygen-substituted phase.

We sincerely thank the reviewer for this valuable comment, which has allowed us to clarify the structural interpretation and strengthen the consistency between experiment and computation. The corresponding revisions have been incorporated into the manuscript as outlined below.

[Revision]

Revised Supplementary Information (Text)

(Added in Revision) Supplementary Text 6. While the stoichiometry $2\text{Li}_2\text{O} + 5\text{ZrO}_2 + 10\text{Li}_2\text{ZrCl}_6^*$ is used to represent the dominant phase distribution after reaction, the actual structure involves partial and spatially heterogeneous anion substitution (**Supplementary Fig. 9**), which can be described by the generalized expression $2\text{Li}_{2-5x}\text{O}_{1-5x}\text{Cl}_{5x} + 5\text{ZrO}_{2-2y}\text{Cl}_{2y} + 10\text{Li}_{2+x}\text{ZrCl}_{(6-x-y)}\text{O}_{(x+y)}$ to reflect local compositional gradients within the nanocomposite.

3. Questionable PDF and EXAFS fitting details: The current treatment of local structure (i.e., as $\text{ZrO}_2 + \text{hcp-LZC}$ and $\text{ZrS}_2 + \text{ccp-LZC}$) in the PDF and EXAFS analyses is not convincing. It is unclear whether the O-doped and S-doped LZC structures obtained from the authors' calculations fail to provide reasonable fits to the PDF data. For EXAFS, further clarification is required regarding the "S02" parameter values: why is it set to 0.6 in Supplementary Table 7 but 1.0 in Supplementary Table 14? Additionally, the reported coordination numbers appear unrealistically high. For example, the total coordination number for $0.8\text{Li}_2\text{O}-\text{ZrCl}_4$ is 36, which is physically implausible. As such, the reliability of the current fitting results is highly questionable.

Response to comment 3: We sincerely appreciate the reviewer for these important and constructive comments, which have helped us improve the clarity and reliability of our structural analysis. In line with the reviewer's advice, we have revised the EXAFS fitting tables to avoid misinterpretation and added a new supplementary text section that explains our fitting strategy in detail. These revisions improve the clarity, consistency, and reliability of our structural analysis, and we believe they directly address the reviewer's concerns.

1. Rationale for excluding DFT-derived crystal structure models in PDF/EXAFS fitting

We fully acknowledge the reviewer's concern regarding the reliability of our fitting. The calculated O- and S-doped LZC structures adopt the lowest-symmetry P1 space group, representing only one fully occupied configuration without indicating partial occupancies or statistical distributions.

When such DFT-derived models were applied to PDF fitting, the large number of degrees of freedom inherent to P1 symmetry led to severe overfitting and non-unique, physically meaningless solutions, confirming that these structures cannot provide a reliable representation of experimental phase fraction. The refined parameters exhibited strong correlations and were highly sensitive to initial guess values, further confirming the unsuitability of these models for our experimental data.

In addition, the scattering signatures of the marginal O- or S-substituted P1-type Zr-[Cl/S or O] octahedra are essentially indistinguishable from those of Zr-Cl octahedra in pristine hcp- and ccp-LZC. As a result, it is practically impossible to discriminate them in the experimental PDF and EXAFS spectra. For these reasons, we excluded the DFT-derived P1 phases from the refinement and instead adopted the symmetrized ZrO_2 (A = O, S) and Li_2ZrCl_6 structures, which substantially reduce the number of parameters and yield a more robust and physically interpretable description of the phase fraction data. However, other experimental results and theoretical calculations unambiguously prove the O and S substitution in the hcp-LZC and ccp-LZC structure, respectively.

2. EXAFS fitting parameters Response regarding the S_0^2 parameter values

We thank the reviewer's critical point about the S_0^2 value and fully agree with the reviewer's comments. We previously performed EXAFS refinements by using different S_0^2 values for each composition to fine-tune the fits. However, as the reviewer correctly pointed out, this approach is not standard practice and poses limitations for quantitative comparison across different compositions. We have therefore revised the EXAFS fitting by applying a uniform amplitude reduction factor of $S_0^2 = 0.75$ for all compositions. The Supplementary Information has been updated accordingly (**Supplementary Figure 6, 12 and Supplementary Table 7, 14**). This adjustment ensures consistency and reliability of the refinements while maintaining good agreement with the earlier fitting results.

3. Coordination number

We performed the fittings by explicitly considering the occupancies and coordination numbers of each element. However, the table we originally presented omitted the occupancy values, which could potentially have led to misinterpretation of the structural model. N in **Supplementary Table 7** and **Supplementary Table 14** does not correspond to all actual coordination present in the structure but rather reflects a crystallographic multiplicity (degeneracy of atomic pairs at distance R) listed in the table.

The reviewer's cited value of 36 arises from summing all fitted N values in the table, but these do not represent literal coordination numbers. In the EXAFS scattering power as shown in **Eq. (1)**, N_j acts as an effective coordination parameter that scales the scattering amplitude, and in our fitting, it inherently includes contributions from multiplicities and site occupancies.

$$\chi(k) = \sum_j \frac{N_j S_0^2 F_j(k)}{k R_j^2} e^{-2R_j/\lambda(k)} e^{-2\sigma_j^2 k^2} \sin(2kR_j + \delta_j(k)) \quad \text{Eq. (1)}$$

Where: N_j : coordination number, S_0^2 : amplitude reduction factor, $F_j(k)$: effective back-scattering amplitude, k : photoelectron wave number, R_j : effective half path length, $\lambda(k)$: inelastic mean free path, σ_j^2 : Debye-Waller factor.

In this context, N_j is obtained by occupancy \times multiplicity (N as shown in **Supplementary Table 7** and **Supplementary Table 14**) as an effective coordination parameter that incorporates both site occupancies and the corresponding coordination environment. In EXAFS fitting model, each path amplitude is weighted by the site occupancy and multiplicity (e.g., N_j of Zr(2a)-Zr(4g) = N (multiplicity) \times 0.8 (occupancy of 2a site) \times 0.05 (occupancy of 4g site) = 0.08).

In response to the reviewer's careful comment, we have newly added **Supplementary Text 3**, which provides a detailed description of our EXAFS fitting model and explicitly incorporates occupancy-based weighting for each scattering path. In the revised version, **Supplementary Table 7** and **Supplementary Table 14** are directly cross-referenced to this supplementary text and include the variables used in the fitting: (a) crystallographic multiplicity (degeneracy of atomic pairs at distance R), (b) interatomic distance between absorber and scatterer atoms, (c) Debye–Waller factor (mean square disorder of the absorber–scatterer distance), (d) edge energy shift parameter, and (e) reliability factor. The detailed fitting procedures and the definition of N employed in the fitting are also clarified in **Supplementary Text 3**, ensuring that the reported coordination numbers are interpreted in a transparent and physically consistent manner.

[Revision]

Revised Manuscript (Methods)

Detailed fitting procedures for PDF and EXAFS are shown in **Supplementary Text 2,3**.

Revised Supplementary Information (Figure)

(Revised) Supplementary Fig. 6. Zr K-edge EXAFS results. The real/imaginary part of FT (left) and

$k^3\chi(k)$ (right) based on R-space curve fitting of $x\text{Li}_2\text{O}-\text{ZrCl}_4$ ($x = 0.6, 0.8, 1.0$). The detailed results are summarized in **Supplementary Table 7**.

(Revised) Supplementary Fig. 12. Zr K-edge EXAFS results. The real/imaginary part of FT (left) and $k^3\chi(k)$ (right) based on R-space curve fitting of $x\text{Li}_2\text{S}-\text{ZrCl}_4$ ($x = 0.6, 0.8, 1.0$). The detailed results are summarized in **Supplementary Table 14**.

Revised Supplementary Information (Table)

(Revised) Supplementary Table 7. EXAFS fitting results of $x\text{Li}_2\text{O}-\text{ZrCl}_4$ ($x=0.6, 0.8, 1.0$) corresponding to **Supplementary Fig. 6**. a) crystallographic multiplicity (degeneracy of atomic pairs at distance R). b) interatomic distance between absorber and scatterer atoms. c) Debye-Waller factor (mean square disorder of the absorber-scatterer distance). d) edge energy shift parameter. e) Reliability factor. Detailed fitting procedures and definition of N used in the fitting are shown in **Supplementary Text 3**.

(Revised) Supplementary Table 14. EXAFS fitting results of $x\text{Li}_2\text{S}-\text{ZrCl}_4$ ($x=0.6, 0.8, 1.0$) corresponding to **Supplementary Fig. 12**. a) crystallographic multiplicity (degeneracy of atomic pairs at distance R). b) interatomic distance between absorber and scatterer atoms. c) Debye-Waller factor (mean square disorder of the absorber-scatterer distance). d) edge energy shift parameter. e) Reliability factor. Detailed fitting procedures and definition of N used in the fitting are shown in **Supplementary Text 3**.

Revised Supplementary Information (Text)

(Added in revision) Supplementary Text 3. EXAFS fitting model of $\text{Li}_2\text{O}-\text{ZrCl}_4$ and $\text{Li}_2\text{S}-\text{ZrCl}_4$.

For EXAFS fitting, the crystallographic information of hcp- Li_2ZrCl_6 was taken from the Supporting Information (Table S1) of ref [38] in manuscript. Additionally, the model of ccp- Li_2ZrCl_6 was constructed by adopting Li_3YCl_6 (ICSD 29963), replacing Y with Zr, which has comparable atomic mass and structure. In hcp- Li_2ZrCl_6 , occupancies were set to 1.0 for M1 and 0.5 for M2/M3, based on PDF refinements yielding ≈ 0.5 for the latter sites. Similarly, the occupancies of Zr at the ccp- Li_2ZrCl_6 2a, 4g, and 2d sites were fixed at 0.8, 0.05, and 0.1, respectively to simplify the model.

N in **Supplementary Table 7** and **Supplementary Table 14** does not correspond to actual coordination numbers present in the structure but rather reflects a crystallographic multiplicity (degeneracy of atomic pairs at distance R) listed in the table. In the EXAFS scattering power as shown in below theoretical EXAFS scattering power equation, N_j acts as an effective coordination parameter that scales the scattering amplitude, and in our fitting, it inherently includes contributions from multiplicities and site occupancies.

$$\chi(k) = \sum_j \frac{N_j S_0^2 F_j(k)}{k R_j^2} e^{-2R_j/\lambda(k)} e^{-2\sigma_j^2 k^2} \sin(2kR_j + \delta_j(k))$$

Where: N_j : coordination number, S_0^2 : amplitude reduction factor, $F_j(k)$: effective back-scattering amplitude, k : photoelectron wave number, R_j : effective half path length, $\lambda(k)$: inelastic mean free path, σ_j^2 : Debye-Waller factor.

In this context, N_j is obtained by occupancy \times multiplicity (N as shown in Supplementary Table 7 and Supplementary Table 14) as an effective coordination parameter that incorporates both site occupancies and the corresponding coordination environment. In EXAFS fitting model, each path amplitude is weighted by the site occupancy and multiplicity (e.g., N_j of Zr(2a)-Zr(4g) = N (multiplicity) \times 0.8 (occupancy of 2a site) \times 0.05 (occupancy of 4g site) = 0.08).

We included Zr-Cl and Zr-Zr paths within 1.35 to 3.5 Å considering these occupancies for the O-doped samples. For the S-doped samples, Zr-Cl, Zr(2a)-Zr(4g), and Zr(2a)-Zr(2d) paths within 1.7 to 3.7 Å were included, while the Zr(4g)-Zr(2d) path was excluded based on theoretical EXAFS scattering power.

As scattering amplitude is proportional to the coordination number and site occupancy, we weight each path amplitude by the corresponding occupancy. All Zr–Zr paths involve identical scatterers- Zr - the element-dependent component of $F(k)$ is constant across these paths. The table below lists the pairwise products of Zr-site occupancies used as scattering power weights for the respective paths.

Path	Relative scattering power
Zr(2a)-Zr(4g)	$0.8 \times 0.05=0.04$
Zr(2a)-Zr(2d)	$0.8 \times 0.1=0.08$
Zr(4g)-Zr(2d)	$0.05 \times 0.1=0.005$

Accounting for the three Zr-site occupancies in ccp-Li₂ZrCl₆, the scattering power weight of the Zr(4g)-Zr(2d) pair amounts to ~4.17% of the combined Zr(2a)-Zr(4g) and Zr(2a)-Zr(2d) contributions ($\frac{[Zr(4g)-Zr(2d) \text{ path}]}{[Zr(2a)-Zr(4g) \text{ path}]+[Zr(2a)-Zr(2d) \text{ path}]} = \frac{0.005}{0.04+0.08} = 0.0417$). Therefore, considering marginal scattering power between Zr(4g)-Zr(2d) path, we excluded the path from the fitting.

Reviewer #4

I believe the authors have solved all the concerns I raised.

Response to comment: We greatly appreciate the reviewer's thoughtful assessment and are glad that our efforts have resolved the raised concerns.